# Reduction of vegetation-accessible water storage capacity after deforestation affects catchment travel time distributions and increases young water fractions in a headwater catchment

Markus Hrachowitz[1], Michael Stockinger[2,3], Miriam Coenders-Gerrits[1], Ruud van der Ent[1], Heye Bogena[2], Andreas Lücke[2], Christine Stumpp[3]

[1]Department of Watermanagement, Faculty of Civil Engineering and Geosciences, Delft University of Technology, Stevinweg 1, 2628CN Delft, Netherlands
[2]Institute of Bio- and Geosciences, Agrosphere Institute (IBG-3), Forschungszentrum Jülich, Wilhelm-Johnen-Straβe, 52425 Jülich, Germany
[3]Institute for Soil Physics and Rural Water Management, University of Natural Resources and Life Sciences Vienna, Muthgasse 18, 1190 Vienna, Austria

*Correspondence to*: Markus Hrachowitz (m.hrachowitz@tudelft.nl)

**Abstract.** Deforestation can considerably affect transpiration dynamics and magnitudes at the catchment-scale and thereby alter the partitioning between drainage and evaporative water fluxes released from terrestrial hydrological systems. However, it has so far remained problematic to directly link reductions in transpiration to changes in the physical properties of the system and to quantify these changes of system properties at the catchment-scale. As a consequence, it is difficult to quantify the effect of deforestation on parameters of catchment-scale hydrological models. This in turn leads to substantial uncertainties in predictions of the hydrological response after deforestation but also to a poor understanding of how deforestation affects principal descriptors of catchment-scale transport, such as travel time distributions and young water fractions. The objectives of this study in the Wüstebach experimental catchment are therefore to provide a mechanistic explanation of *why* changes in the partitioning of water fluxes can be observed after deforestation and how this further affects the storage and release dynamics of water. More specifically, we test the hypotheses that (1) post-deforestation changes in water storage dynamics and partitioning of water fluxes are largely a direct consequence of a reduction of the catchment-scale effective vegetation-accessible water storage capacity in the unsaturated root-zone ($S_{U,max}$) after deforestation and that (2) the deforestation-induced reduction of $S_{U,max}$ affects the shape of travel time distributions and results in shifts towards higher fractions of young water in the stream. Simultaneously modelling stream flow and stable water isotope dynamics using meaningfully adjusted model parameters both for the pre- and post-deforestation periods, respectively, a hydrological model with integrated tracer routine based on the concept of storage age selection functions is used to track fluxes through the system and to estimate the effects of deforestation on catchment travel time distributions and young water fractions $F_{yw}$.

It was found that deforestation led to a significant increase of stream flow, accompanied by corresponding reductions of evaporative fluxes. This is reflected by an increase of the runoff ratio from $C_R = 0.55$ to $0.68$ in the post-deforestation period despite similar climatic conditions. This reduction of evaporative fluxes could be linked to a reduction of the catchment-scale

water storage volume in the unsaturated soil ($S_{U,max}$) that is within the reach of active roots and thus accessible for vegetation transpiration from ~258 mm in the pre-deforestation period to ~ 101 mm in the post-deforestations period. The hydrological model, reflecting the changes in the parameter $S_{U,max}$ indicated that in the post-deforestation period stream water was characterized by slightly, yet statistically not significantly higher mean fractions of young water ($F_{yw}$ ~ 0.13) than in the pre-deforestation period ($F_{yw}$ ~ 0.12). In spite of these limited effects on the overall $F_{yw}$, changes were found for wet periods, during which post-deforestation fractions of young water increased to values $F_{yw}$ ~ 0.37 for individual storms. Deforestation also caused a significantly increased sensitivity of young water fractions to discharge under wet conditions from $dF_{yw}/dQ$ = 0.25 to 0.36.

Overall, this study provides quantitative evidence that deforestation resulted in changes of vegetation-accessible storage volumes $S_{U,max}$ and that these changes are not only responsible for changes in the partitioning between drainage and evaporation and thus the fundamental hydrological response characteristics of the Wüstebach catchment, but also for changes in catchment-scale tracer circulation dynamics. In particular for wet conditions, deforestation caused higher proportions of younger water to reach the stream, implying faster routing of stable isotopes and plausibly also solutes through the subsurface.

# 1 Introduction

Plant transpiration is, globally, the largest continental water flux (Jasechko, 2018). Notwithstanding considerable uncertainties (Coenders-Gerrits, 2014), its magnitude depends on the interplay between canopy water demand and subsurface water supply (Eagleson, 1982; Milly and Dunne, 1994; Donohue et al., 2007; Yang et al., 2016; Jaramillo et al., 2018; Mianabadi et al., 2019). The latter is regulated by water volumes that are within the reach of roots and can be taken up by plants. Many plant species across humid climate zones develop only rather shallow root systems (Schenk, 2005) that do not directly tap the groundwater (Fan et al., 2017). In regions that are dominated by such shallow-rooting vegetation, the pore volume between field capacity and permanent wilting point that is *within the reach of active roots* becomes a core property of many terrestrial hydrological systems (Rodriguez-Iturbe et al., 2007). This maximum vegetation-accessible water storage volume in the unsaturated root-zone of soils, hereafter referred to as vegetation-accessible water storage capacity $S_{U,max}$ [mm], constitutes a major partitioning point of water fluxes. It regulates the temporally varying ratio between drainage, such as groundwater recharge or shallow lateral flow, on the one hand and transpiration fluxes on the other hand (Savenije and Hrachowitz, 2017), which can in turn generate considerable feedback effects on downwind precipitation and drought generation (e.g. Seneviratne et al., 2013; Ellison et al., 2017; Teuling, 2018; Wang-Erlandsson et al., 2018; Wehrli et al., 2019).

Traditionally, $S_{U,max}$ is determined as the product of root-depths or root-distributions and pore water content between field capacity and permanent wilting point. Although correct in principle, this method has several weaknesses for applications at the catchment-scale as much of the required data are typically not available at sufficient levels of detail. While soil maps and the associated soil water retention curves have become globally available at resolutions < 1km (Arrouays et al., 2017; Hengl et al., 2017), they are characterized by considerable uncertainties. Similarly, direct and detailed observations of root-systems

are very scarce. They are, globally, limited to a few thousand individual plants only (e.g. Schenk and Jackson, 2002; Fan et al., 2017) and many of the observations are based on biomass extrapolations after excavating only the first meter of soil or less (Schenk and Jackson, 2003). Consequently, soil and root data largely remain inaccurate snapshots in space. As such, they are likely to be inadequate reflections of the spatial heterogeneity of soils and roots. In addition, these available data are also mostly snapshots in time and therefore disregard the adaptive behaviour of plant communities, whose compositions, and thus characteristics, at ecosystem level continuously evolve over multiple scales in space and time in response to changes in ambient conditions (e.g. Laio et al., 2006; Brunner et al., 2015; Tron et al., 2015).

There is increasing evidence that vegetation does not only actively adapt to its (changing) environment, but that it does so in an way that allows the most efficient use of available energy and resources (e.g. Guswa, 2008; Schymanski et al., 2008). The vegetation, i.e. a collective of individual different plants within an area of interest that is present at any given moment at any given location has survived past conditions. This in itself is a manifestation of the successful adaption of individual plants to their environment in the past. They have optimally allocated resources to balance sub- and above-surface growth to simultaneously meet water, nutrient and light requirements. This implies that these plants developed root-systems that, amongst other factors, ensure continuous access to *sufficient* water – but not more – to bridge dry periods. An individual plant that is not adapted to meet its water and nutrient requirements through its root-system as well as its light requirements through its foliage system in competition with other plants will disappear and be replaced by a better adapted plant. The root-system of vegetation at ecosystem level, and the associated vegetation-accessible water storage capacity $S_{U,max}$, is therefore at a dynamic equilibrium with and responding to the ever changing conditions of its environment. Similarly, any type of direct human interference with vegetation, such as deforestation, has an impact on transpiration water demand, the extent and structure of active root-systems and consequently on $S_{U,max}$ (Nijzink et al., 2016a).

For a meaningful quantification of $S_{U,max}$ at larger scales, such as the catchment-scale, it is therefore necessary to adopt a Darwinian perspective (Harman and Troch, 2014) and to estimate effective values of $S_{U,max}$, reflecting the collective and adaptive behaviour of all individual plants within a catchment. Results from many previous studies suggest, broadly speaking, three methods to do so. The first is the use of inverse approaches that treat $S_{U,max}$ as model calibration parameter (Fenicia et al., 2008; Speich et al., 2018; Bouaziz et al., 2020; Knighton et al., 2020). Alternatively, the second type of methods is based on optimality principles that maximize variables such as net primary production or carbon gain (Kleidon, 2004; Guswa, 2008; Hwang et al., 2009; Yang, et al., 2016; Speich et al., 2018, 2020), nitrogen uptake (McMurtrie et al., 2012) or transpiration rates (Collins and Bras, 2007; Sivandran and Bras, 2012). Lastly, $S_{U,max}$ and its evolution over time can be directly estimated through magnitudes of annual water deficits as determined from observed water balance data (Gentine et al., 2012; Donohue et al., 2012; Gao et al., 2014; DeBoer-Euser et al., 2016; van Oorschot et al., 2021).

For transpiration, shallow-rooting plants extract pore water of unsaturated soils that is held against gravity, i.e. between field capacity and permanent wilting point, and within the reach of roots. Significant vertical or lateral drainage only occurs at water contents above field capacity. By extracting soil water below that, transpiration therefore generates a root-zone water storage reservoir between field capacity and permanent wilting point that is characterized by a storage capacity $S_{U,max}$, i.e. a *maximum*

vegetation-accessible storage volume, and that is at any given moment filled with a specific water volume $S_U(t)$, depending on the past sequence of water inflow and release.

Storage reservoirs as $S_{U,max}$, or others such as groundwater bodies, are key for hydrological functioning (Sprenger et al., 2019b) as they provide a buffer against hydrological extremes, such as floods and droughts. With larger storage reservoirs, the hydrological memory of a system can increase as more water can be stored and held over longer periods of time (e.g.

Hrachowitz et al., 2015; Sprenger et al., 2019b). This also implies that while increased actual volumes of water stored in and thus the degree of filling of storage reservoirs, e.g. $S_U(t)$, can reduce water ages (Harman, 2015), increased sizes of storage reservoirs, e.g. $S_{U,max}$, can increase water ages, thereby both controlling catchment travel time distributions (TTD; Soulsby et al., 2010). As fundamental descriptors of hydrological functioning TTDs describe the age structure of water held in and released from catchments (Birkel et al., 2015; Rinaldo et al., 2015), which is critical for regulating solute transport and thus

nutrient and contaminant dynamics (Hrachowitz et al., 2016).

However, neither the effects of land cover change (Blöschl et al., 2019) nor the individual roles of different storage compartments in terrestrial hydrological systems are well understood (McDonnell et al., 2010; Penna et al., 2018, 2020). This is mostly a consequence of the lack of suitable observational technology to directly observe their respective volumes at larger scales. It remains therefore also unclear how deforestation affects $S_{U,max}$ (e.g. due to a less developed and complex rooting

system for subsequent younger vegetation) and how changes in $S_{U,max}$ may propagate to affect both, the partitioning of water fluxes as well as the age structure of water stored in and released from catchments as described by residence and travel time distributions.

For the study site of this paper, the Wüstebach experimental catchment (Germany), a previous study quantified the effects of deforestation on the partitioning of water fluxes (Wiekenkamp et al., 2016). It was found that forest removal significantly

reduced evaporative fluxes. This led to more persistent higher soil moisture levels and eventually to increases in stream flow. Similarly, in the same catchment, Wiekenkamp et al. (2020) found evidence for increased post-deforestation occurrence of preferential flows while Stockinger et al. (2019) reported minor post-deforestation reductions in travel times.

To establish a quantitative mechanistic link between these studies we here aim to trace back and attribute the above reported post-deforestation changes in the hydrological response of the Wüstebach to deforestation-induced changes in (subsurface)

system properties. The overall objective of this study is thus to analyse whether changes in these (subsurface) properties can explain *why* deforestation affects water flux partitioning and reduces travel times in the Wüstebach in an attempt to improve our quantitative understanding of critical zone processes (Brooks et al., 2015). Specifically we test the hypotheses that (1) post-deforestation changes in water storage dynamics and partitioning of water fluxes are largely a direct consequence of a reduction of the catchment-scale effective vegetation-accessible water storage capacity in the unsaturated root-zone ($S_{U,max}$)

after deforestation and that (2) the deforestation-induced reduction of $S_{U,max}$ affects the shape of travel time distributions and results in shifts towards higher fractions of young water in the stream.

## 2 Study site

The experimental Wüstebach headwater catchment (0.39 km$^2$; Fig. 1a) is part of the Lower Rhine/Eifel Observatory of the Terrestrial Environmental Observatories network (TERENO; Bogena et al., 2018) located in the Eifel National Park in Germany (50°30'16"N, 06°20'00"E). The catchment is characterized by a humid, temperate climate with warm summers, mild winters and a mean annual temperature of around 7°C (Zacharias et al., 2011). Mean annual precipitation is about 1200 mm yr$^{-1}$ and mean annual runoff about 700 mm yr$^{-1}$ (Fig. 2). Although most of the precipitation occurs in the winter months, the fraction that falls as snow is typically less than 10 % of the annual precipitation and snow cover is present for no more than 3-4 weeks per year.

The catchment is drained by a perennial 2$^{nd}$-order stream and extends from 595 to 630 m asl. The landscape is characterized by the gentle slopes of the surrounding hills and a flatter riparian area close to the stream, covering approximately 10 % of the catchment (Fig. 1a). The underlying bedrock is largely Devonian shales with sandstone inclusions (Richter, 2008) covered by periglacial layers (Borchardt, 2012). While cambisols dominate the hillslopes, gleysols and histosols characterize much of the riparian area (Bogena et al., 2015). The average soil depth in the catchment reaches about 1.6 m with a maximum of 2 m (Graf et al., 2014). In 1946, after the Second World War, the catchment was homogeneously and completely afforested (Fig. 1) with Sitka spruce (*Picea sitchensis*) and Norway spruce (*Picea abies*; Etmann, 2009). The maximum observed rooting depth of these spruce trees in the catchment is 50 cm and no roots were observed below this depth. In the course of the development of the area into a national park approximately 21 % of the catchment, including the entire riparian zone, were deforested in September 2013 and kept largely vegetation free since (Wiekenkamp et al., 2016; Fig. 1).

## 3 Data

### 3.1 Hydro-meteorological data

Daily hydro-meteorological data were available for the period 01/10/2009 – 30/09/2016 (Fig. 2). Precipitation $P$ [mm d$^{-1}$] and mean daily temperature $T$ [°C] were available from the Monschau-Kalterherberg meteorological station operated by the German Weather Service (Deutscher Wetterdienst DWD station 3339), located 9 km northwest of the Wüstebach catchment. The precipitation data were corrected for evaporation and wind drift losses according to Richter (1995) and as described in detail by Graf et al. (2014). Stream discharge $Q$ [mm d$^{-1}$] at the outlet of the Wüstebach was observed with a V-notch weir for low flow measurements and a Parshall flume for medium to high flows (Bogena et al., 2015). Daily potential evaporation $E_P$ [mm d$^{-1}$] was estimated using the Penman-Monteith equation. Daily depth-weighted average daily soil water content for the study period was estimated from a network of soil moisture sensors placed at 5, 20 and 50 cm depths at >100 locations across the study catchment as described by Graf et al. (2014) and Bogena et al. (2015). In addition, throughfall rates $P_E$ [mm d$^{-1}$] were

measured at one continuously forested location in the study catchment (Fig. 1) with an array of samplers as described in detail by Stockinger et al. (2015) over irregular intervals over the period 01/10/2012 – 30/09/2016.

## 3.2 Stable isotope data

Regular weekly $\delta^{18}O$ data from bulk precipitation samples collected in a cooled wet deposition gauge at the meteorological station Schleiden-Schöneseiffen (Meteomedia station) 3 km northeast of the catchment, were available for the period 01/10/2010 – 24/09/2012. After that, precipitation was sampled at half-daily intervals until 30/09/2016 using an automatic, cooled sampler (Eigenbrodt GmbH, Germany). The half-daily samples were precipitation volume-weighed to daily sampling intervals (Stockinger et al., 2016, 2017). Weekly stream water grab samples for stable water isotope analysis were taken at the outlet of the Wüstebach catchment in the 01/10/2010 – 30/09/2016 period (Fig. 3a; Bogena et al., 2020).

Isotope analysis was carried out using laser-based cavity ringdown spectrometers (L2120-i/L2130-i, Picarro Inc.). Internal standards calibrated against VSMOW, Greenland Ice Sheet Precipitation (GISP) and Standard Light Antarctic Precipitation (SLAP2) were used for calibration and to ensure long-term stability of analyses (Brand et al., 2014). The long-term precision of the analytical system was $\leq 0.1$ ‰ for $\delta^{18}O$.

## 4 Methods

To quantify effects of deforestation on $S_{U,max}$ and, due to the role of $S_{U,max}$ as a mixing volume also on the age structure of water as described by TTDs and the associated young water fractions $F_{yw}$, the following stepwise experiment was designed: (1) quantify changes in the partitioning of annual water fluxes between the pre- and the post-deforestation periods based on observed water balance data; (2) estimate the effect of these changes on the magnitudes of pre- and post-deforestation $S_{U,max}$, respectively, using the same data; (3) calibrate a hydrological model to simultaneously reproduce stream flow and stream $\delta^{18}O$ dynamics for the pre-deforestation period; (4) use the calibrated parameter sets to run the model in the post-deforestation period and evaluate the model's post-deforestation performance without further calibration; (5) re-calibrate the model for the post-deforestation period and evaluate if changes in calibrated $S_{U,max}$ (and other parameters) are plausible and reflect changes in $S_{U,max}$ directly estimated from water balance data in step (2); and finally (6) use the calibrated pre- and post-deforestation parameter sets, respectively, to track modelled water fluxes through the system and quantify changes in TTDs and $F_{yw}$ between the pre- and the post-deforestation periods.

### 4.1 Water balance-based estimation of $S_{U,max}$

To survive, plants need continuous access to water to satisfy canopy water demand. The root-systems of vegetation are therefore adapted to provide access to water volumes that correspond to annual water deficits that result from the combination of (1) the phase lag between and (2) the difference in the respective magnitudes of seasonal precipitation and solar radiation signals (Donohue et al., 2012; Gentine et al., 2012; Gao et al., 2014). On a daily basis, these water deficits $S_{D,j}(t)$ can be

estimated as the cumulative sum of daily throughfall $P_E$ [mm d$^{-1}$] minus transpiration $E_T$ [mm d$^{-1}$]. The maximum deficit $S_{D,j}$ for a specific year $j$ is then equivalent to the soil water volume that was accessible to and actually accessed by vegetation through its root system for transpiration during the dry season over that period when $E_T$ exceeded $P_E$ (deBoer-Euser et al., 2016; Nijzink et al., 2016a):

$$S_{D,j}(t) = \begin{cases} \int_{t_0}^{t} \left( P_E(t) - E_T(t) \right) dt, & if\ S_{D,j}(t) \leq 0 \\ 0, & if\ S_{D,j}(t) > 0 \end{cases}$$

(Eq. 1)

$$S_{D,j} = max\left( \left| S_{D,j}(t) \right| \right)$$

(Eq. 2)

where $t$ is the time step [d], and $t_0$ is the last preceding time step for which the storage deficit $S_{D,j}(t) = 0$. As an approximation, Equation 1 implies that if $S_{D,j}(t) = 0$, the water content in the root-accessible pore space at day $t$ is at field capacity and cannot hold additional water. If water supply then exceeds canopy water demand on that day, i.e. $P_E(t) - E_T(t) > 0$, this water surplus is drained from the root zone, e.g. to recharge groundwater or directly to the stream, and cannot be used for transpiration.

Daily throughfall $P_E$, i.e. precipitation that actually reaches the soil, was estimated on basis of the water balance of a canopy interception storage (Nijzink et al., 2016a):

$$\frac{dS_I(t)}{dt} = P(t) - E_I(t) - P_E(t)$$

(Eq. 3)

Where $E_I$ [mm d$^{-1}$] is daily interception evaporation and $S_I$ [mm] the canopy interception storage. For each time step, $E_I$ can then be computed as:

$$E_I(t) = \begin{cases} E_P(t), & if\ E_p(t)dt < S_I(t) \\ \dfrac{S_I(t)}{dt}, & if\ E_P(t)dt \geq S_I(t) \end{cases}$$

(Eq. 4)

This then further allows to estimate $P_E$ according to:

$$P_E(t) = \begin{cases} 0, & if\ S_I(t) < I_{max} \\ \dfrac{S_I(t) - I_{max}}{dt}, & if\ S_I(t) \geq I_{max} \end{cases}$$

(Eq. 5)

where $I_{max}$ [mm] is the canopy interception capacity. In the absence of more detailed information $P_E$ was estimated with a range of different interception capacities, i.e. $I_{max} = 0, 1, 2, 3,$ and 4 mm, in a sensitivity analysis approach.

Note that the catchment average $P_E$ after deforestation was estimated as the areal weighted mean of $P_E$ in the deforested area (21% of catchment area) computed with an assumed $I_{max} = 0$ mm and $P_E$ from the remaining area computed based on the above range of $I_{max}$ between 0 and 4 mm. In a next step, assuming negligible groundwater imports or exports (cf. Bouaziz et al., 2018), data errors and storage changes, long-term mean transpiration $\overline{E_T}$ was estimated according to the water balance:

$$\overline{E_T} = \overline{P_E} - \overline{Q}$$

(Eq. 6)

Where $\overline{P_E}$ [mm d$^{-1}$] is the long-term mean throughfall and $\overline{Q}$ [mm d$^{-1}$] is the long-term mean observed stream discharge. Daily transpiration $E_T$ [mm d$^{-1}$] for use in Eq. (1) is then estimated by scaling the long-term mean transpiration to the signal of daily potential evaporation to approximate the seasonal fluctuation of energy input (Bouaziz et al., 2020):

$$E_T(t) = \left(E_P(t) - E_I(t)\right) \frac{\overline{E_T}}{\overline{E_P} - \overline{E_I}}$$

(Eq. 7)

A range of previous studies provided evidence that mature forests develop root-systems that allow access to sufficiently large
pore water storage volumes $S_{U,max}$ to bridge droughts with return periods $T_R \sim 40$ years (Gao et al., 2014; deBoer-Euser et al., 2016; Nijzink et al., 2016a; Wang-Erlandsson et al., 2016). The maximum annual water deficits $S_{D,j}$ (Eq. 2) for all $j$ years in the pre-deforestation study period were therefore used to fit a Gumbel extreme value distribution (Gumbel, 1941). This subsequently allowed the estimation of a water deficit with a 40-year return period, which is for this study defined as vegetation-accessible water storage $S_{U,max}$ so that $S_{U,max} = S_{D,40yr}$.

Note that due to the limited length of the data series the $S_{U,max}$ estimates are rather uncertain and need to be understood as merely indicative approximations. This is in particular true for the post-deforestation period, where attempts to explicitly link $S_{U,max}$ to a specific return period are subject to additional uncertainty: as the catchment was not reforested and natural recovery of vegetation is negligible (see aerial images in Figure 1), it is not implausible to assume that the development of the root-system after the disturbance is far from equilibrium and likely to be actively evolving over time. Also note that although $E_T$
is, for brevity, referred to as transpiration throughout this manuscript, it also contains soil evaporation. However, no explicit and quantitative distinction could be made between these two fluxes with the available data. A further critical assumption of the above method required that roots do not tap the groundwater and that water for transpiration is exclusively extracted from the unsaturated soil. In contrast to other landscapes (Fan et al., 2017; Roebroek et al., 2020), it is likely that this assumption largely holds in the Wüstebach as throughout the catchment the groundwater levels, also in the riparian zone, remains largely

below a depth of 50 cm during the relatively dry growing season (Bogena et al., 2015) when storage deficits $S_D$ typically accumulate (~ May to October) and no roots have so far been observed for the dominant *picea* species below that depth in the Wüstebach catchment. This is also broadly consistent with the results of Evaristo and McDonnell (2017), who show rather limited groundwater use by *picea* species.

## 4.2 Model architecture

A semi-distributed, process-based catchment model, iteratively customized and tested within the previously developed DYNAMITE modular modelling framework (Hrachowitz et al., 2014; Fovet et al., 2015), was adapted with additional, hydrologically passive storage volumes to allow for simultaneous representation of water fluxes and tracer transport (Hrachowitz et al., 2013) based on the general concept of storage-age selection functions (SAS; Rinaldo et al., 2015). This model type was chosen over simpler, more data-based methods (e.g. McGuire and McDonnell, 2006; Kirchner, 2016) as it did not only allow a simultaneous representation of water and tracer fluxes but also allowed to attribute observed pattern to specific process hypotheses and the associated model parameters that represent (subsurface) system properties, thereby providing potential quantitative mechanistic explanations of why deforestation affects the hydrology in the Wüstebach. As an intermediate model type between purely data-driven (e.g. Kirchner, 2016) and spatially explicit physically-based models (e.g. Maxwell et al., 2016), it requires assumptions on underlying processes and effective parameters and does not allow a detailed spatial analysis. Yet this model type provides the possibility to test these process hypotheses at the scale of the semi-distributed model units thereby integrating and accounting for the natural heterogeneity of system properties across the model domain (Hrachowitz and Clark, 2017).

### 4.2.1 Hydrological model

The model domain of the Wüstebach catchment was spatially discretized into two functionally distinct response units, i.e. hillslopes and riparian areas. These are represented in the model as two parallel suites of storage components, linked by a common groundwater body as shown in Figure 4 (e.g. Euser et al., 2015; Nijzink et al., 2016b). According to elevation data and distribution of soil types (Fig.1), 90% of the catchment area was classified as hillslope and the remaining 10% as riparian area. Below a threshold temperature $T_T$ [ºC] precipitation $P$ [mm d$^{-1}$] accumulates as snow $P_S$ [mm d$^{-1}$] in $S_{Snow}$ [mm]. Above that temperature precipitation is falling as rain $P_R$ [mm d$^{-1}$] and snow melt $P_M$ [mm d$^{-1}$] is released from $S_{Snow}$ according to a melt factor $F_M$ [mm d$^{-1}$ ºC$^{-1}$] using a simple degree-day method (e.g. Arsenault et al., 2015; Ala-aho et al., 2017; Gao et al., 2017). The total liquid water input $P_R + P_M$ [mm d$^{-1}$] entering the hillslope is routed through the canopy interception storage $S_{I,H}$ [mm]. Water that is not evaporated as $E_{I,H}$ [mm d$^{-1}$] enters the unsaturated root-zone $S_{U,H}$ [mm], whose storage capacity is defined by the calibration parameter $S_{U,max,H}$ [mm]. Water can be released from $S_{U,H}$ as combined root-zone transpiration and soil evaporation flux $E_{T,H}$ [mm d$^{-1}$] or eventually recharge the groundwater $S_{S,a}$ [mm] over a fast, preferential recharge pathway as $R_{F,H}$ [mm d$^{-1}$] and a slower percolation flux $R_{S,H}$ [mm d$^{-1}$]. Similarly, water entering the riparian zone, i.e. $P_R + P_M$ [mm d$^{-1}$], is routed through $S_{I,R}$ [mm]. Excess water $P_{E,R}$ [mm d$^{-1}$] that is not evaporated infiltrates into the unsaturated root-zone $S_{U,R}$

[mm], defined by calibration parameter $S_{U,max,R}$ [mm]. In addition, a fraction of the upwelling groundwater $R_{S,R}$ [mm d$^{-1}$] replenishes $S_{U,R}$ and thus, in addition to precipitation, sustains soil moisture levels in the riparian zone (e.g. Hulsman et al., 2021a), while the remainder $Q_S$ [mm d$^{-1}$] drains directly into the stream. While water stored in $S_{U,R}$ is available for transpiration (and soil evaporation) $E_{T,R}$ [mm d$^{-1}$], water that cannot be held is released as $R_{F,R}$ [mm d$^{-1}$] to a fast responding reservoir $S_{F,R}$ [mm] from where it reaches the stream as $Q_R$ [mm d$^{-1}$]. The relevant model equations can be found in Table 1.

### 4.2.2 Tracer transport model

The δ$^{18}$O composition of water fluxes and storages was tracked through the model using the storage age selection approach (SAS; Rinaldo et al., 2015), which allows a catchment-scale description of conservative transport based on time-variant travel time distributions. The method builds on the fact that a water volume $S$ [mm] stored in any storage component can, at any moment $t$ [d], consist of parcels of water of different age $T$ [d]. The composition of ages in the stored volume at $t$ depends on the history of water inflows and outflows. Consequently, it evolves over time as new inputs enter into and outflows are released from the storage component, whereby each inflow $I$ [mm d$^{-1}$] and outflow volume $O$ [mm d$^{-1}$] can have a different age composition. A convenient way to implement the SAS approach is the use of age-ranked storage $S_T(T,t)$ [mm], which represents, "at any time $t$ the cumulative volumes of water in a storage component as ranked by their age $T$" (Benettin et al., 2017). Similarly, decomposing each inflow and outflow of a storage component into their respective cumulative, age-ranked volumes $I_T(T,t)$ and $O_T(T,t)$ [mm d$^{-1}$], respectively, then allows to update the age-ranked storage $S_T(T,t)$ at each time step according to the general water age balance (Botter et al., 2011; van der Velde et al., 2012; Benettin et al., 2015a, 2017; Harman, 2015):

$$\frac{\partial S_{T,j}(T,t)}{\partial t} + \frac{\partial S_{T,j}(T,t)}{\partial T} = \sum_{n=1}^{N} I_{T,n,j}(T,t) - \sum_{m=1}^{M} O_{T,m,j}(T,t)$$

(Eq.36)

where the term $\partial S_T / \partial T$ represents the aging of water in storage. Reflecting the slightly more abstract approach by Rodriguez and Klaus (2019) and similar to previous studies based on the functionally equivalent mixing coefficient approach (e.g. Fenicia et al., 2010; McMillan et al., 2012; Birkel and Soulsby, 2016; Hrachowitz et al., 2015), the water age balance is here individually formulated for each storage reservoir $j$ (e.g. $S_{I,H}$, $S_{U,H}$, etc.), which each can have varying numbers $N$ and $M$ of inflows $I$ (e.g. $P_R$, $P_M$, $R_{S,H}$, etc.) and outflows $O$ (e.g. $P_M$, $R_{S,H}$, $Q_S$, etc.), respectively (see Figure 4). It is assumed that the entire volume of a precipitation signal P(t) entering the system at $t$ has an age $T$ of zero so that the associated $I_{T,P,j}(T,t) = P_T(T,t) = P(t)$ for all $T$. As all other inflows to any following storage component in the system are outflows of storage components prior in the sequence (see Figure 4), the corresponding $I_{T,n,j}(T,t)$ entering a storage component are identical to the $O_{T,m,j}(T,t)$ released from the storage component above.

Each age-ranked outflow $O_{T,m,j}(T,t)$ of a specific storage component $j$ depends on the outflow volume $O_{m,j}(t)$ along this outflow pathway and the cumulative age distribution $P_{o,m,j}(T,t)$ of that outflow:

$$O_{T,m,j}(T,t) = O_{m,j}(t)P_{O,m,j}(T,t)$$

(Eq.37)

The outflow volume $O_{m,j}(t)$ is estimated via the hydrological model (see Section 4.2.1; Figure 4) and thus assumed to be known. In contrast, the cumulative age-distribution $P_{o,m,j}(T,t)$ can in general not be directly parametrized, as it depends on the temporally varying age distribution of water in the storage component $j$ represented by $S_{T,j}(T,t)$ and thus on the history of past inflows and outflows (Botter et al., 2011; Harman, 2015). Instead, it is possible to define a SAS function $\omega_{o,m,j}$ (or $\Omega_{o,m,j}$ in its cumulative form) for each outflow $m$ from each storage component $j$ that describes how outflow is sampled (or selected) from the temporally varying water volumes of different age present in the age-ranked storage $S_{T,j}(T,t)$ at any time $t$:

$$P_{O,m,j}(T,t) = \Omega_{O,m,j}\big(S_{T,j}(T,t),t\big)$$

(Eq.38)

From the cumulative age-distribution $P_{o,m,j}(T,t)$ the associated probability density function, which represents the outflow age distribution $p_{o,m,j}(T,t)$, frequently also referred to as backward travel time distribution of that outflow (TTD; e.g. Benettin et al., 2015a; Wilusz et al., 2017), can be obtained according to:

$$p_{O,m,j}(T,t) = \varpi_{O,m,j}\big(S_{T,j}(T,t),t\big)\frac{\partial S_{T,j}}{\partial T}$$

(Eq.39)

Note that conservation of mass requires that any SAS function $\omega_{O,m,j}$ integrates to the total storage volume $S_j(t)$ present in $j$ at any time $t$. To avoid the resulting need for rescaling $\omega_{O,m,j}$ at each time step, it is helpful to normalize the age-ranked storage to $S_{T,norm,j}(T,t) = S_{T,j}(T,t)/S_j(t)$ so that it remains bounded to the interval [0,1] and defines a residence time distribution (RTD). For this study beta distributions, which are conveniently bound between the limits [0,1] and defined by two shape parameters $\alpha$ and $\beta$, were used as SAS functions $\omega_{o,m,j}$ to sample water of different age for outflows from storage components. The parameters $\beta$ were fixed at a value of 1 for all SAS functions $\omega_{o,m,j}$ used here. However, there is substantial evidence for preferential flow through macropores in the shallow subsurface (e.g. Weiler and Naef, 2003; Zehe et al., 2006, 2007; Weiler and McDonnell, 2007; Beven, 2010; Beven and Germann, 2013; Klaus et al., 2013; Angermann et al., 2017; Loritz et al., 2017). Such preferential flow can, with increasing wetness, increasingly bypass water volumes stored in small pores with little exchange (Sprenger et al., 2016, 2018, 2019a; Cain et al., 2019; Evaristo et al., 2019; Knighton et al., 2019). This then leads to an increasing preferential release of younger water as the system becomes wetter (Brooks et al., 2010). To mimic this, the shape parameters $\alpha$ of the preferential fluxes $R_{F,H}$ and $R_{F,R}$ released from the two unsaturated root-zone storage components $S_j$

$= S_{U,H}$ and $S_{U,R}$ (Figure 4), were allowed to vary as a function of the water volumes stored in $S_{U,H}$ and $S_{U,R}$, respectively
(Hrachowitz et al., 2013; van der Velde et al., 2015):

$$\alpha_{m,j}(t) = 1 - \left( \frac{S_j(t)}{S_{U,max,j}} (1 - \alpha_0) \right)$$

(Eq.40)

Where $\alpha_0$ is a calibration parameter representing a lower bound so that $\alpha_{m,j}(t)$ can vary between $\alpha_0$ and 1. A value of $\alpha_{m,j} = 1$
indicates complete mixing in dry conditions. Any value below that entails incomplete mixing and thus increases the preference
towards releasing younger water in wet conditions (Benettin et al., 2017). Although there is evidence for the presence of
preferential flow in other components of the system, such as in the groundwater (e.g. Berkowitz and Zehe, 2020), initial model
testing suggested that the inclusion of the additional calibration parameters is not warranted by the available data. For simplicity
and following the principle of model parsimony we assumed complete mixing for all other outflows from all other storage
components (Figure 4; cf. Fenicia et al., 2010; Kuppel et al., 2018a; Rodriguez et al., 2018). Parameter $\alpha$ was therefore fixed
to value of 1 for these SAS functions.

The $\delta^{18}O$ precipitation input signals are damped to the level of fluctuation observed in the stream by subsurface storage volumes
that remain to some extent hydrologically passive (e.g. Birkel et al., 2011b). While the hydrologically active storage volumes
are represented by the individual storage components of the model (Figure 4; Equations 8-14), an additional  hydrologically
passive storage volume $S_{S,p}$ [mm] was added as a calibration parameter to the active groundwater storage $S_{S,a}$. (Zuber, 1986;
Hrachowitz et al., 2015, 2016), so that $S_{S,tot} = S_{S,a} + S_{S,p}$ (Figure 4). While $dS_{S,p}/dt = 0$, the age-ranked  groundwater storage
was computed as $S_{T,Ss,tot}$ and the outflows from the groundwater component consequently thus sampled from the entire storage
volume $S_{S,tot}$, thereby representing the combined contributions from $S_{S,a}$ and $S_{S,p}$ to the age structure of the outflow $Q_S$ according
to Eq. 39. Note that the effects of the hydrologically passive water volume stored in the unsaturated soil below the wilting
point are assumed to be negligible due to the small size of that storage volume and the low diffusive exchange rates with the
hydrologically active storage volume in the unsaturated zone.

Each individual volume with different age in $I_{T,n,j}(T,t)$ and, as a consequence, also in $S_{T,j}(T,t)$ is also characterized by a different
tracer concentration $C_{I,n,j}(I_{T,n,j}(T,t),t)$ and $C_{S,j}(S_{T,j}(T,t),t)$, respectively. For a conservative tracer such as $\delta^{18}O$ that is not
significantly affected by decay, evapoconcentration, retention or any other biogeochemical transformation (e.g. Bertuzzo et
al., 2013; Benettin et al., 2015b; Hrachowitz et al., 2015) the concentration $C_{O,m,j}(t)$ in any outflow at any time $t$ can then be
obtained from:

$$C_{O,m,j}(t) = \int_0^{S_j} C_{S,j}\big(S_{T,j}(T,t),t\big) \varpi_{o,m,j}\big(S_{T,j}(T,t),t\big) dS_T$$

(Eq.41)

Due to data availability, age tracking was here limited to 4 years in the pre- and 3 years in the post-deforestation period. For
age beyond that it can only be said that water is older than these 4 and 3 years, respectively. The TTDs reported hereafter are
thus truncated at these ages. The model generates TTDs for all fluxes and storage components (Figure 4) for each time step.

As a summary metric, we will here use the fraction of young water $F_{yw}$ as robust descriptor of the left tail of TTDs. Following the definition of Kirchner (2016), $F_{yw}$ is here the fraction of water that is younger than 3 months, which can be extracted directly from any TTD generated by the model. Note, we here only analyse water ages in stream flow as these are the only ones that are directly constrained by available data, while for all other model components, such as transpiration $E_T$, such direct data support was not available, and the resulting age estimates may thus be characterized by considerable additional uncertainty.

### 4.3 Model calibration and post-calibration evaluation

The model was run with a daily time step and has a total of 14 free calibration parameters, which were calibrated for the model to simultaneously reproduce flow and $\delta^{18}O$ dynamics in the stream. The uniform prior parameter distributions (Table 2) were sampled using a Monte Carlo approach with $3*10^6$ realizations. To limit equifinality (Beven, 2006) and to ensure robust posterior parameter distributions for a meaningful process representation (e.g. Kuppel et al., 2018b), an extensive multi-objective calibration strategy was applied. Briefly, this was done using a total of 14 performance metrics that describe the model's skill to reproduce different signatures associated to streamflow ($E_Q$) and $\delta^{18}O$ dynamics ($E_{\delta^{18}O}$) as shown in Table 3. To be accepted as feasible, solutions had to exceed a threshold value of 0.5 for all performance metrics, with the exception of $E_{NS,\delta18O}$ for which a threshold of 0.2 was used. To further constrain the model, we only accepted solutions that could reproduce the dynamics in observed soil moisture as well as the average observed magnitudes of canopy throughfall. To do so we used a simplified limits-of-acceptability approach (e.g. Coxon et al., 2014) with a rectangular step function so that all solutions that fall within the limits of the step function receive a weight of one while all others are assigned a weight of zero and thus rejected (Bouaziz et al., 2021). More specifically, we rejected solutions whose modelled normalized relative soil moisture fell outside the acceptable limits, here defined as ± 0.15 of the observed relative soil moisture, in more than 75% of the time steps in the calibration periods. Similarly, we rejected solutions for which the modelled mean ratio $P_E/P$ in the continuously forested part of the catchment was outside ± 0.15 of the observed mean ratio $P_E/P = 0.71$. This strategy was chosen instead of directly calibrating the time-series of associated model variables $S_U$ (Eq.20) and $P_E$ (Eq.18) to explicitly account for commensurability errors between the point-scale and the scale of the model application (Bouaziz et al., 2021). Subsequently, the 14 metrics of the solutions retained as feasible were combined into two equally weighted classes, describing stream flow (Q) and tracer ($\delta^{18}O$) dynamics, respectively. This then allowed to obtain solutions with balanced overall model performances using the mean Euclidean Distance $D_E$ [-] from the "perfect" model (i.e. $D_E = 1$; Hrachowitz et al., 2014; Hulsman et al., 2020):

$$D_E = 1 - \sqrt{\frac{1}{2}\left(\frac{\sum_{n=1}^{N}\left(1 - E_{Q,n}\right)^2}{N} + \frac{\sum_{m=1}^{M}\left(1 - E_{\delta^{18}O,m}\right)^2}{M}\right)}$$

(Eq.42)

Where $N = 12$ is the number of different performance metrics describing streamflow and $M = 2$ the number of different performance metrics for $\delta^{18}O$. To construct the posterior parameter distributions and the corresponding model uncertainty intervals, the retained parameter sets where then weighted according to a likelihood measure $L = D_E^p$ (cf. Freer et al., 1996), where the exponent $p$ was set to a value of 10 to emphasize models with good overall calibration performance.

In a first step, the model was calibrated for the pre-deforestation period 01/10/2009 – 31/08/2013. Note that due to a lack of regular and weekly $\delta^{18}O$ precipitation data before 01/10/2010, the performance metric $E_{\delta^{18}O}$ describing the $\delta^{18}O$ dynamics was computed from that date onwards only. The feasible parameter sets were then used to test the model without further calibration in the post-deforestation period. In a second step, the model was re-calibrated for the 01/09/2013 – 30/09/2016 post-deforestation period and the changes in the resulting model performance and posterior distributions compared to those

from the pre-deforestation calibration. The estimation of the effects of deforestation on TTDs is based on model parameter sets obtained from calibration in the pre-deforestation and post-deforestation periods, respectively.

## 5 Results

### 5.1 Observed deforestation effects on the hydrological system

Initial analysis of water balance data suggests that the hydro-meteorological conditions as expressed by the aridity index $I_A = \overline{E_P}/\overline{P}$, do not show significant differences between the pre-deforestation ($I_A = 0.50 \pm 0.02$) and the post-deforestation periods ($I_A = 0.51 \pm 0.03$), respectively (Figure 5a). However, and in spite of these comparable climatic conditions, the results show a shift in the partitioning of water fluxes between runoff $Q$ and actual evaporation $E_A$ (note that $E_A = E_I + E_T$). While the fraction of precipitation that was released into the atmosphere as vapour was reduced ($\overline{E_A}/\overline{P}$; Figure 5a), the mean runoff ratio ($C_R = $

$1 - \overline{E_A}/\overline{P}$) increased correspondingly from $C_R = 0.55 \pm 0.04$ to $C_R = 0.68 \pm 0.03$ after deforestation of 21 % of the catchment with p = 0.049 based on a Wilcoxon rank sum test. In absolute terms this entails that, notwithstanding rather stable mean annual precipitation $P = 1269 \pm 24$ mm yr$^{-1}$ and potential evaporation $E_P = 632 \pm 9$ mm yr$^{-1}$ over the entire study period, the annual actual evaporation $E_A$ decreased from $576 \pm 11$ mm yr$^{-1}$ to $401 \pm 6$ mm yr$^{-1}$ whereas annual runoff $Q$ increased by $\sim 25$ % from $694 \pm 47$ mm yr$^{-1}$ to $870 \pm 63$ mm yr$^{-1}$.

In spite of similar climatic conditions, the above is reflected in a significantly higher (p = 0.047) mean annual maximum storage deficit in the pre-deforestation than in the post-deforestation period. In the pre-deforestation period values between $105 \pm 23$ mm for $I_{max} = 0$ mm and $95 \pm 21$ mm for $I_{max} = 4$ mm, respectively, were found (Figure 5b). Whereas in the post-deforestation period the mean storage deficit only reached between $49 \pm 10$ mm and $33 \pm 7$ mm for the same values of $I_{max}$ (Figure 5b). Note that in both periods, $S_{D,j}$ is relatively insensitive to the magnitude of $I_{max}$ (cf. Gerrits et al., 2009). From the

above maximum annual storage deficits $S_{D,j}$, the corresponding catchment-scale vegetation-accessible water storage capacity, assuming vegetation adaptation to dry conditions with 40-year return periods (see Section 4.1), was estimated at values of

$S_{U,max} = 258 \pm 125$ mm for the pre-deforestation ($R^2 = 0.91$, p = 0.04; Figure 5c) and $S_{U,max} = 101 \pm 149$ mm for the post-deforestation period ($R^2 = 0.83$, p = 0.27; not shown).

## 5.2 Modelled deforestation effects on the hydrological system

### 5.2.1 Model calibration for pre-deforestation period

The model parameter sets retained as feasible after calibration in the 2009-2013 pre-deforestation period reproduce the general features of the hydrograph in that period rather well (Figures 2c,d), similar to a previous modelling study (Cornelissen et al., 2014). This is true for both, the timing and magnitudes of high flows, with an associated Nash-Sutcliffe Efficiency $E_{NS,Q}$ = 0.83 for the best performing model in terms of $D_E$ (Figure 6a) but also for low flows ($E_{NS,log(Q)}$ = 0.70), with the exception of some overestimation in summer 2011. The modelled runoff ratio comes with $C_R$ = 0.54 (5/95$^{th}$ IQR: 0.52 – 0.58) very close to the observed runoff ration of $C_R$ = 0.55 ($E_{R,CR}$ = 0.98). In addition, the model could also simultaneously mimic most other observed flow signatures reasonably well (Figure 6a), in particular the flow duration curve ($E_{NS,FDC}$ = 0.79; Figure 6b), the peak distribution ($E_{NS,PD}$ = 0.85; Figure 6d) and the auto correlation function ($E_{NS,AC}$ = 0.98; Figure 6f). The limits-of-acceptability constraints for $P_E/P$ allowed the identification and removal of a few additional parameter sets ($\sim$ 5%) that likely overestimate throughfall $P_E$ (Figure 7a). The soil moisture constraint was more effective as it allowed to reject a considerable additional proportion of solutions (> 90%) that did not sufficiently well match the observed soil moisture dynamics according to the pre-defined limits-of-acceptability (Figure 7c). With the parameter sets eventually retained as feasible the modelled temporal dynamics of relative soil moisture broadly reflect the observed ones (Figure 7e) Similarly, the model captures the substantial attenuation of the precipitation $\delta^{18}O$ variability ($E_{R,RD}$ = 0.98; Figure 6a), while at the same time largely preserving the limited but visible low-frequency temporal fluctuations in the stream $\delta^{18}O$ composition (Figures 3a,b). In comparison to the flow performance metrics the Nash-Sutcliffe Efficiency of the $\delta^{18}O$ composition for the best model is somewhat lower ($E_{NS,\delta18O}$ = 0.37; Figure 6a), which mostly results from the low variability of such a damped signal, where even very small absolute errors (MAE = 0.11 ‰) and a few scattered outliers can lead to very low Nash-Sutcliffe Efficiencies (cf. Hrachowitz et al., 2009).

The posterior distributions (Table 2, Figure 8) show that most model parameters are reasonably well identified. Individually calibrated for their respective landscape class, i.e. hillslope and riparian zone, $S_{U,max,H}$ = 242 mm (5/95$^{th}$ IQR: 213 – 311 mm) and $S_{U,max,R}$ = 213 mm (186 – 280 mm) showed similar optimal values and distributions (Figures 7a,b), reflecting the catchment-wide relatively homogenous forest cover in the pre-deforestation period (Figure 1). Remarkably, these calibrated values also come close to catchment-scale estimates of $S_{U,max}$ = 258 $\pm$ 125 mm that were directly derived from water balance data without any calibration, as described in 5.1 (Figure 5c).

### 5.2.2 Application of pre-deforestation model to post-deforestation period

In a next step, the parameter sets obtained from the above calibration in the pre-deforestation period were used to run the model without further re-calibration in the post-deforestation period. This entails the implicit and clearly wrong assumption that the physical characteristics of the system remained unaffected by deforestation. The consequence of that can be seen in Figures 2c and 2d (red line). While the low flows remain well reproduced, the post-deforestation application of the model substantially and systematically underestimates high flows, partly by 50% or more, such as in November 2013 or August 2014. The inability of the model to reproduce several aspects of post-deforestation high-flow dynamics of the system is also evident in the lower model performance metrics associated with high flows (Figure 6a). Besides the time series of flow ($E_{NS,Q} = 0.65$), notably the model's skill to capture the rising limb density ($E_{NS,PD} = 0.78$), the autocorrelation function ($E_{NS,AC} = 0.58$; Figure 6g) and the runoff ratio ($E_{R,CR} = 0.81$) were negatively affected. In contrast to the pre-deforestation period, the modelled runoff ratio $C_R = 0.55$ ($0.54 – 0.58$) in the post-deforestation period considerably underestimates the observed $C_R = 0.68 \pm 0.03$ (Figure 5a). The problems to describe the high flow periods are accompanied by the model's reduced ability to describe the post-deforestation $\delta^{18}O$ dynamics in stream water ($E_{NS,\delta18O} = 0.11$), although the observed general degree of damping of the $\delta^{18}O$ signal ($E_{R,RD} = 0.98$) remains well reproduced as shown in Figures 3 and 6a. While the low $E_{NS,\delta18O}$ values are partly an effect of the above explained low signal-to-noise ratio of such a damped signal and thus of the chosen performance metric, the model also struggles to adequately reproduce the lower-frequency fluctuations, such as between February and July 2014, when the model indicated rather stable $\delta^{18}O$ values while the observed values show a slight yet clear increasing trend over the same period (Figure 3b). Together with the lower overall model performance metric $D_E$ (Figure 6a), these results illustrate that the pre-deforestation model parameter sets provide an unsuitable characterization of the system characteristics in the post-deforestation period.

### 5.2.3 Recalibrate model for post-deforestation period

To estimate the effect of forest removal on the characteristics of the hydrological system and thus on the model parameters, the model was in a next step recalibrated for the post-deforestation period. This led to a slight improvement of the overall model performance from $D_E = 0.77$ to $0.80$ (Figure 6a). Most notably, it can be observed that the recalibrated model can much better reproduce the increased high flows in that period (Figures 2c,d), as reflected by improvements in the performance metrics associated with high flows (Figure 6a), but most notably $E_{NS,Q} = 0.70$, $E_{NS,FDC} = 0.95$ (Figure 6c) or $E_{NS,AC} = 0.92$ (Figure 6g). Similarly, the limits-of-acceptability constraints ensured a choice of solutions that broadly reflect the observed throughfall ratios $P_E/P$ (Figure 7b) as well as the observed soil moisture dynamics (Figures 7d, f). In addition and perhaps most importantly, the runoff ratio also increased and was with a modelled value of $C_R = 0.62$ ($0.56 – 0.63$) closer to the observed $C_R = 0.68$ ($E_{R,CR} = 0.91$). This further implies that, in contrast to the initial model, the recalibrated model also features expected reductions of evaporative fluxes $E_A$ by about 10%, which can be seen in Figure 2b. Mirroring the improvements in the reproduction of flows, recalibration also allowed the model to better capture the stream water $\delta^{18}O$ dynamics ($E_{NS,\delta18O} = 0.24$; MAE = 0.10 ‰; Figure 6a). While there is little change in the model's ability to mimic the general level of damping of the $\delta^{18}O$ signal ($E_{R,RD} = 0.99$)

and its low-frequency fluctuations, the more pronounced, albeit in absolute terms still small, high-frequency fluctuations, as short-term response to individual storms are better described (Figures 3a,b).

Inspection of the posterior parameter distributions reveals that the catchment-scale $S_{U,max}$ experienced considerable reductions after recalibration. While in the hillslope parts of the catchment, which were less affected by deforestation (~ 10% of the hillslope area; Figure 1) an average decrease by ~ 50 mm to $S_{U,max,H}$ = 212 mm (137 – 270 mm) can be seen (Figure 8a), the completely deforested riparian area exhibits an average decrease by ~ 100 mm to $S_{U,max,R}$ = 93 mm (92 – 190 mm; Figure 8b). As an indicative value, the area-weighted catchment-average $S_{U,max}$ = 199 mm of the best performing parameter set falls into

the plausible range of $S_{U,max}$ = 101 ± 149 mm as described in Section 5.1. While there is little evidence for reductions of $I_{max}$ on the less deforested hillslopes (Figure 8d), a clear decrease of interception capacities by on average ~ 2mm to $I_{max,R}$ = 1.1 mm (0.1-1.3 mm; Figure 8e) can be observed in the fully deforested riparian zone.

### 5.3 Deforestation effects on travel time distributions, SAS-functions and young water fractions

While the volume weighted mean $\delta^{18}O$ compositions of observed precipitation with -7.9 ‰ and stream water with -8.2 ‰ are

comparable, a substantial difference in their fluctuations, with standard deviations of 3.6 ‰ and 0.2 ‰, respectively, is evident (Figures 3a,b). This difference suggests a remarkably elevated degree of damping rarely found elsewhere (e.g. Speed et al., 2010), indicative of the importance of old water contributions to the stream in the study catchment. No significant difference in damping ratios was observed between the pre- and post-deforestation period, which further corroborates the prevalence of old water.

Tracking the $\delta^{18}O$ signals through the model then allowed to estimate travel time distributions (TTD). Note that any results reported hereafter are necessarily conditional on the assumptions made in and the uncertainties arising from the modelling process.

In general and consistent with the observed high degree of damping, it was found that pre-deforestation the system was characterized by rather old water. The range of truncated TTDs of stream water exhibits considerable variability in response

to changing wetness conditions with on average about 27 % of the discharge younger than 3 years (Figure 9b,c). In spite of the low mean $F_{yw}$ ~ 0.12 (Figure 10a), stream water can contain up to 34 % water younger than 3 months (i.e. $F_{yw}$ ~ 0.34) for individual storm events in the wet period, while frequently dropping to < 1 % during elongated summer dry periods (Figures 8c, 9a), similar to what has been reported elsewhere (e.g. Gallart et al., 2020b). It can also be observed that the age composition of stream water (Figure 9c) and the associated $F_{yw}$ (Figure 10a) do considerably vary throughout wet periods. Dry periods are

characterized by considerably less variability and more stable stream water TTDs. This is further corroborated by the significantly higher sensitivity of $F_{yw}$ to changes in stream flow in wet-up and wet periods ($dF_{yw}/dQ_n$ ~ 0.35 and 0.25, respectively) as compared to dry periods ($dF_{yw}/dQ_n$ ~ 0.05; Figure 10c). In spite of the low mean $F_{yw}$ ~ 0.12 (Figure 10a), the above also entails that very fast switches towards higher young water fractions can be observed when the system is wetting up after dry periods as well as for storm events throughout the wet season. In general, the above observations are also encapsulated

in the catchment-overall storage age selection functions ω, that represent the ratio of stream water TTD over the combined

RTD of all model storage elements (Benettin et al., 2015a). While for dry periods under-sampling of young water ages with relatively little variability is evident, it can also be seen that in particular during wet-up and wet periods a considerable, yet highly variable preference for very young water can be seen (Figure 11a), similar to what has been reported previously in other environments (e.g. Benettin et al, 2015a; Remondi et al., 2018).

The overall picture did not change in the post-deforestation period. Similar to the pre-deforestation period, the TTDs can exhibit considerable variability. However, in contrast to the pre-deforestation period and depending on the wetness conditions, considerable shifts towards younger water can be observed for the TTDs (Figure 9d-g). There are little discernible changes in $F_{yw}$ during the dry summer months (Figure 9d). However, storms in wet-up periods, mostly during autumn, led to considerable increases in the fractions of water younger than $10 - 20$ days (Figure 9e). During wet periods clear shifts towards younger

water can be observed throughout the entire spectrum of tracked ages (Figure 9f). During the wet period $\sim 36$ % of the stream water are on average younger than the tracked three years (Figure 9i). The mean $F_{yw}$ only slightly increased to 0.13 (Figure 10b), compared to 0.12 in the pre-deforestation period (Figure 10a), which corroborates earlier results by Stockinger et al. (2019) that suggested only minor fluctuations in mean $F_{yw}$ over multiple moving time windows. For individual winter storm events, $F_{yw}$ slightly increased to up to $\sim 0.37$ (Figures 9j, 10b) compared to $F_{yw}$ of up to $\sim 0.34$ in the pre-deforestation period

(Figures 9c, 10a). Besides the generally higher $F_{yw}$ during wet periods, the $F_{yw}$ became more sensitive to flow during wet conditions, with $dF_{yw}/dQ_n \sim 0.36$ (Figure 10d), similar to what has been previously reported by von Freyberg et al. (2018) and Gallart et al. (2020a). The above described post-deforestation changes are also manifest in the corresponding storage age selection function ω (Figure 11b) for that period. While the degree of under-sampling of young water during dry periods significantly decreased, a substantially higher preference for young water during wet-up and wet periods can be observed than

during the pre-deforestation period, with a clear overall shift towards younger water for all wetness conditions.

## 6 Discussion

### 6.1 Observed deforestation effects on the hydrological system

The observed post-deforestation changes to the hydrological response, in particular the increase of $C_R$ from $\sim 0.55$ to $\sim 0.68$

correspond well with the findings of an earlier study in the Wüstebach, based on a shorter study period (2011 – 2015; Wiekenkamp et al., 2016), which estimated an increase of $C_R$ from $\sim 0.58$ to $\sim 0.66$ during that period using eddy-covariance measurements. The overall pattern found here also broadly reflect the effects of land cover/use change in many different environments (Creed et al., 2014; Jaramillo and Destouni, 2014; Renner et al., 2014, van der Velde et al., 2014; Moran-Tejada et al., 2015; Nijzink et al., 2016; Zhang et al., 2017; Jaramillo et al., 2018). The vast majority of these studies suggest that

forest removal leads to an increase in the runoff ratio $C_R$ at the cost of reduced evaporation $E_A$, although the magnitudes of these changes do substantially vary between individual catchments and studies, which is consistent with our physical understanding of the importance of forest for transpiration in hydrological systems.

Under the assumption that reduction of $E_A$ is largely a direct consequence of forest removal in the Wüstebach, a plausible hypothesis to directly attribute this shift in water partitioning from $E_A$ to $Q$ to a physical process can be formulated as follows: the roots of harvested trees stopped extracting water for transpiration from the subsurface. In addition, the limited turbulent exchange of vapour at depth effectively limits soil evaporation to the first few centimetres of the soil (e.g. Brutsaert, 2014). Thus, the felling of trees led to a situation where under comparable atmospheric water demand $E_P$, water volumes held at depths below that and previously within the reach of active roots became largely unavailable for transpiration and evaporation after deforestation. This implies that the water volumes *accessible* to satisfy atmospheric water demand, i.e. $S_{U,max}$ and $I_{max}$, are drastically reduced. Most notably, the available water balance data suggest that catchment-scale $S_{U,max}$ decreased from pre-deforestation $S_{U,max} = 258 \pm 125$ mm to post-deforestation $S_{U,max} = 101 \pm 149$ mm.

Note, however, that in particular the estimates for the post-deforestation period are characterized by considerable uncertainty and therefore need to be understood as merely indicative as they are inferred from only 3 years of data, and a system that is likely to be far from equilibrium, because the deforested part cannot have adapted yet (e.g. Nijzink et al., 2016; Teuling and Hoek van Dijke, 2020). These considerable uncertainties are also reflected in the surprisingly low post-deforestation $S_{U,max}$. Notwithstanding these limitations, the above results illustrate that here the reduction of transpiration due to deforestation is likely a direct consequence of the considerable reduction of $S_{U,max}$ and thus the catchment-scale sub-surface pore volume between field capacity and permanent wilting point that can be actively accessed by vegetation to satisfy the evaporative demand. These post-deforestation decreases in transpiration due to reductions in accessible water volumes $S_{U,max}$, further lead to reduced soil water storage deficits $S_{D,j}$ (Eq.2) in dry seasons, which is consistent with observed post-deforestation increases in soil moisture (Wiekenkamp et al., 2016).

## 6.2 Modelled deforestation effects on the hydrological system

The model application provided further evidence for the central role of $S_{U,max}$ as dominant control on the hydrological response as well as for the direct effects of deforestation on $S_{U,max}$. The model calibration in pre-deforestation period resulted in a set of solutions that could simultaneously reproduce multiple signatures, as expressed by 14 individual performance metrics, while also satisfying two additional limits-of-acceptability constraints. Overall this suggests a rather robust representation of the system.

In a next step, the parameter sets obtained from the calibration in the pre-deforestation period were used to run the model without further re-calibration in the post-deforestation period. This entails the implicit and clearly wrong assumption that the physical characteristics of the system remained unaffected by deforestation. As a consequence, that model exhibited a considerably reduced ability to reproduce the hydrological response in the post-deforestation period, in particular high flows as well as the runoff ratio $C_R$. The latter implies that the model also overestimates post-deforestation evaporative fluxes $E_A$. Therefore, it can, without re-calibration, not deal with the observed changes in the partitioning between drainage and evaporative fluxes (Figure 5a). A likely explanation for the pattern produced by the model is that, in contrast to the real world, no reduction in $E_A$ due to the reduced forest cover is achieved because the model still relies on the catchment-scale vegetation-

accessible storage volume $S_{U,max}$ that characterizes the extent of the catchment-scale active root-system before deforestation. This $S_{U,max}$ *falsely* provides sufficient water supply to sustain $E_A$ at high levels comparatively close to $E_P$ throughout the year (see red line in Figure 2b), although, in the parts of the catchment where trees were removed, water stored at depths below a few centimetres is not available for significant evaporation anymore in reality. Such an overestimation of $S_{U,max}$ implies also

that in the model a more pronounced water storage deficit can and does develop throughout dry periods. The model therefore assumes that soils dry out to deeper depths. Consequently, to establish connectivity and to eventually generate flow during and after rainstorms, more water needs to be stored in the model than in the real world system to overcome this deficit. This water is then in the model held against gravity and thus only available for evaporation but *not* for drainage, thereby underestimating in particular the magnitude of high flows. Although it is reasonable to assume that groundwater recharge is affected in a similar

way, the model can better reproduce low flows. The reason for this is that the draining groundwater body, which sustains summer low flows, is, due to limited recharge during these drier periods, largely disconnected from and thus largely unaffected by subsurface – vegetation interaction in shallower parts of the subsurface. In the parts of the catchment where trees were removed a similar reasoning also holds for the interception capacity $I_{max}$ and the associated likely overestimation of interception evaporation $E_I$, yet, due to the smaller magnitude of $I_{max}$, to a lesser extent than for $S_{U,max}$.

Recalibration of the model in the post-deforestation period led to a considerably improved representation of the hydrological response and in particular of the high-flows as well as the runoff ratio $C_R$. The latter implies that the modelled partitioning of water fluxes and in particular $E_A$ (see orange line in Figure 2b) is more consistent with the observed post-deforestation reductions in $E_A$. In addition, analysis of the modelled fluxes indicates that a higher proportion of flows, mostly during wet-up periods, is rapidly released from the root-zones as fluxes $R_{F,H}$ and $R_{F,R}$ (Figure 4; Table 1), representing preferential flows.

Such a post-deforestation increase in preferential flow occurrence is supported by observations recently reported by Wiekenkamp et al. (2020).

It is of course unsurprising that recalibration leads to an improved model performance in the post-deforestation period. Without further analysis, such a mere model fitting exercise allows in the presence of model equifinality only little insight into the underlying processes (Beven, 2006; Kirchner, 2006). To gain more confidence that the improvements in the recalibrated model

are at least partly due to the right reasons (Kirchner, 2006), the changes in the posterior parameter distributions resulting from the two calibration runs were thus analysed. In the pre-deforestation period, the range of the posterior distributions of $S_{U,max,H}$ and $S_{U,max,R}$ (Figure 8a,b) as well as the modelled catchment-average $S_{U,max} = 240$ mm, estimated as area-weighted average of $S_{U,max,H}$ and $S_{U,max,R}$, come close to the catchment-scale estimate of $S_{U,max} = 258 \pm 125$ mm that was directly derived from water balance data without any calibration (Figure 5c). The modelled post-deforestation reductions of $S_{U,max,H}$ and $S_{U,max,R}$ are evident

in the shifts of their respective posterior distributions (Figure 8a,b) and the lower catchment-average $S_{U,max} = 199$ mm of the best performing parameter set, falling into the plausible range of $S_{U,max} = 101 \pm 149$ mm as estimated from water balance data. In addition and quite remarkably, the re-calibrated model is able to broadly represent the differences in forest removal on the hillslopes and in the riparian zone. While in the fully deforested riparian area $S_{U,max,R}$ decreased by $\sim 100$ mm (Figure 8b), $S_{U,max,H}$ on the only partly deforested hillslopes decreased by merely $\sim 50$mm (Figure 8a). Similarly, there is little evidence for

reductions of $I_{max,H}$ on the less deforested hillslopes (Figure 8d). Yet, a clear decrease of interception capacities by on average $\sim$ 2mm to $I_{max,R}$ = 1.1 mm (0.1 – 1.3 mm; Figure 8e) can be observed in the riparian zone. Comparing to the posterior distributions of other parameters, the results illustrate that the storage parameters $S_{U,max}$ and $I_{max}$ of the completely deforested riparian zone, and to a lesser extent of the hillslope, were subject to the most pronounced changes. For most other parameters, the pre- and post-deforestation posterior distributions exhibit much less pronounced differences (Figure 8). Together, these results suggest that deforestation mostly affects $S_{U,max}$ and $I_{max}$, while there is less evidence for systematic changes in other parameters. However, it can also be observed that the individual parameter values associated with the best model solutions in the pre- and post-deforestation periods, respectively, do vary to a stronger degree for most parameters. Notwithstanding the distinct overall effects of forest removal on the individual posterior distributions, this clearly highlights the influence of parameter compensation effects and related uncertainties. This is also illustrated by a few parameters, such as $R_{S,max}$ (Figure 8c, Equation 22), that remain poorly constrained.

It was hypothesized above that reductions in evaporative fluxes are directly and exclusively linked to reduced water volumes $S_{U,max}$ and $I_{max}$, respectively, which are accessible and available for evaporation and transpiration at the catchment-scale. In the theoretical ideal case, the representations of the associated storage capacities in the model, i.e. the parameters $S_{U,max}$ and $I_{max}$, should thus be the only ones to significantly change after deforestation. However, note that this is unlikely for two reasons. First, while it is plausible to assume that these storage capacities are significantly affected by forest removal, it is not unlikely that other system characteristics and their mutual interactions, so far unknown and not considered, are similarly influenced, potentially causing considerable ontological uncertainty. Second, model parameter interactions that arise as artefacts to compensate overly simplistic process representations and/or data uncertainty are also likely to affect parameters seemingly unrelated to deforestation. Note that in spite of these uncertainties and the associated compensation effects, in particular $S_{U,max}$ remains rather well constrained. However, after preliminary unsuccessful testing, no further attempts were made to re-calibrate only the above discussed four storage parameters, i.e. $S_{U,max,H}$, $S_{U,max,R}$, $I_{max,H}$ and $I_{max,R}$, acknowledging the limitations introduced by parameter compensation effects.

Overall these results suggest that the model formulation together with the multi-objective calibration strategy ensured the identification of solutions that provide a robust description of the system and allow a simultaneous representation of flow and isotope dynamics in the stream. There are indications that at least some processes and parameters can be directly linked to real world quantities. In particular, the results provide strong evidence that the parameters $S_{U,max,H}$ and $S_{U,max,R}$ are not merely abstract quantities, but that it is plausible to assume that they, taken together, provide a catchment-scale representation of vegetation-accessible and -accessed water volumes, which can be estimated based on water balance data without calibration as defined by Equation 2, thereby providing an alternative to small-scale in-situ observations. As such, the parameter $S_{U,max}$ is also a means to directly and independently estimate the catchment-scale effects of deforestation, and plausibly other types of land cover disturbances, on sub-surface system properties, which underlie and control the changes in the post-disturbance partitioning of water fluxes into drainage and evaporative fluxes.

### 6.3 Deforestation effects on travel time distributions, SAS-functions and young water fractions

Tracking water fluxes through the system it was observed that wet periods are characterized by substantially more variability and more stable stream water TTDs than dry periods. This is largely a consequence of increased bypass flow that has little interaction with resident water as the system gets wetter and which may reach the stream over preferential flow paths and increased contributions from the riparian zone with its shorter flow paths. In other words, in a wet system where little additional water can be stored, the precipitation volumes of individual storm events control the shape of TTDs, resulting in considerable variability (Heidbüchel et al, 2020). In the summer dry season, however, precipitation is to a higher degree buffered in the root-zone and used for transpiration (Stockinger et al., 2014). Conversely, stream flow is then mostly sustained by groundwater which is characterized by large volumes of older water. This effectively attenuates fluctuations by the proportionally much lower volumes of younger precipitation water that cannot be stored and is thus quickly released to the stream.

In particular, at the beginning of the wet period, elsewhere also referred to as "autumn flush" (e.g. Dawson et al., 2011), the switches towards younger water at given flow levels occur considerably faster in the post-deforestation period than in the pre-deforestation period. Therefore, where, at the same discharge, previously relatively little young water reached the stream, a much higher fraction of young water can now be observed in the stream. Underlining the role of transpiration (e.g. Douinot et al., 2019; Kuppel et al., 2020), this is a direct effect of the reduced evaporative removal of relatively young near-surface water (Maxwell et al., 2019) in the post-deforestation period, which in turn is intimately linked to the reduced water supply for evaporative fluxes, i.e. smaller storage volumes $S_{U,max}$ and $I_{max}$. This modelled relatively young, surface-near water, not taken up by vegetation anymore is thus to a higher degree flushed from the system mostly via preferential flow paths to the stream (i.e. $R_{F,H}$, $R_{F,R}$) and thus bypassing older resident water with little exchange, which is consistent with recent observations of more frequent activation of preferential flow paths (Wiekenkamp et al., 2020). Once connectivity and the associated higher degree of bypass flow are established in the wet period, the post-deforestation peak sensitivity of $dF_{yw}/dQ$ to flow increased to ~ 0.36, as under these conditions when little additional water can be stored in the shallow subsurface, $F_{yw}$ is largely controlled by magnitude of the individual precipitation signals and to a lesser extent by the footprint of the pre-storm history of evaporative fluxes in the shallow subsurface storage. In contrast, no significant post-deforestation changes could be observed for the sensitivity of $F_{yw}$ to discharge during dry periods, as during that period, the composition of water ages is controlled by large volumes of old water.

Altogether these results suggest that even in systems dominated by old water, such as the Wüstebach, the removal of forest has the potential to increase the importance of bypass flow through fast flow paths and thus increase the risk of fast, often underestimated propagation of contaminant pulses into ground- and stream water (e.g. Hartmann et al., 2021).

### 6.2 Uncertainties, unresolved questions and limitations

As emphasized above, all results are conditional on the assumptions taken throughout the modelling process. These assumptions, present in model structure, parameterization and parameters, can lead to uncertainties. Yet, notwithstanding these

potential uncertainties, extensive preliminary model testing together with the use of multiple model calibration and evaluation criteria suggest that there is relatively strong evidence to support the main results in this study: the post-deforestation reduction of evaporative fluxes can, at least partially, be linked to a relatively clear reduction in the catchment-scale storage capacities $S_{U,max}$ and $I_{max}$, which in turn triggered a shift towards younger water ages in the stream, particularly during wet-up and wet conditions.

This is further corroborated when comparing the estimates of $S_{U,max}$ to estimates of physically plausible upper limits of $S_{U,max}$. By definition, $S_{U,max}$ is physically bound by the depth of the groundwater table. Although fluctuating, the groundwater table in the Wüstebach remains at depths below 1 m for much of the year even in the riparian zone (Bogena et al., 2015) and can be expected to be considerably deeper on the hillslopes. Thus assuming a conservative upper bound of catchment-average depth of the groundwater table at $\sim 5$ m, assuming that the lowest groundwater table at each point in the catchment is at the elevation of the nearest stream, a porosity of the silty clay loam soil of 0.4 (Bogena et al., 2018) and field capacity at a relative pore water content of 0.5 suggests an upper limit of $S_{U,max,GW} \sim 1000$ mm. However, actual roots are very often shallower than these 5 m of the groundwater table. Although sufficient detailed data on root depths are not available in the study catchment, there is no evidence for systematic and wide-spread roots extending to below 2 m. This is broadly consistent with direct experimental evidence that roots of temperate forests in general (Schenk and Jackson, 2002) and *Picea* species in particular mostly remain rather shallow (< 1 m; e.g. Schmid and Kazda, 2001) and with indirect evidence that *Picea* species rarely tap groundwater and are thus comparatively shallow (e.g. Evaristo and McDonnell, 2017). As a conservative back-of-the-envelope calculation, assuming thus a maximum plausible catchment-average root depth of 2 m, which comes close to the average observed soil depth reported in Graf et al. (2014), rather suggests a physically plausible upper limit of $S_{U,max,RD} \sim 400$ mm, which is not exceeded by the water balance inferred catchment-scale estimates of $S_{U,max} = 258 \pm 125$ mm.

Note that the above also suggests the presence of an unsaturated transition zone between the root-zone and the groundwater table, i.e. $S_{U,max,TZ} = S_{U,max,GW} - S_{U,max,RD} \sim 600$ mm. In the absence of root water uptake and likely negligible soil evaporation in that zone the water content will remain close to field capacity for much of the year, except for days when a wetting front infiltrates towards the groundwater. This transition zone can therefore be considered as hydrologically largely passive so that at time scales of more than a few days $dS/dt \sim 0$. However, this zone also provides a mixing volume that affects tracer circulation and thus water ages (Hrachowitz et al., 2015). Given its hydrologically passive nature and following the idea of a parsimonious model to limit uncertainty, we here, in a simplification, implicitly added the mixing volume $S_{U,max,TZ}$ to the passive groundwater mixing volume $S_{S,p}$.

For a meaningful interpretation, two specific observations resulting from our analysis warrant special scrutiny. First, model calibration-based estimations of hillslope $S_{U,max,H}$ (Figures 7a) suggest post-deforestation median $S_{U,max,H}$ reductions of $\sim 25$ % as a consequence of clear cutting only ~10 % of the hillslope part of the catchment (Figure 1). While this may be surprising at the first, it can be plausibly explained by considerable further thinning of the remaining forest on the hillslopes in 2015, two years after deforestation and thus by reduced catchment-scale transpiration demand. Yet, no detailed and systematic data on the degree of forest thinning is available to meaningfully test this hypothesis.

Second, our results suggest that a passive mixing volume $S_{S,p}$ of at least $\sim 8.000$ mm is necessary for the model to attenuate the amplitudes of the precipitation $\delta^{18}O$ signals to those in the stream water. Although, $S_{S,p}$ is rather well constrained (Figure 8h), there has in the past been no hydrogeological evidence for the presence of such a surprisingly large groundwater volume nor for its hydrological relevance in the study catchment. Indeed, the authors are not aware of any catchment-scale study that reported similarly high values for $S_{S,p}$ or functionally equivalent parameters (e.g. Birkel et al., 2011a,b; Hrachowitz et al., 2013,2015; Benettin et al., 2013,2015a; Harman, 2015; van der Velde et al., 2015). Yet, to achieve the degree of damping observed in the stream water, such a volume is necessary, if the current understanding of conservative tracer dynamics holds e.g. Maloszewski and Zuber, 1982; McGuire and McDonnell, 2006). Reflecting our insufficient knowledge to which depths exchange with surface water occurs (e.g. Condon et al., 2020), a potential explanation for this observation is that the frequently layered and fractured structure of the Devonian shale bedrock may provide relatively high-permeability pathways for the circulation of and exchange with water at depth. Another, yet, given the current understanding of the Wüstebach (e.g. Graf et al., 2014), less likely hypothesis is the presence of significant lateral groundwater exchange (e.g. Bouaziz et al., 2018; Hulsman et al., 2021b). In other words the possibility that the subsurface catchment does not match with the surface catchment (Figure 1) and that older groundwater is imported from "outside" the surface catchment, while an equivalent volume of younger groundwater is exported, maintaining the mass balance. These are hypotheses to be tested in future studies, as the currently available data do not allow a conclusive answer to this question.

**7 Conclusions**

The small Wüstebach catchment experienced significant deforestation in 2013. Analyzing the effects of this deforestation on the hydrology and stable isotope circulation dynamics in the study catchment our main findings are:

(1) Water balance data suggest that deforestation led to a significant increase of stream flow, accompanied by corresponding reductions of evaporative fluxes. This is reflected by an increase of the runoff ratio from $C_R = 0.55$ to $0.68$ in the post-deforestation period despite similar climatic conditions, supporting previous results based on eddy covariance measurements (Wiekenkamp et al., 2016).

(2) Based on water balance data, this reduction of evaporative fluxes, as a consequence of reduced vegetation water uptake, could at least partly be linked to a reduction of the catchment-scale water storage volume in the unsaturated soil ($S_{U,max}$) that is within the reach of active roots and thus accessible for vegetation transpiration from $\sim$258 mm in the pre-deforestation period to $\sim 101$ mm in the post-deforestations period.

(3) Estimating $S_{U,max}$ as calibration parameter of a process-based hydrological model led to similar conclusions. The catchment-average calibrated model parameters representing $S_{U,max}$ for both, the pre- and deforestation periods, respectively, correspond with $\sim 240$ mm and $\sim 199$ mm broadly with $S_{U,max}$ directly estimated from water balance data. Other model parameters, assumed to have a less direct link to vegetation, exhibited much lower levels of systematic change following deforestation.

(4) Using the model to track the age composition of stream water suggested that, in general, water reaching the stream in the pre-deforestation period was rather old with a mean young water fraction $F_{yw} \sim 0.12$. In spite of the overall low $F_{yw}$, clear shifts in the shape of travel time distributions towards younger water can be seen under wet conditions with young water fractions increasing up to $F_{yw} \sim 0.34$.

(5) Deforestation and the associated reduction of $S_{U,max}$ led to shifts in travel time distributions towards younger water. Under wet conditions, this resulted in increases of young water fractions to up to $F_{yw} \sim 0.37$ for individual storms. In contrast, dry period travel time distributions exhibited only minor changes. Overall the mean fraction of young water in the stream increased to $F_{yw} \sim 0.13$.

(6) Deforestation resulted in a considerable increase of the sensitivity of young water fractions to discharge under wet
conditions from $dF_{yw}/dQ = 0.25$ to $0.36$. This implies faster switches towards younger water and thus faster routing of solutes during and shortly after storm events and thus faster routing of solutes with increasing wetness.

The above results suggest that deforestation has not only the potential to affect the partitioning between drainage and evaporation, and thus the fundamental hydrological response characteristics of catchments, but also catchment-scale tracer
circulation dynamics. In particular for wet and wet-up conditions, sometimes also referred to as "autumn flush", deforestation in the Wüstebach caused higher proportions of younger water to reach the stream, implying faster routing of water and plausibly also solutes through the subsurface, thereby also increasing the risk for faster propagation of contaminants into stream- and groundwater.

Overall, this study demonstrates that post-deforestation changes in both, the hydrological response and travel times, can to a
large extent be traced back and attributed to changes in $S_{U,max}$, a readily quantifiable catchment-scale subsurface property (and model parameter) representing the maximum water volume that can be stored within the reach of roots. As such, $S_{U,max}$ and changes therein provide a quantitative, mechanistic hypothesis that can explain *why* deforestation in the Wüstebach decreased evaporative fluxes, increased stream flow – particularly generated by preferential flows – and reduced travel times. The catchment-scale quantification of $S_{U,max}$ based on water balance data therefore provides a potentially valuable way towards
meaningful and data-based catchment-scale representation of vegetation-accessible water where soil and root observations are not available at sufficient spatial and temporal detail to meaningfully represent their respective natural heterogeneities. In addition and perhaps more importantly, the method may also hold considerable potential for the formulation of temporally adaptive root-zone parameterizations in catchment-scale hydrological models for more reliable predictions in a changing environment.


*Data availability.* The meteorological and hydrological data of the Wüstebach TERENO site used in this study can be made available by the co-author HB upon request. The model results, including states, fluxes, hydrological signatures, parameter sets, and performance metrics underlying this paper are available online in the 4TU data repository at https://doi.org/10.4121/14626050.v1.


*Code availability.* The model code used can be made available by the first author upon request. The equations used in the model are described in the paper.

*Author contributions.* MH and MS designed the experiment. MH did the analysis and wrote the first draft. All authors discussed
the design, results and the first draft and contributed to writing the final manuscript.

*Competing interests.* The authors declare that they have no conflict of interest.

*Acknowledgments.* We would like to thank the editor and four anonymous reviewers for providing a list of critical and very
valuable comments that helped to considerably improve the manuscript.

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

**Table 1: Water balance, state and flux equations used in the hydrological model. Symbols shown in bold are model parameters. Subscripts H and R indicate hillslope and riparian zone, respectively. Model variables: $P$ is total precipitation [mm d$^{-1}$], $P_S$ is solid precipitation (snow) ]mm d$^{-1}$], $P_M$ is snow melt [mm d$^{-1}$], $P_R$ is rain [mm d$^{-1}$], $P_E$ is throughfall [mm d$^{-1}$], $E_P$ is potential evaporation [mm d$^{-1}$], $E_I$ is interception evaporation [mm d$^{-1}$], $R_F$ is preferential recharge [mm d$^{-1}$], $R_S$ is slow recharge [mm d$^{-1}$], $E_T$ is transpiration [mm d$^{-1}$], $Q_S$ is flow from slow responding reservoir [mm d$^{-1}$], $Q_R$ is flow from the fast responding riparian reservoir [mm d$^{-1}$], $Q$ is the total flow [mm d$^{-1}$] and $E_A$ is the total actual evaporation [mm d$^{-1}$]. Model parameters: $T_T$ is the threshold temperature [ºC], $F_M$ is a melt factor [mm d$^{-1}$ ºC$^{-1}$], $I_{max}$ is the interception capacity [mm], $S_{U,max}$ is the root-zone storage capacity [mm], $\gamma$ is a shape factor [-], $R_{S,max}$ is the maximum percolation rate [mm d$^{-1}$], $L_p$ is a transpiration water stress factor [-], $f_{QS}$ is a factor determining the fraction of groundwater flow that is upwelling into the riparian zone [-], $k_S$ is the storage coefficient of the slow responding reservoir [d$^{-1}$], $k_R$ is the storage coefficient for the fast responding riparian reservoir [d$^{-1}$] and $f$ is the areal fraction of the riparian zone [-].**

| Landscape unit | Storage component | Water balance | Eq. | Constitutive equations | Eq. |
|---|---|---|---|---|---|
| Hillslope | Snow storage | $dS_{snow}/dt = P_S - P_M$ | (8) | $P_S = \begin{cases} P, & T < \mathbf{T_T} \\ 0, & T \geq \mathbf{T_T} \end{cases}$ | (15) |
| | | | | $P_M = \begin{cases} 0, & T < \mathbf{T_T} \\ min\left(\mathbf{F_M}(T - \mathbf{T_T}), \frac{S_{snow}}{dt}\right), & T \geq \mathbf{T_T} \end{cases}$ | (16) |
| | Interception storage | $dS_{I,H}/dt = P_R + P_M - P_{E,H} - E_{I,H}$ | (9) | $P_R = \begin{cases} 0, & T < \mathbf{T_T} \\ P, & T \geq \mathbf{T_T} \end{cases}$ | (17) |
| | | | | $P_{E,H} = max\left(0, \frac{S_{I,H} - \mathbf{I_{max,H}}}{dt}\right)$ | (18) |
| | | | | $E_{I,H} = min\left(E_P, \frac{S_{I,H} - P_{E,H}}{dt}\right)$ | (19) |
| | Unsaturated root-zone storage | $dS_{U,H}/dt = P_{E,H} - R_{F,H} - R_{S,H} - E_{T,H}$ | (10) | $S'_{U,H} = (1 + \mathbf{\gamma})\mathbf{S_{U,max,H}}\left(1 - \left(1 - \frac{S_{U,H}}{\mathbf{S_{U,max,H}}}\right)^{\left(\frac{1}{1+\gamma}\right)}\right)$ | (20) |
| | | | | $R_{F,H} = P_{E,H} - \left(\mathbf{S_{U,max,H}} + S_{U,H} + \mathbf{S_{U,max,H}}\left(1 - \frac{P_{E,H}dt + S_{U,H}'}{(1 + \mathbf{\gamma})\mathbf{S_{U,max,H}}}\right)^{(1+\gamma)}\right)dt^{-1}$ | (21) |
| | | | | $R_{S,H} = min\left(\mathbf{R_{S,max}}\frac{S_{U,H}}{\mathbf{S_{U,max,H}}}, \frac{S_{U,H}}{dt}\right)$ | (22) |
| | | | | $E_{T,H} = min\left((E_P - E_{I,H})min\left(\frac{S_{U,H}}{\mathbf{S_{U,max,H}}}\frac{1}{\mathbf{L_P}}, 1\right), \frac{S_{U,H}}{dt}\right)$ | (23) |
| | Slow responding storage | $dS_{S,a}/dt = (1 - \mathbf{f})(R_{F,H} + R_{S,H}) - R_{S,R} - Q_S$ | (11) | $R_{S,R} = \mathbf{f_{QS}}S_{S,a}(1 - e^{-k_S t})dt^{-1}$ | (24) |
| | | | | $Q_S = (1 - \mathbf{f_{QS}})S_{S,a}(1 - e^{-k_S t})dt^{-1}$ | (25) |
| Riparian zone | Interception storage | $dS_{I,R}/dt = P_R + P_M - P_{E,R} - E_{I,R}$ | (12) | $P_{E,R} = max\left(0, \frac{S_{I,R} - \mathbf{I_{max,R}}}{dt}\right)$ | (26) |
| | | | | $E_{I,R} = min\left(E_P, \frac{S_{I,R} - P_{E,R}}{dt}\right)$ | (27) |
| | Unsaturated root-zone storage | $dS_{U,R}/dt = P_{E,R} + R_{S,R}/\mathbf{f} - R_{F,R} - E_{T,R}$ | (13) | $S'_{U,R} = (1 + \mathbf{\gamma})\mathbf{S_{U,max,R}}\left(1 - \left(1 - \frac{S_{U,R}}{\mathbf{S_{U,max,R}}}\right)^{\left(\frac{1}{1+\gamma}\right)}\right)$ | (28) |
| | | | | $R_{F,R} = P_{E,R} + \frac{R_{S,R}}{f} - \left(\mathbf{S_{U,max,R}} + S_{U,R} + \mathbf{S_{U,max,R}}\left(1 - \frac{P_{E,R}dt + S_{U,R}'}{(1 + \mathbf{\gamma})\mathbf{S_{U,max,R}}}\right)^{(1+\gamma)}\right)dt^{-1}$ | (29) |
| | | | | $E_{T,R} = min\left((E_P - E_{I,R})min\left(\frac{S_{U,R}}{\mathbf{S_{U,max,R}}}\frac{1}{\mathbf{L_P}}, 1\right), \frac{S_{U,R}}{dt}\right)$ | (30) |
| | Fast responding storage | $dS_{F,R}/dt = R_{F,R} - Q_R$ | (14) | $Q_R = S_{F,R}(1 - e^{-k_R t})dt^{-1}$ | (31) |
| | | | | $Q = Q_S + \mathbf{f}Q_R$ | (32) |
| | | | | $E_I = (1 - \mathbf{f})E_{I,H} + \mathbf{f}E_{I,R}$ | (33) |
| | | | | $E_T = (1 - \mathbf{f})E_{T,H} + \mathbf{f}E_{T,R}$ | (34) |
| | | | | $E_A = E_I + E_T$ | (35) |

**Table 2: Parameter prior distributions and 5/95th percentiles of the posterior distributions. Note that *) parameter *f*, characterizing the areal proportion of the riparian zone was fixed according to soil and elevation data and **) the interception capacity *I*max was assumed to be identical on the hillslopes and the riparian zone in the pre-deforestation period.**

| Model | Parameter | Prior distribution | Posterior distribution | |
| --- | --- | --- | --- | --- |
| | | | Pre-deforestation | Post-deforestation |
| Hydrological model | $f$ [-]* | 0.1 | 0.1 | 0.1 |
| | $F_M$ [mm d$^{-1}$ $^\circ$C$^{-1}$] | 1.0 – 5.0 | 2.0 – 4.8 | 1.4 – 4.7 |
| | $f_{QS}$ [-] | 0.00 – 0.20 | 0.02 – 0.11 | 0.01 – 0.11 |
| | $I_{max,H}$ [mm] | 0.0 – 6.0 | 1.9 – 4.8 | 2.5 – 4.1 |
| | $I_{max,R}$ [mm]** | 0.0 – 6.0 | 1.9 – 4.8 | 0.1 – 1.3 |
| | $k_R$ [d$^{-1}$] | 0.01 – 2.00 | 0.26 – 1.28 | 0.29 – 1.01 |
| | $k_S$ [d$^{-1}$] | 0.01 – 0.20 | 0.02 – 0.15 | 0.03 – 0.17 |
| | $L_p$ [-] | 0.0 – 1.0 | 0.2 – 0.8 | 0.1 – 0.3 |
| | $R_{S,max}$ [mm d$^{-1}$] | 0.0 – 4.0 | 0.5 – 2.8 | 0.9 – 3.1 |
| | $S_{U,max,H}$ [mm] | 0 – 400 | 213 – 311 | 137 – 270 |
| | $S_{U,max,R}$ [mm] | 0 – 400 | 186 – 280 | 92 – 190 |
| | $T_T$ [$^\circ$C] | -1.5 – 1.5 | -0.6 – 1.1 | -0.2 – 1.1 |
| | $\gamma$ [-] | 0.0 – 5.0 | 0.2 – 1.0 | 0.5 – 4.3 |
| Tracer model | $\alpha_0$ [-] | 0.00 – 1.00 | 0.80 – 0.99 | 0.61 – 0.96 |
| | $S_{S,p}$ [mm] | 1000 – 30 000 | 7999 – 16228 | 7612 - 13920 |



**Table 3: Signatures of flow and $\delta^{18}O$ and the associated performance metrics used for model calibration and evaluation. The performance metrics used include the Nash-Sutcliffe efficiency ($E_{NS}$), the volume error ($E_V$) and the relative error ($E_R$).**


| Variable/Signature | Symbol | Performance Metric | Reference |
|---|---|---|---|
| Time series of flow | Q | $E_{NS,Q}$ | Nash and Sutcliffe (1970) |
| | log(Q) | $E_{NS,log(Q)}$ | |
| | Q | $E_{V,Q}$ | Criss and Winston (2008) |
| Flow duration curve | FDC | $E_{NS,FDC}$ | Jothityangkoon et al. (2001) |
| Flow duration curve high flow period | FDC,h | $E_{NS,FDCh}$ | Yilmaz et al. (2008) |
| Peak distribution | PD | $E_{NS,PD}$ | Euser et al. (2013) |
| Rising limb density | RLD | $E_{R,RLD}$ | Shamir et al. (2005) |
| Declining limb density | DLD | $E_{R,DLD}$ | Sawicz et al. (2011) |
| Autocorrelation function of flow | AC | $E_{NS,AC}$ | Montanari and Toth (2007) |
| Lag-1 autocorrelation | AC1 | $E_{R,AC1}$ | Hrachowitz et al. (2014) |
| Lag-1 autocorrelation low flow period | AC1,l | $E_{R,AC1,l}$ | Fovet et al. (2015) |
| Runoff ratio | CR | $E_{R,CR}$ | Yadav et al. (2007) |
| Time series of $\delta^{18}O$ in stream water | $\delta^{18}O$ | $E_{NS,\delta18O}$ | Birkel et al. (2011a) |
| Damping ratio of $\delta^{18}O^{*)}$ | RD | $E_{R,RD}$ | |

$^{*)}RD = \dfrac{stdev_Q(\delta^{18}O)}{stdev_P(\delta^{18}O)}$



**Figure 1: Map of the Wüstebach study catchment showing the spatial distribution of soil types. The riparian zone is defined by the parts of the catchment covered by Gleysols, Planosols and Halfbogs. The red line indicates the outline of the deforested part of the catchment, as can also be seen on the aerial images (Google Earth, Maxar Technologies 2020) from 2013 and 2016.**

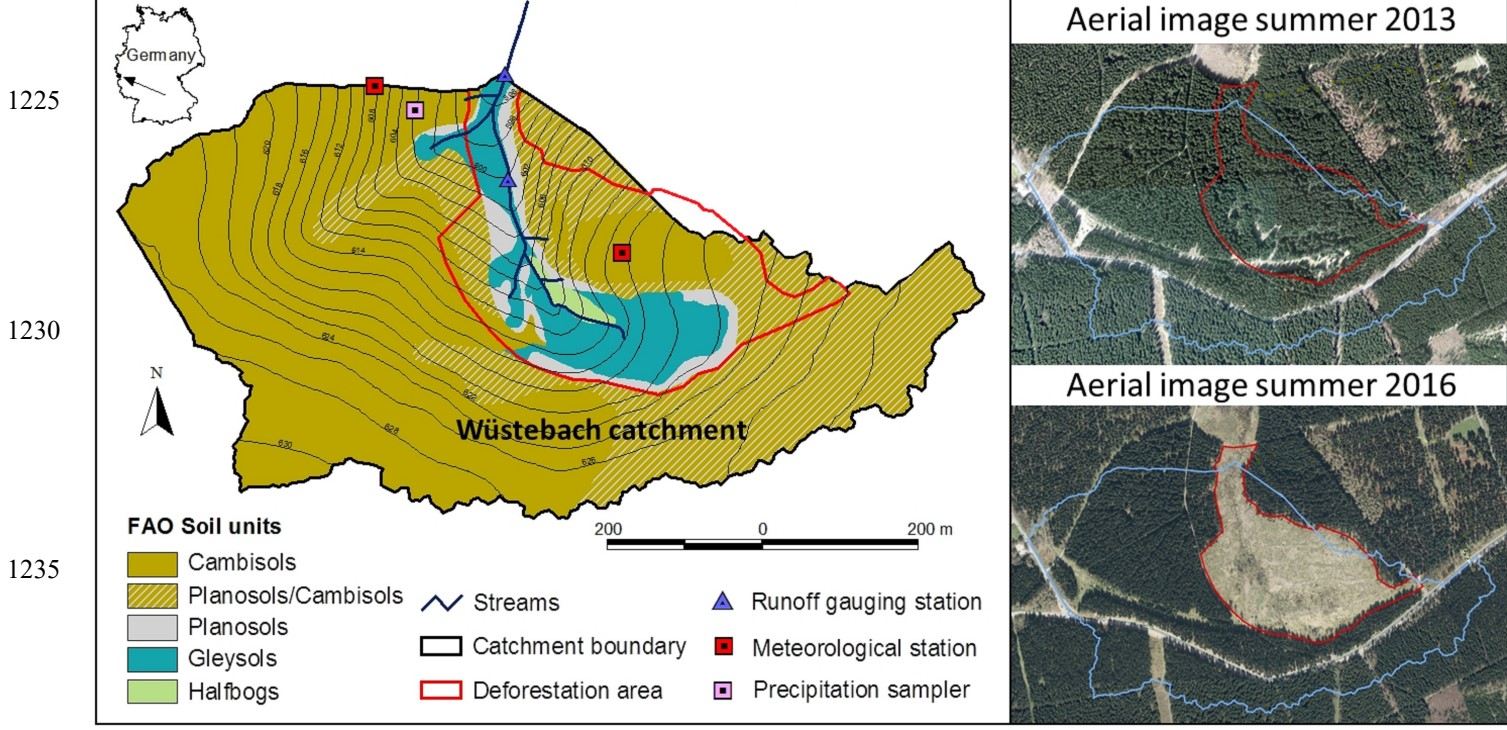

**Figure 2: (a)** Time series of observed weekly precipitation *P*; **(b)** daily cumulative evaporative fluxes for the pre- and post-deforestation period, where the dark brown line indicates potential evaporation $E_P$ and the orange lines and the yellow shaded areas show the actual evaporation $E_A$ modelled using the best fit parameter sets and the associated 5/95th percentiles of all feasible solutions of the pre- and post-deforestation periods, respectively. The dashed red line indicates the modelled $E_A$ in the post-deforestation period using the best fit pre-deforestation parameter set; **(c)** observed (dark blue line) and modelled daily stream flow *Q*; light blue line indicates best fit model and the shaded area the 5/95th percentile of all feasible solutions for the pre- and post-deforestation periods, respectively. The dashed red line indicates the modelled stream in the post-deforestation period using the best fit pre-deforestation parameter set; **(d)** zoom-in to the observed and modelled stream flow for the 10/2012 – 10/2014 period. The grey shaded area indicates the deforestation period.

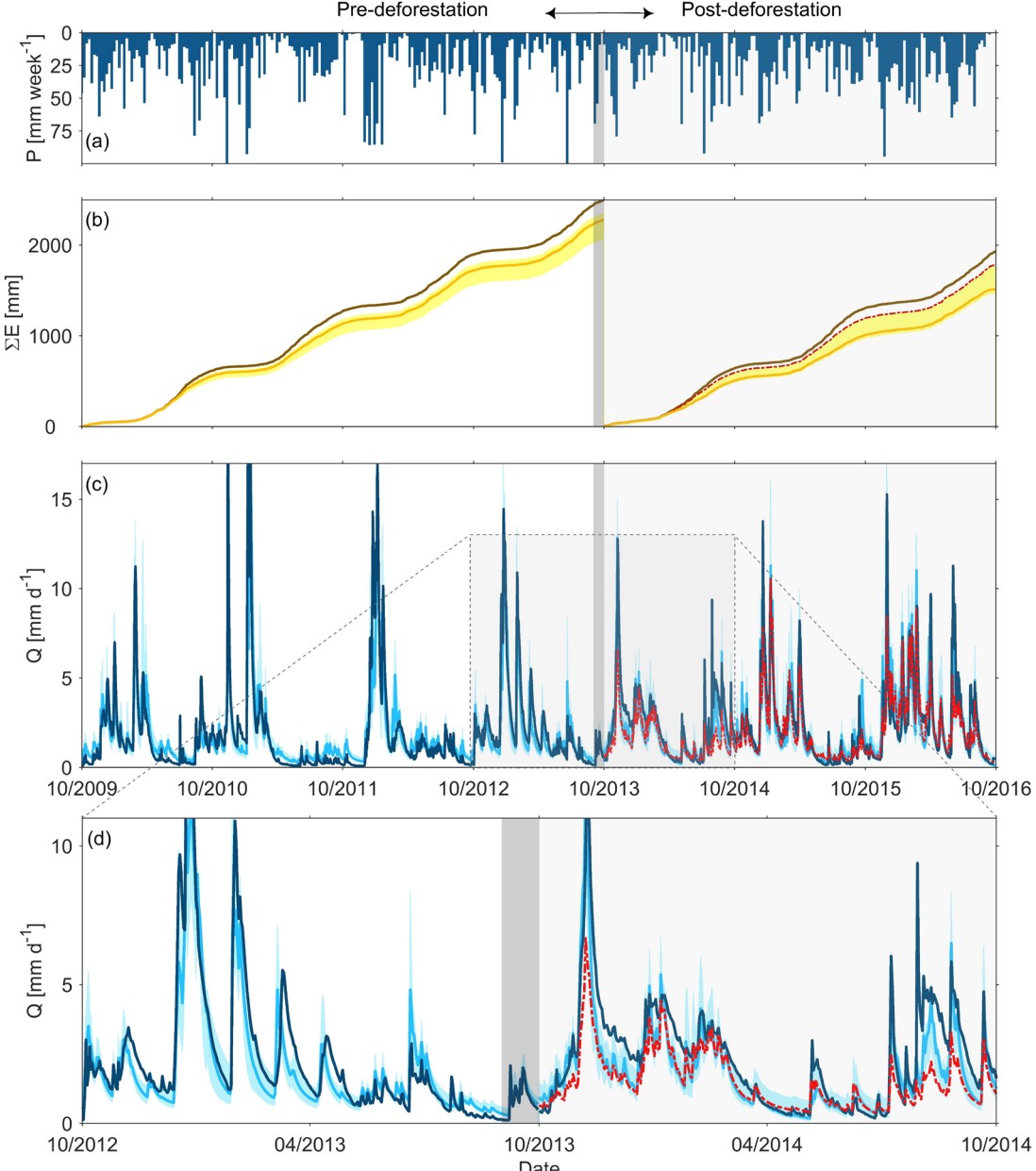

**Figure 3: (a)** Observed volume weighted monthly $\delta^{18}O$ signals in precipitation (grey dots; size of dots indicates the precipitation volume) and stream flow (green dots) as well as the best fit modelled $\delta^{18}O$ signal in the stream (green line) and the 5/95th percentile of all feasible solutions from pre- and post-deforestation calibration (green shaded area); **(b)** zoom-in of observed and modelled $\delta^{18}O$ signal in the stream for the 10/2012 – 10/2014 period.

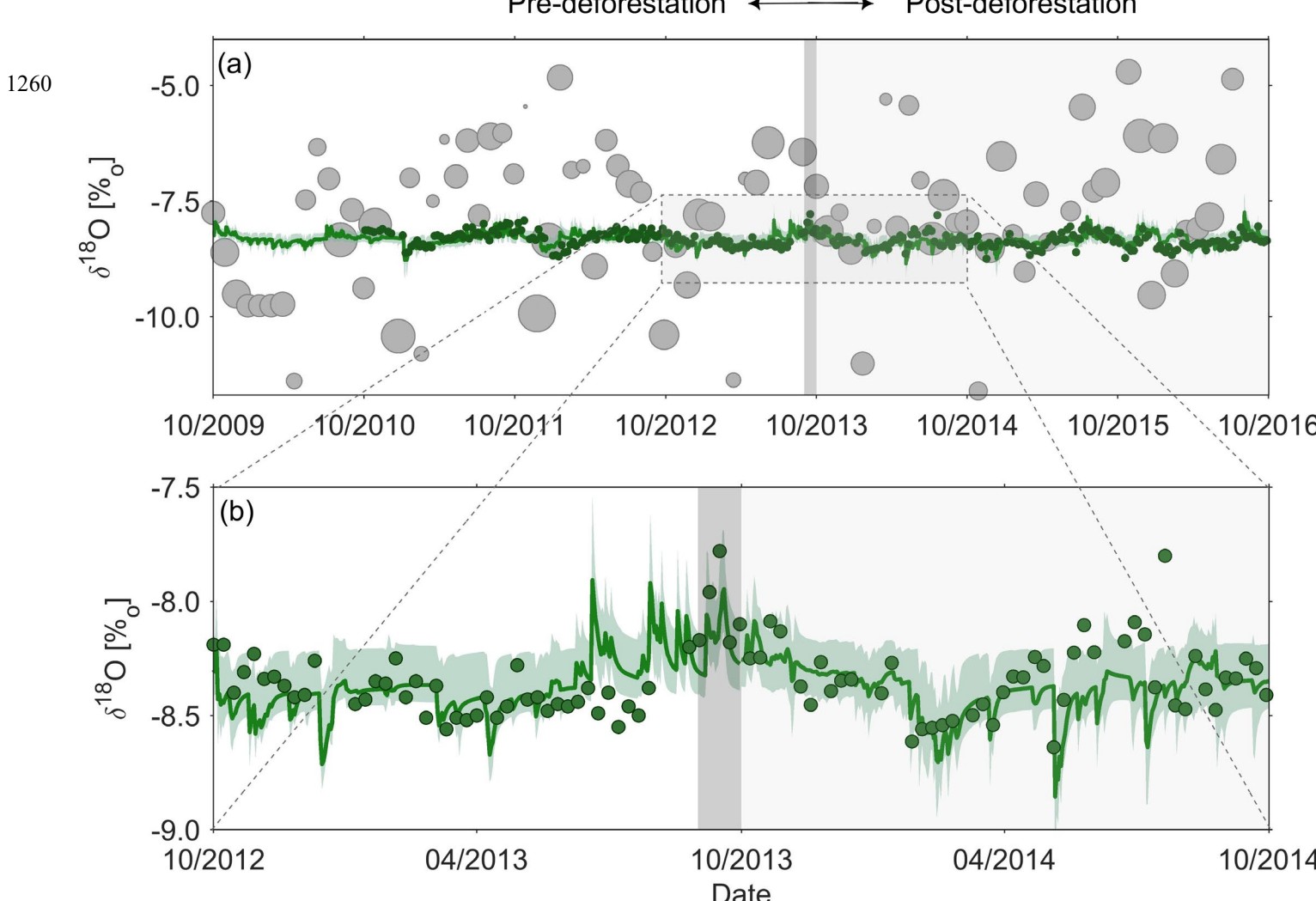

**Figure 4: Model structure used in this study. The light blue boxes indicate the hydrologically active individual storage volumes in the hillslope and riparian zones, respectively. The darker blue box $S_{S,p}$ indicates a hydrologically passive, i.e. $dS_{S,p}/dt = 0$, mixing volume. The blue lines indicate liquid water fluxes, the green lines indicate vapour fluxes. Model parameters are shown in red, adjacent to the model component they are associated with. All symbols are defined in Table 1.**

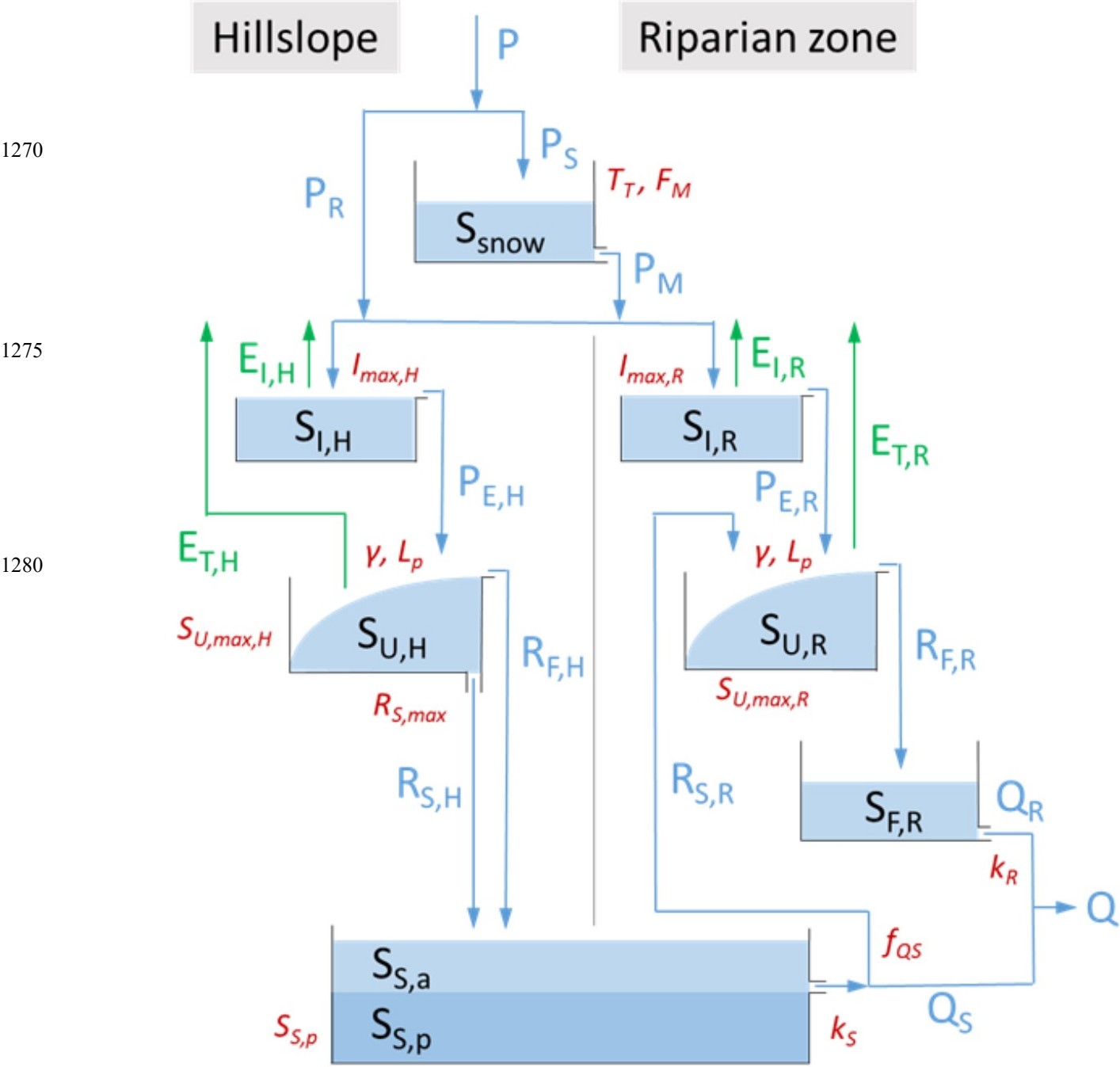

**Figure 5: (a) Positions of the individual years of the study period in the Budyko framework. The x-axis shows the aridity index $I_A = E_P/P$, the y-axis indicates the evaporative ration $E_A/P$ and the runoff ratio $C_R = 1 - E_A/P$. Pre-deforestation years are shown with blueish shades, post-deforestation years with greenish shades. The bold black lines indicate the energy and water limits, respectively. The dashed grey line is the theoretical-analytical Turc-Mezentsev relationship (Turc, 1954; Mezentsev, 1955). (b) The range of time series of storage deficits as computed according to equation 2, using values of $I_{max}$ from 0 to 4 mm. The maximum annual storage deficits $S_{D,j}$ are indicated by the arrows. The grey shaded area indicates the deforestation period. (c) Estimation of $S_{U,max}$ as the storage deficit associated with a 40-year return period $S_{D,40yr}$ using the Gumbel extreme value distribution for the pre-deforestation period. The blueish dots indicate the range of maximum annual storage deficits $S_{D,j}$ for each year in the four year pre-deforestation period. The dark grey shaded area indicates the envelop of least-square fits for the individual values of $I_{max}$. The light grey shaded area indicates the envelope of the 5/95th confidence intervals. The red line shows the plausible range for $S_{U,max}$.**

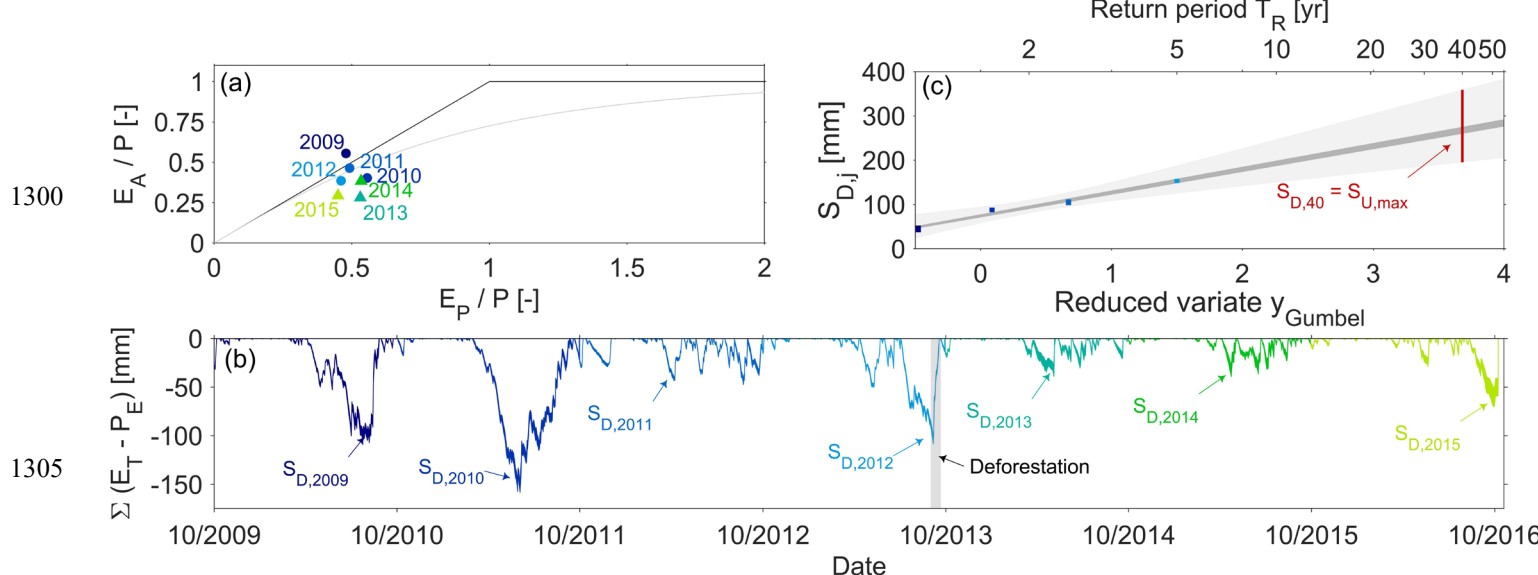

**Figure 6: (a) Model performance metrics for all variables and signatures.** $D_E$ **is the Euclidean distance to the perfect model. It combines all other performance metrics (Table 3) into one number (Eq.42). All performance metrics are formulated in a way that a value of 1 indicates a perfect fit. The boxplots summarize the performances of all parameter sets retained as feasible. The circle symbols indicate the performance of the best performing model in terms of** $D_E$**. The dark red shades indicate pre-deforestation model performance based on calibration in the pre-deforestation period. Orange shades indicate post-deforestation performance using the pre-deforestation parameter sets without further re-calibration. Yellow shades show the post-deforestation performance after model re-calibration in the post-deforestation period. (b)-(c) show flow duration curves, (d)-(e) show the peak distributions and (f)-(g) the autocorrelation functions for the pre- (red) and the post deforestation periods (orange and yellow), respectively. The black lines indicate the observed values, the dashed lines indicate the best fits and the shaded areas the 5/95th uncertainty interval of all solutions retained as feasible. The dark red shades indicate pre-deforestation model results based on calibration in the pre-deforestation period. Orange shades indicate post-deforestation model results using the pre-deforestation parameter sets without further re-calibration. Yellow shades show the post-deforestation model results after model re-calibration in the post-deforestation period.**

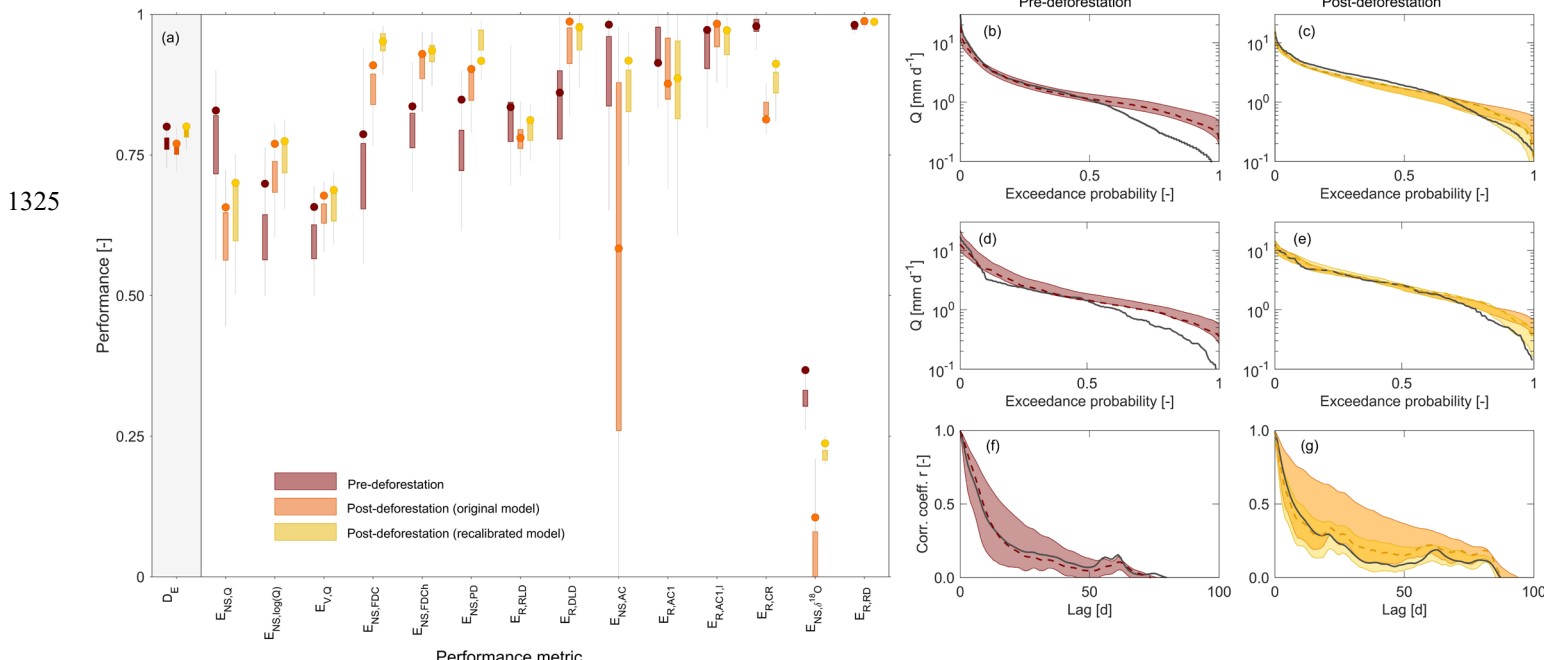

**Figure 7:** Observed mean P$_E$/P (dashed line), the range around observed mean P$_E$/P defined as acceptable (grey shaded area), the distribution of modelled mean P$_E$/P from all solutions that satisfy the behavioural thresholds for all performance metrics (Table 3) as well as the mean P$_E$/of the best solution in terms of D$_E$ (orange symbol) for (a) the 2009-2013 pre-deforestation period and (b) the 2013-2016 post-deforestation period. Note, only modelled solutions (yellow) that fall into the acceptable observed range (grey shaded) are kept as feasible. The fractions of time steps in the pre-deforestation (c) and the post-deforestation periods (d) in which the modelled relative soil moisture S$_{U,rel}$ falls within the pre-defined acceptable range around the observed relative catchment-average soil moisture. The blue symbols indicates the best solution in terms of D$_E$, the distributions indicate the set of solutions that satisfy the behavioural thresholds for all performance metrics (Table 3). The grey shaded areas indicate the region of acceptable solutions, i.e. solutions that fall at least 75% of the time steps into the acceptable interval. the fractions time steps. Note that only modelled solutions (light blue) that fall into the acceptable observed range (grey shaded) are kept as feasible. Pre-deforestation (e) and post-deforestation (f) time series of the acceptable range around the observed normalized, relative soil moisture (light grey shade), range of modelled normalized relative soil moisture for all solutions that satisfy all performance metrics ("unconstrained"; light blue) and for the set of feasible solutions that satisfy both, P$_E$/P and soil moisture constraints as shown in (a)-(d) ("constrained"; blue). The dark blue line indicates the modelled normalized, relative soil moisture of the best solution in terms of D$_E$.

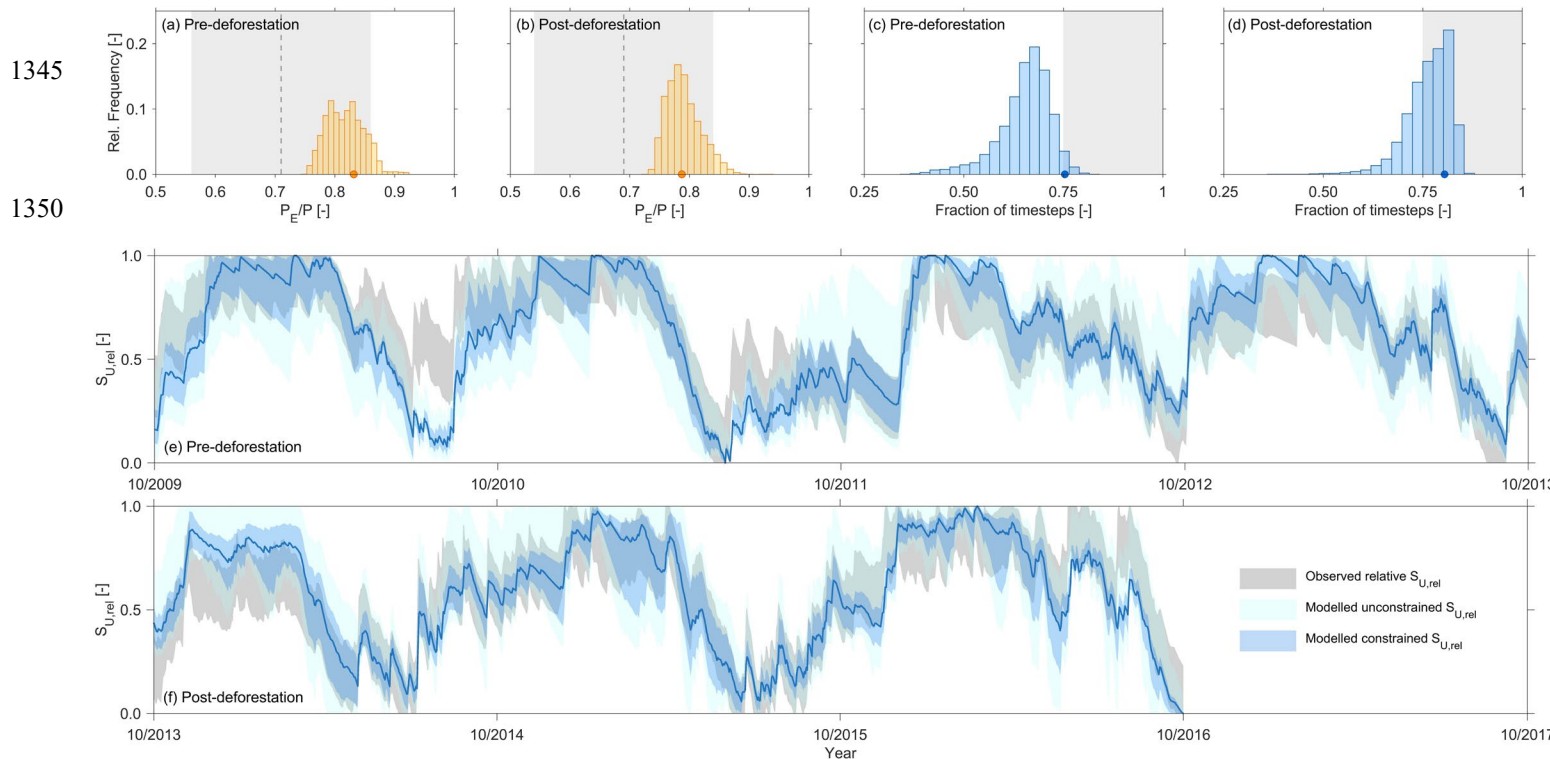

**Figure 8: Posterior distributions of selected parameters shown as empirical cumulative distribution function (lines) and the associated relative frequency distributions (bars). Red shades indicate calibration in the pre-deforestation period, Yellow shades indicate post-deforestation calibration. The dots indicate the parameter values associated with respective best fit models.**

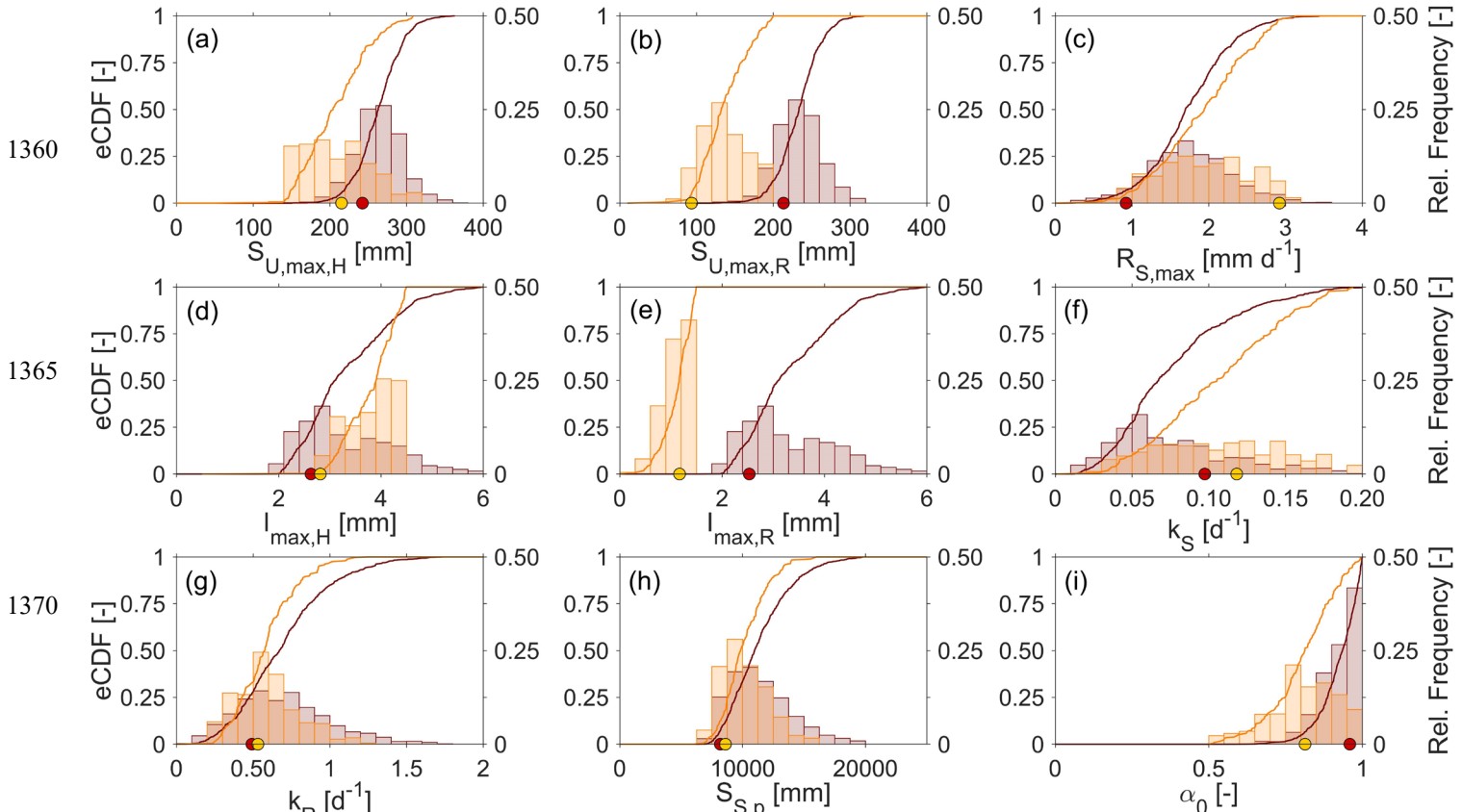

**Figure 9:** Panels in the left column show pre-deforestation (a) discharge, the coloured dots indicate to which period (dry, wet-up, wet, drying) the individual selected time steps belong; (b) the 5/95th percentiles of the empirical cumulative TTDs for wet (blue) and dry (red) periods, respectively; (c) the ensemble of the individual TTDs at the time steps indicated in (a). Panels in the middle column (d-g) compare the 5/95th percentiles of empirical cumulative TTDs between pre-deforestation (dark shades) and post-deforestation (light shades) periods for dry, wet-up, wet and drying conditions, respectively. Panels in the right column show post-deforestation (h) discharge, the coloured dots indicate to which period (dry, wet-up, wet, drying) the individual selected time steps belong; (i) the 5/95th percentiles of the empirical cumulative TTDs for wet (blue) and dry (red) periods, respectively; (j) the ensemble of the individual TTDs at the time steps indicated in (h). All distributions shown are truncated at 3 (post-deforestation) for 4 years (pre-deforestation), which coincides with the tracked period. For the remaining fractions, i.e. the difference to 1, it can only be said that they are older than 3 years but nothing more than that. The grey shaded areas indicate regions with ages > 3 months, thereby exceeding $F_{yw}$.

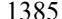

**Figure 10: (a)-(b) Pre- and post-deforestation time series of young water fractions $F_{yw}$ in discharge. The colour code indicates the transition between dry, wetting-up, wet and drying conditions. The bold black line shows the mean $F_{yw}$ of the best model fit, the grey shaded area shows the 5/95th percentile of $F_{yw}$ for all feasible model solutions; (c)-(d) pre- and post-deforestation sensitivity of $F_{yw}$ to discharge, using the same colour code as above to indicate dry, wetting-up, wet and drying conditions. The arrows in (d) indicate if there are statistically significant ($\uparrow$; $p < 0.05$) changes or not ($\leftrightarrow$) in the sensitivities between the post-deforestation period and the pre-deforestation period.**

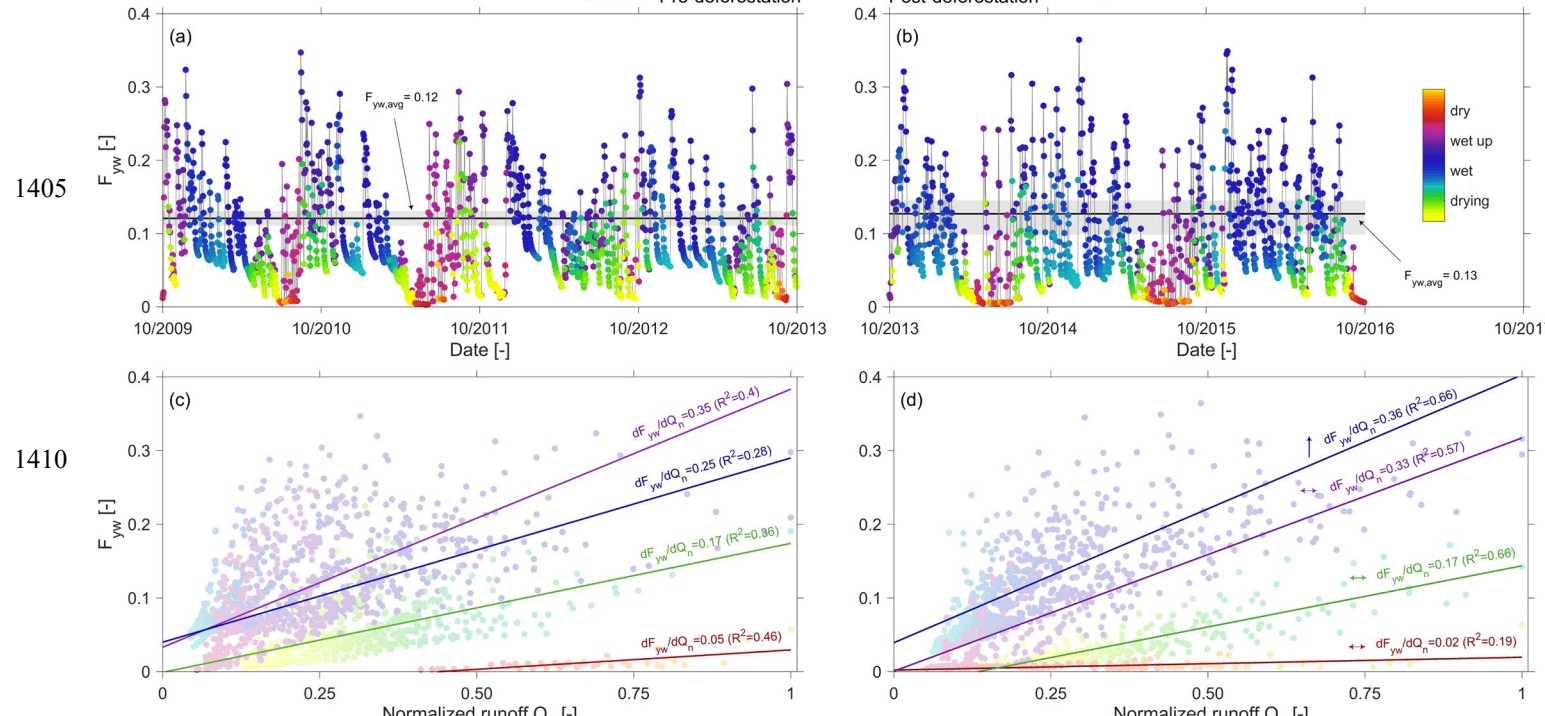

**Figure 11: Individual catchment overall SAS ω-functions for individual time steps under different wetness conditions in the (a) pre-deforestation period and (b) post-deforestation period. The insets show the relative water content in $S_{U,rel,mod} = S_U/S_{U,max}$ at the individual time steps.**

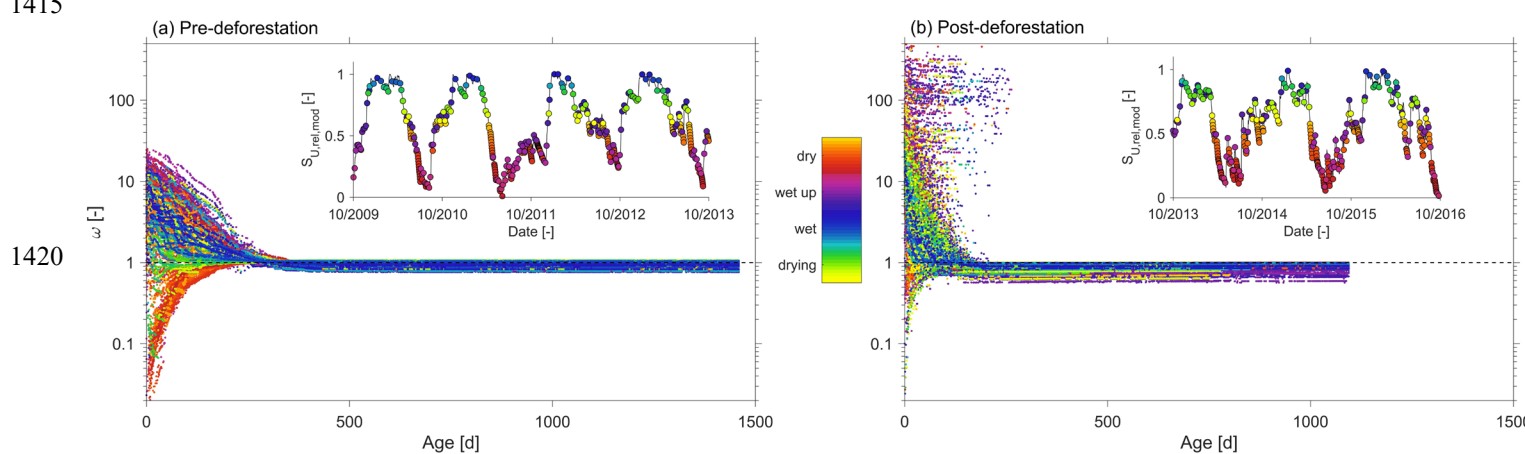
