# Peer review of "Reduction of vegetation-accessible water storage capacity after deforestation affects catchment travel time distributions and increases young water fractions in a headwater catchment"

_Hydrology and Earth System Sciences, 2020_

## Referee Comment (RC1) · Anonymous Referee #1 · 8 Jul 2020

This paper analyses the results of a deforestation experiment on the Wüstebach experimental catchment using conceptual flow and transport models. Overall the paper uses interesting approaches, but needs better structuring, and a more appropriate title. My major comments are as follows:

1. This paper essentially focuses on the Wüstebach catchment, a 0.39 km2 catchment in Germany. Clearly this catchment is a small and specific area. Whatever is found on this catchment cannot have general relevance, considering the place and scale dependence of hydrological processes. The title, however, is very trenchant and general, which contrasts with the specificity of the case study. I suggest a more specific title, more reflective of the individual headwater catchment that has been used in the analyses.

2. The introduction is very general, and projects a status quo that is much broader of what is needed to introduce this specific study. The readers unavoidably end up asking themselves: what is already known about this specific catchment? The Wüstebach catchment has been the object of a countless number of studies, which analysed the results of the deforestation experiment, and modelled its behaviour using many modelling approaches (https://experimental-hydrology.net/wiki/index.php?title=W%C3%BCstebach longterm experimental catchment). In particular, Wiekenkamp et al (2016) already analysed issues related to water balance, potential evaporation, and water storage associated to deforestation. I can see that the authors here use different methods. However, that in this catchment "Deforestation reduces the vegetation-accessible water storage in the unsaturated soil" (part of the paper title) is already clear from Wiekenkamp et al (2016), and other studies (e.g. works by Stockinger) have analysed the isotope data and evaluated MRT. These references are cited in the current manuscript. But they are not discussed to provide a clear motivation for the current paper, and to justify the novelty of the results.

3. The motivation for the choice of methods is unclear. E.g. why conceptual models, and SAS are chosen for the problem at hand? Wouldn't the result be obtainable with much simpler methods? It seems to me that if I look at the abstract or conclusions, a simple water balance calculation, and simple hydrograph separation techniques could have been sufficient to end up with the same outcomes. I understand that the authors are proposing a more elaborate approach. But why is that necessary?

4. The authors mention that they use an extensive multi-objective strategy, but in the end they use a single objective function. True that the objective function aggregates multiple objectives, but this cannot be defined multi-objective optimization, which would
require determining the Pareto-front between the various objectives.

5. I would have preferred to see separate results and discussion sections, to clearly see separate the outcome of this work from the outcomes of other works. Currently the blend adds to the confusion of not being able to appreciate the value of the current work compared to earlier work on the specific catchment.

6. Not clear to me why the Su,max of the model would be reflective of available water storage. For example, the model could have an Su,max of 200, but the variable storage between the reservoir can vary between e.g. 30 and 40 over one year, or between 10 and 150. So, Su,max sets an upper bound, but the real variability of the storage can be much smaller. The observation that 10.000 mm of water is necessary to attenuate the isotope signal suggests that indeed Su,max can be much larger than the dynamic range experienced by the catchment.

7. In terms of isotopes, it seems from the figure that there is an increase in the variability of the inputs. This leads to the question of how different are the inputs in the two periods, and whether the increase in young water fraction can be partly attributed to non-stationary inputs.

8. From the uncertainty analysis, it appears that some parameters, e.g. Rs,max, are poorly constrained. But I am guessing that this parameter can strongly affect the behaviour of the Su reservoir, which is the key storage analysed in this paper. Any comment on this?

---

## Referee Comment (RC2) · Anonymous Referee #2 · 20 Jul 2020

The manuscript by Hrachowitz et al investigates the effects of deforestation on several variables of ecohydrologic interest, such as the rate of transpiration, the partitioning between streamflow and evapotranspiration, the water storage available for plant uptake and the residence times of water in the landscape. To do so, the authors use data from a highly-instrumented small watershed in Germany (the Wustebach experimental catchment, 0.39 km2) that was subjected to a permanent deforestation of 21% of its surface. The main strategy used by the authors was to develop a parsimonious catchment-scale hydrologic and transport model, calibrated on observations, and use

it to quantify post-deforestation changes. The specific goals of the work, as stated in the introduction, are to analyse the observed post-deforestation changes through the lenses of change in the plant-available storage. The latter is a topic where the leading author has developed extensive research in the last years. Based on this main goal, the authors formulate three hypotheses that form the basis for the results and discussion.

The goals of the manuscript fully fit the scope of the journal, the data is of high quality and the methodology appears elaborate and generally appropriate. The authors produce a very large number of results, which may even be enough material for two papers, and I think this is where the manuscript becomes difficult to navigate. Although the overarching questions are clearly outlined in the introduction, the focus of the paper gets somewhat lost across the numerous results and analyses. For example (and see additional comments below), I find the water age analysis too detailed and beyond the main point of the article. I believe the focus of the manuscript should be narrowed and the material more selected to provide a cleaner storyline. Overall, while I believe these results deserve publication, I think major improvements on the manuscript structure and focus are required. I list additional comments below.

**Major points**

Novelty and relevance: I think the paper introduction is well developed but I somewhat miss the novelty and relevance of the results. E.g. what did we not know before? Does novelty lie specifically in the quantitative (rather than qualitative) evaluation of change?

Visual assessment of model results: Figures are all high-quality but they are not always informative. Figure 2 is particularly important because it shows timeseries of modelled and observed variables, but it is rather ineffective and it should be redesigned (see detailed comments below).

Model purpose: the model produces a large number of outputs and I appreciate that the authors clearly discuss the limitation of the modelling framework. I think it would be

[Figure]

useful to declare upfront the capabilities of such a model. For example, given its design and the data used to assess its performance, what can the model be reasonably used for in this context? Also it seems that the model would in principle be able to estimate the age of evaporative fluxes, but the authors do not show such results. Why? And why is the model not used to estimate the (change in) partitioning between Q and ET?

Estimates of SU,max: the reduction in SU,max after deforestation is very pronounced. I'd appreciate some discussion of how a 21% reduction of forest cover may lead to a 50% reduction in SU,max. Does the location (riparian area VS hillslope) play a role? Would you expect important differences if the deforestation had occurred away from the river network? Could this model be used to make such a prediction?

Age analysis: as mentioned before, I think this is a very complete analysis but it seems to go too deep compared to the scope of the paper. There are several concepts and results that appear complex (e.g. not just the TTD/RTD ratio or the young water fraction, but also their sensitivities to wetness, before and after deforestation) and would fit a paper that specifically focuses on the effects of deforestation on catchment residence times. But given the broader objectives, a selection of the results might help clarify the message.

**Detailed comments**

Title: isn't it obvious that deforestation "affects" travel times? Can suggest how it affects them

50-56: very long sentence. Consider breaking it

70-71: unclear sentence

93-96: this sentence includes some slang. Consider rephrasing. Also, it partly seems a repetition of lines 49-52

98: "With increasing storage, the hydrological memory of a system increases" → can

be easily misinterpreted because one may think that, for a system with a given water storage, streamflow age would increase when the storage increases, but it is usually found (e.g. Harman, 2015, Water Resources Research) that the opposite applies.

Figure 2: this figure is very important for the manuscript but I find it rather inefficient. Subplot (a) does not seem to show anything that a reader could grasp. Subplot (c) is very compressed –it is never easy to compress 7 years of data into a single panel– but then it is almost impossible to inspect the results. Subplot (d) is uninformative because tracer in streamwater is not shown at an adequate scale. Is tracer precipitation data really useful here? If you really want to show it, maybe you could report some monthly means (so we get at least the seasonality of the input) and on a rescaled y-axis? Can you also add a legend directly into the plot to facilitate understanding of what the plotted variables are?

162: "To quantify effects of deforestation on SU,max and, as a consequence of that, on the age structure of water" I find this slightly misleading as it seems that the age structure is only affected through SU,max (and not directly).

166: similar to 162, I don't see why you estimate the effects of these changes only. Aren't the two problems partially separate? You can get to (2) even without (1) or am I uncorrect?

Eq 39 and 41: what is the horizontal bar above the lowercase omega?

336: this explains how you translate a physical process into a model component. But I recommend explaining why a "passive" storage is needed. It is not a model artefact, but rather the real amount of water that is involved in the transport process. Other good classic references for this is "Birkel, C., C. Soulsby, and D. Tetzlaff (2011), Modelling catchment-scale water storage dynamics: Reconciling dynamic storage with tracer- inferred passive storage, Hydrol. Processes, 25(25), 3924–3936, doi:10.1002/hyp.8201"

353: Ok fine, but how do you initialize the model? Which d18O initial composition is

assigned to the different compartments?

385: I suggest replacing "the data" with "results"

408: start with "In our study, "

417: SU,max = 90+-149? If the standard deviation is much larger than the mean, wouldn't it be better to avoid completely this exercise? Or maybe mention it here but not stress it later as a real result, given it comes with considerable uncertainty?

428 and 435: as noted in comments on Figure 2, it is difficult to judge from this figure because too much data is compressed in it and the scaling does not look appropriate

Figure 5: please add some legend to make the colouring more intuitive

Figure 6: why showing cumulative frequencies? I find it difficult to evaluate how much parameters are constrained from these curves.

440: "...show that most model parameters are reasonably well identified" but it is difficult to actually evaluate

494: if I haven't stressed this enough, this is invisible in the figure

513: I do not see how a value of 120 falls "reasonably well" into a "plausible" range [-59, 239]

519: I get the point, but cannot really evaluate overlapping from figure 6. Rather from table 2. Then I see that Lp is the only other parameter with a significant change pre-post deforestation. Might this be worth of a comment?

534: [again on figures] "is evident", but nothing is evident from figure 2d

541-566: this is a nice analysis but I wonder whether it is really necessary as it is more about general TTD and RTD dynamics rather than on effects of deforestation. In other words, this seems to go beyond the scope of the manuscript. A large simplification would make the manuscript easier to read.

[Figure]

544: please specify how you computed Fyw because a reader will expect it is estimated using kirchner's 2016 method.

33: the change in Fyw from 0.11 to 0.13 does not seem significant considering the possible uncertainties in the estimate.

**Typos and language**

Figure 2d: typo in the permil symbol in the y-axis label.

70: the vegetation… presentS

264: it is reaches

548: "Stream water can contain Fyw" is not a correct formulation. Rather "stream water can contain up to 30% of young water" or similar

532-534: please reformulate this initial sentence, which is unclear The terminology "partly much lower" (452) and "partly considerable" (572) is a bit odd

425: title: Deforestation effects "on the catchment model" sounds a bit odd.

---

## Referee Comment (RC4) · Anonymous Referee #4 · 14 Aug 2020

This manuscript presents the effects of partial deforestation on water storage and water ages in the German Wüstebach catchment. For this study, the authors performed water balance analyses and modelling exercises based on 7 years of hydrometric and water stable isotope data. One major finding of the study is that the vegetation-accessible storage volume in the unsaturated zone, *SUmax*, was significantly reduced after the partial deforestation; the authors hypothesize that this reduction in *SUmax* can largely be explained with young water being routed quickly to the stream during wet conditions, so that less water reached the unsaturated zone *SU*.

The paper is well written and the figures are informative. I only have some minor comments and questions that the authors should address.

1. The physical meaning of *SUmax* not fully clear to me: its definition in the introduction is "water-filled pore volume between field capacity and permanent wilting point that is within the reach of active roots". This suggests that *SUmax* depends on water content in the soil and the active rooting depth. Does this mean that *SUmax* will decrease when water influx is reduced and/or roots become shorter? Then, the major result of the study (i.e., *SUmax* is reduced after deforestation; L421-424) is not surprising but rather expected because fewer roots will lead to a smaller catchment-average active rooting depth.

2. L135: How many measurements of rooting depth are available to justify the general assumption that the maximum rooting depth across the catchment is 50cm? What is the depth of the groundwater table and is it possible that capillary rise from the groundwater supplies these shallow-rooted plants?

3. Are there any additional data that support your claim of a large groundwater storage in the Wüstebach catchment? It is surprising to me that no groundwater table and soil moisture observations have been considered for explaining many of the processes you propose.

4. How were dry, drying, wet and wetting-up periods defined (L545, L561, Fig. 8)?

5. Fig. 8 and Sect. 5.3: How was the daily young water fraction calculated and what is the associated uncertainty? Are your interpretations robust with respect to the uncertainties in *Fyw*?

6. L434: From Fig 2d it is hard to see how well the model simulated the $\delta$18O time series because the data points cover each other too much.

7. Fig 8d, c: It is not clear to me, which data points were used to obtain these regression lines? Especially the dark-blue regression lines (wet conditions) do not seem to fit the

dark blue data points at all, and thus, the associated regression slopes should be considered with caution (e.g. in L588).

---

## Author Comment (AC1) · 8 Oct 2020

Comment:

This paper analyses the results of a deforestation experiment on the Wüstebach experimental catchment using conceptual flow and transport models. Overall the paper uses interesting approaches, but needs better structuring, and a more appropriate title.

*Reply:*

*We thank the reviewer for the overall positive assessment of our study as well as for her/his thoughtful and detailed comments.*

Comment:

This paper essentially focuses on the Wüstebach catchment, a 0.39 km2 catchment in Germany. Clearly this catchment is a small and specific area. Whatever is found on this catchment cannot have general relevance, considering the place and scale dependence of hydrological processes. The title, however, is very trenchant and general, which contrasts with the specificity of the case study. I suggest a more specific title, more reflective of the individual headwater catchment that has been used in the analyses.

*Reply:*

*On re-reading the title we agree with the reviewer and realize that the title is more general than intended. To better reflect the location specificity but also to place more emphasis on the actual objective and innovative aspect of this study we will reformulate the title into: "Reduction of vegetation-accessible water storage capacity after deforestation affects travel time distributions and increases young water fractions in a headwater catchment."*

Comment:

The introduction is very general, and projects a status quo that is much broader of what is needed to introduce this specific study. The readers unavoidably end up asking themselves: what is already known about this specific catchment? The Wüstebach catchment has been the object of a countless number of studies, which analysed the results of the deforestation experiment, and modelled its behaviour using many modelling approaches (https://experimental-hydrology.net/wiki/index.php?title=W%C3%BCstebach_long-term_experimental_catchment). In particular, Wiekenkamp et al. (2016) already analysed issues related to water balance, potential evaporation, and water storage associated to deforestation. I can see that the authors here use different methods. However, that in this catchment "Deforestation reduces the vegetation-accessible water storage in the unsaturated soil" (part of the paper title) is already clear from Wiekenkamp et al. (2016), and other studies (e.g. works by Stockinger) have analysed the isotope data and evaluated MRT. These references are cited in the current manuscript. But they are not discussed to provide a clear motivation for the current paper, and to justify the novelty of the results.

*Reply:*

*The reviewer is completely correct in saying that the Wüstebach study catchment has over the past years been subject to numerous studies. We highly appreciate the notion of the reviewer that for the reader the added value and innovative aspects of this study may not be completely clear yet. We agree that it will therefore be helpful for the reader to more explicitly describe how this study fits into the context of previous work in the Wüstebach and which knowledge it specifically adds.*

*Briefly, the Wiekenkamp et al. (2016) paper quantified in an elegant study the effects of deforestation on the partitioning of water fluxes, based on discharge and eddy covariance observations. Overall, they found that deforestation decreased actual evaporation/transpiration, which led to higher levels of soil moisture and eventually to increases in discharge. In contrast, the aim of our study was to establish a quantitative mechanistic link between these observed decreases in transpiration and changes to sub-surfaces properties of the system (and thus model parameters) to provide an explanation of **why** deforestation reduces evaporative fluxes and increased soil moisture and discharge – after all the atmospheric water demand (here approximated by potential evaporation) remained stable pre- and post-deforestation.*

*Similarly, detailed previous work of some of our co-authors (Stockinger at al., 2019) quantified travel times and changes therein over time while Wiekenkamp et al. (2020) found evidence for increased post-deforestation occurrence of preferential flows. As an extension of that work, we here quantify the effect of deforestation on subsurface system properties and found that changes in these subsurface properties can to some extent explain **why** deforestation affects travel times and in particular young water ages in the Wüstebach.*

*Overall our results provide evidence that post-deforestation changes in the water balance and the hydrological response dynamics can, to a large part, be traced back to changes in the soil pore space that can be \*__accessed__\* by evaporative fluxes (i.e. evaporation and/or transpiration) to satisfy atmospheric water demand. Vegetation, through its root system, can access pores and efficiently extract water from them for transpiration in, relatively spoken, deep parts of the soil. Soil evaporation, in contrast, can only extract water from comparably much shallower parts of the soil, due to increasingly limited turbulent exchange with depth. In other words, after the top soil is dry, soil evaporation is largely "blocked" while root-water uptake from deeper soil layers for transpiration can continue. In case of deforestation, plants are removed and the associated transpiration stops. The pore space that was accessed by transpiration before, can after deforestation then not be accessed by evaporative fluxes (i.e. soil evaporation) anymore and soil moisture levels in these pores remains longer close to field capacity. Informally spoken, this pore volume is thus lost as "reservoir" for evaporative fluxes.*

*In contrast to previous studies, where this pore volume is either estimated as model calibration parameter or based on very scarce data of root depths and soil porosities, our results here suggest (1) that this root-accessible pore volume ($S_{U,max}$) and its change due to deforestation can be meaningfully estimated from water balance data at the catchment-scale, (2) that it is therefore also a meaningful and directly observable catchment-scale effective parameter for use in process-based models and, most importantly (3) that changes in that hydrologically relevant subsurface property/model parameter can explain much of **why** the hydrological response as well as travel times and young water fractions changed after deforestation.*

*We will provide a more detailed context of previous studies together with a more explicit definition of the objectives of this study and how they fit into the context and actually **link** the results of previous studies.*

Comment:

The motivation for the choice of methods is unclear. E.g. why conceptual models, and SAS are chosen for the problem at hand? Wouldn't the result be obtainable with much simpler methods? It seems to me that if I look at the abstract or conclusions, a simple water balance calculation, and simple

hydrograph separation techniques could have been sufficient to end up with the same outcomes. I understand that the authors are proposing a more elaborate approach. But why is that necessary?

*Reply:*

*The motivation for the choice of the modelling strategy, i.e. a process-based hydrological model with an integrated SAS-function based tracer module, is closely linked to our overall objective of this study, i.e. to develop a better understanding of __why__ the water balance, the hydrological response dynamics as well as the travel times in the Wüstebach catchment were subject to change after deforestation even though climatic conditions did largely not change. We wanted to understand (1) which (subsurface) system property was most affected by deforestation, (2) if this real-world property (or when used in models: parameter) and its change after deforestation can be meaningfully quantified at the catchment-scale and (3) if changes in this property can explain the observed post-deforestation changes in the water balance, response dynamics and travel times.*

*The simpler methods referred to by the reviewer are powerful tools to quantify overall hydrological system characteristics. However, due to their simplicity they mostly need to remain process-agnostic and can therefore provide us only with limited information on the underlying processes and mechanistic reasons. We will clarify this in the revised manuscript.*

Comment:

The authors mention that they use an extensive multi-objective strategy, but in the end they use a single objective function. True that the objective function aggregates multiple objectives, but this cannot be defined multi-objective optimization, which would require determining the Pareto-front between the various objectives.

*Reply:*

*We respectfully but strongly disagree with this comment. Only because multiple performance metrics f (i.e. objectives and criteria) were compressed into one summary statistic does not mean that the parameter selection strategy was not based on multiple objectives.*

*It is true that the solution space of a multi-objective optimization problem typically spans a front of pareto-optimal solutions. However, this pareto front only quantifies uncertainties arising from the choice and weight of and the trade-offs between the individual objectives. It does not account for data uncertainty and the idea of behavioural models, like in the GLUE framework (Beven and Binley, 1992). To embrace that we here did not treat \*__only__\* the actual pareto optimal solutions as feasible but also potential solutions behind the pareto front, i.e. we applied the GLUE method in a multi-dimensional case. This is schematically illustrated in Figure R1 below for a theoretical multi-objective evaluation based on two performance metrics $f_1$ and $f_2$ (note that the analysis of this manuscript was based on n=12 performance metrics instead, but based on the same reasoning).*

*We first defined a performance threshold for each individual performance metric f. Only solutions that fell within that threshold for each performance metric were kept as behavioural. The resulting solution space then implicitly contains the pareto front.*

*Based on that we then further evaluated each solution that was kept as behavioural based on its multi-dimensional distance $D_E$ from the "perfect model". The "best" solution is the one with the shortest distance $D_E$, thus inherently also being a point on the pareto-front. All other solutions accepted as feasible are further away from the "perfect model" and can be located on the pareto front or behind it.*

*If less (or in the extreme case only one) performance metrics than used in our analysis (n=12) had been used, the subset of retained behavioural solutions would be larger and their associated performances in terms of $D_E$ different, which clearly demonstrates the effect of multiple performance metrics even when compressed into one summary statistic $D_E$. The above outlined strategy not only includes all pareto optimal solutions but also allows for additional uncertainty, thereby not only being defined as but actually extending and going beyond classic multi-objective parameter selection strategies as also similarly described and formalized by various recent papers (e.g. Efstradiadis and Koutsoyiannis, 2010; Gharari et al., 2013).*

[Figure]

*Figure R1: Schematic sketch of two-dimensional model performance space as defined by two performance metrics $f_1$ and $f_2$ (i.e. multi-objective). The theoretical "perfect model" would plot in the origin. The yellow shade indicates all model solutions. The light blue lines indicate the respective behavioural thresholds for $f_1$ and $f_2$ below which solutions are kept as behavioural and thus feasible. The red shaded area indicates the two-dimensional space of solutions accepted as behavioural. The bold red line indicates the set of pareto optimal solutions and the blue circle the single "best" solution with the lowest distance $D_E$ to the "perfect model". The dashed blue circles-arcs indicate iso-performance levels in terms of $D_E$, i.e. solutions that have equal distances from the "perfect model" (after Gharari et al., 2013)*

Comment:

I would have preferred to see separate results and discussion sections, to see separate the outcome of this work from the outcomes of other works. Currently the blend adds to the confusion of not being able to appreciate the value of the current work compared to earlier work on the specific catchment.

*Reply:*

*We acknowledge and appreciate this comment. However, given the stepwise nature of the analysis we would strongly prefer to have the results of the individual steps closely associated to the interpretation thereof. However, we will strengthen and clarify the link to previous studies in the revised manuscript.*

Comment:

Not clear to me why the Su,max of the model would be reflective of available water storage. For example, the model could have an Su,max of 200, but the variable storage between the reservoir can vary between e.g. 30 and 40 over one year, or between 10 and 150. So, Su,max sets an upper bound, but the real variability of the storage can be much smaller. The observation that 10.000 mm of water is necessary to attenuate the isotope signal suggests that indeed Su,max can be much larger than the dynamic range experienced by the catchment.

*Reply:*

*The reviewer is absolute right in stating that $S_{U,max}$ is an upper bound. To further reply to this interesting comment it will be helpful to upfront remind us again of how $S_{U,max}$ is actually defined. In the original*

*manuscript, $S_{U,max}$ is described as the "pore volume between field capacity and permanent wilting point that is within the reach of active roots" and in the text referred to as "vegetation-accessible water storage **capacity**" and thus the **maximum** vegetation-accessible water storage volume in the unsaturated soil.*

*As such, the upper limits of $S_{U,max}$ are by definition physically bound by*

(1) *Depth of groundwater table. Although fluctuating, the groundwater table in the study catchment remains at depths below 1 m for much of the year even in the riparian zone (Bogena et al., 2015) and can be expected to be considerably deeper on the hillslopes. Thus assuming a conservative upper bound of catchment-average depth of the groundwater table at ~ 5 m (assuming that the lowest groundwater table at each point in the catchment is at the elevation of the nearest stream), porosity of the silty clay loam (Bogena et al., 2018) soil of 0.4 and field capacity at a relative pore water content of 0.5 suggests an upper limit of $S_{U,max}$ at ~ 1000 mm.*

(2) *Root-depths. However, actual roots are often shallower than these 5m of the groundwater table. Although sufficient detailed data on root depths are not available in the study catchment, there is no evidence for systematic and wide-spread roots extending to below 0.5 m. This is broadly consistent with direct experimental evidence that roots of temperate forests in general (Schenk and Jackson, 2002) and Picea species in particular mostly remain rather shallow (< 1 m; e.g. Schmid and Kazda, 2001) and with indirect evidence that Picea species rarely tap groundwater and are thus comparatively shallow (e.g. Evaristo and McDonnell, 2017). As a conservative back-of-the-envelope calculation, assuming a maximum plausible catchment-average root depth of 2 m (which comes close to the average observed soil depth reported in Graf et al., 2014), porosity of the silty clay loam (Bogena et al., 2018) soil of 0.4 and field capacity at a relative pore water content of 0.5 suggests an upper limit of $S_{U,max}$ at ~ 400 mm.*

*Thus given these estimates, we think it is very unlikely that $S_{U,max}$ is much larger than that. In addition, there is increasing and robust evidence that the magnitude of $S_{U,max}$ is actually based on optimality principles (e.g. Kleidon, 2004; Schymanski et al., 2008; Guswa et al., 2008; Yang et al., 2016; Speich et al., 2018 and many more): individual plants which have survived and are thus present in an ecosystem were well adapted to past conditions. This entails that they have root systems **large enough** to provide them with access to enough water, **but not more than that**. Unnecessary allocation of resources to subsurface growth (and maintenance) would have led to a lack of resources for above-surface growth. These plants would have eventually lost out in competition for light. By extension, the question of how much water was necessary to be stored (i.e. held below field capacity and thus against gravity) and accessible for vegetation (i.e. $S_{U,max}$) can then be inferred from what vegetation \*actually\* transpired under extreme conditions (here: dry spells with 20 year return periods), as done in this manuscript.*

*However, please also note that as mentioned above, here and in many other landscapes world-wide (e.g. Fan et al., 2017) most roots do not extend to the depth of the groundwater table. This leads to the presence of an unsaturated zone **below** the root zone. This zone is hydrologically largely "passive" as it is outside the influence of roots and given its relatively deep position essentially also outside the influence of soil evaporation: evaporative fluxes can therefore not extract water from pores in this part of the subsurface and water entering this zone only undergoes some delay before it is released from that zone again as groundwater recharge or direct drainage to a stream (e.g. through preferential flow paths). As no significant volumes of water can be drained below field capacity the unsaturated soil below the root zone remains close to field capacity for much of the year (except for the moments when a wetting front passes through). This is illustrated by the much lower variability in soil moisture content in these deeper soil layers (see e.g. Graf et al., 2014). Albeit being hydrologically largely passive (at least at time-scales of more than a few days dS/dt~0), this transitional zone between the root zone and the groundwater provides a mixing volume for tracers. This was implemented in one of our recent*

*studies (Hrachowitz et al., 2015). However, here we finally decided not to explicitly account for this effect as it requires 2 additional model calibration parameters, i.e. the size of the mixing volume and the time lag for releasing water. After intensive model testing it was found that these two parameters cannot be well constrained at all. In particular the size of the mixing volume showed considerable trade-offs with the parameter of the passive mixing volume $S_{S,p}$ here. Instead of explicitly representing this volume as individual model component, we therefore implicitly accounted for this additional mixing volume in $S_{S,p}$. It can thus be assumed that, in reality, part of the 10.000 mm of $S_{S,p}$ is actually a mixing volume in the transitional zone. It plausible to assume that here the magnitude of this volume does realistically not exceed ~500-600mm (i.e. the difference between the storage capacity above the groundwater and the root zone – see (1) and (2) above).*

*We will clarify this in the revised manuscript.*

Comment:

In terms of isotopes, it seems from the figure that there is an increase in the variability of the inputs. This leads to the question of how different are the inputs in the two periods, and whether the increase in young water fraction can be partly attributed to non-stationary inputs.

*Reply:*

*The variability in the observed precipitation isotope inputs indeed slightly increased after deforestation (mostly due to higher sampling frequency) from a standard deviation of ~3.2 ‰ to ~3.8 ‰, while the stream water isotope variability remained stable at a standard deviation of ~0.2 ‰. This entails that the damping ratio between precipitation slightly increased and the input signal was thus proportionally more strongly damped. Following our current understanding of tracer dynamics, a higher damping ratio is associated with older water (e.g. McGuire and McDonnell, 2006). We therefore believe that, if this slight change in variability had an effect, it would rather **decrease** than increase young water fractions.*

Comment:

From the uncertainty analysis, it appears that some parameters, e.g. Rs,max, are poorly constrained. But I am guessing that this parameter can strongly affect the behaviour of the Su reservoir, which is the key storage analysed in this paper. Any comment on this?

*Reply:*

*As correctly observed by the reviewer, some model parameters are less well constrained than others, which remains a problem in essentially all inverse model implementations in environmental sciences. It is also true that the specifically mentioned parameter $R_{S,max}$ has some influence on $S_{U,max}$. However, the results suggest that the absolute magnitudes of the associated flux $R_{S,H}$ are too low (i.e. < 10% of the water balance) to significantly influence $S_{U,max}$. This is further supported by the fact that $S_{U,max}$ is, for conceptual model standards, very well constrained: no matter the value of $R_{S,max}$, $S_{U,max}$ remains within the same narrow range. We will add this to the discussion in the revised manuscript.*

References:

Beven, K., & Binley, A. (1992). The future of distributed models: model calibration and uncertainty prediction. Hydrological processes, 6(3), 279-298.

Bogena, H. R., Bol, R., Borchard, N., Brüggemann, N., Diekkrüger, B., Drüe, C., ... & Missong, A. (2015). A terrestrial observatory approach to the integrated investigation of the effects of deforestation on water, energy, and matter fluxes. Science China Earth Sciences, 58(1), 61-75.

Bogena, H.R., Montzka, C., Huisman, J.A., Graf, A., Schmidt, M., Stockinger, M., ... & Lücke, A. (2018). The TERENO-Rur Hydrological Observatory: A multiscale multi-compartment research platform for the advancement of hydrological science. Vadose Zone Journal, 17(1).

Efstratiadis, A., & Koutsoyiannis, D. (2010). One decade of multi-objective calibration approaches in hydrological modelling: a review. Hydrological Sciences Journal–Journal Des Sciences Hydrologiques, 55(1), 58-78.

Evaristo, J., & McDonnell, J. J. (2017). Prevalence and magnitude of groundwater use by vegetation: a global stable isotope meta-analysis. Scientific reports, 7(1), 1-12.

Fan, Y., Miguez-Macho, G., Jobbágy, E.G., Jackson, R.B., & Otero-Casal, C. (2017). Hydrologic regulation of plant rooting depth. Proceedings of the National Academy of Sciences, 114(40), 10572-10577.

Gharari, S., Hrachowitz, M., Fenicia, F., & Savenije, H. H. G. (2013). An approach to identify time consistent model parameters: sub-period calibration. Hydrology and Earth System Sciences, 17(1), 149.

Graf, A., Bogena, H.R., Drüe, C., Hardelauf, H., Pütz, T., Heinemann, G., & Vereecken, H. (2014). Spatiotemporal relations between water budget components and soil water content in a forested tributary catchment. Water resources research, 50(6), 4837-4857.

Guswa, A.J. (2008). The influence of climate on root depth: A carbon cost-benefit analysis. Water Resources Research, 44(2).

Hrachowitz, M., Fovet, O., Ruiz, L., & Savenije, H.H.G. (2015). Transit time distributions, legacy contamination and variability in biogeochemical $1/f\alpha$ scaling: how are hydrological response dynamics linked to water quality at the catchment scale?. Hydrological Processes, 29(25), 5241-5256.

Kleidon, A. (2004). Global datasets of rooting zone depth inferred from inverse methods. Journal of Climate, 17(13), 2714-2722.

McGuire, K. J., & McDonnell, J. J. (2006). A review and evaluation of catchment transit time modeling. Journal of Hydrology, 330(3-4), 543-563.

Schenk, H.J., & Jackson, R.B. (2002). Rooting depths, lateral root spreads and below-ground/above-ground allometries of plants in water-limited ecosystems. Journal of Ecology, 90(3), 480-494.

Schmid, I., & Kazda, M. (2001). Vertical distribution and radial growth of coarse roots in pure and mixed stands of Fagus sylvatica and Picea abies. Canadian Journal of Forest Research, 31(3), 539-548.

Schymanski, S.J., Sivapalan, M., Roderick, M.L., Beringer, J., & Hutley, L B. (2008). An optimality-based model of the coupled soil moisture and root dynamics. Hydrology and earth system sciences, 12(3), 913-932.

Speich, M.J., Lischke, H., & Zappa, M. (2018). Testing an optimality-based model of rooting zone water storage capacity in temperate forests. Hydrology and Earth System Sciences, 22(7), 4097-4124.

Stockinger, M. P., Bogena, H. R., Lücke, A., Stumpp, C., & Vereecken, H. (2019). Time variability and uncertainty in the fraction of young water in a small headwater catchment. Hydrology & Earth System Sciences, 23(10).

Wiekenkamp, I., Huisman, J. A., Bogena, H. R., Lin, H. S., & Vereecken, H. (2016). Spatial and temporal occurrence of preferential flow in a forested headwater catchment. Journal of hydrology, 534, 139-149.

Wiekenkamp, I., J.A. Huisman, H.R. Bogena, and H. Vereecken (2020): Spatiotemporal Changes in Sequential and Preferential Flow Occurrence after Partial Deforestation. Water 12(1), 35.

Yang, Y., Donohue, R.J., & McVicar, T.R. (2016). Global estimation of effective plant rooting depth: Implications for hydrological modeling. Water Resources Research, 52(10), 8260-8276.

---

## Author Comment (AC2) · 8 Oct 2020

Comment:

The specific goals of the work, as stated in the introduction, are to analyse the observed post-deforestation changes through the lenses of change in the plant-available storage. The latter is a topic where the leading author has developed extensive research in the last years. Based on this main goal, the authors formulate three hypotheses that form the basis for the results and discussion. The goals of the manuscript fully fit the scope of the journal, the data is of high quality and the methodology appears elaborate and generally appropriate. The authors produce a very large number of results, which may even be enough material for two papers, and I think this is where the manuscript becomes difficult to navigate. Although the overarching questions are clearly outlined in the introduction, the focus of the paper gets somewhat lost across the numerous results and analyses. For example, I find the water age analysis too detailed and beyond the main point of the article. I believe the focus of the manuscript should be narrowed and the material more selected to provide a cleaner storyline. Overall, while I believe these results deserve publication, I think major improvements on the manuscript structure and focus are required.

*Reply:*

*We thank the reviewer for her/his interest in our work. We highly appreciate the thought- and insightful comments and the overall positive assessment of our manuscript. We will address all comments in detail below.*

Comment:

Novelty and relevance: I think the paper introduction is well developed but I somewhat miss the novelty and relevance of the results. E.g. what did we not know before? Does novelty lie specifically in the quantitative (rather than qualitative) evaluation of change?

*Reply:*

*We agree, as also pointed out to Reviewer #1, that the research objectives and in particular the novel aspects of our analysis were not defined clearly enough in the original manuscript.*

*Briefly, previous analyses in the study catchment documented changes in the individual water balance components (i.e. evaporation and discharge, Wiekenkamp et al., 2016, 2020) and also fluctuations in young water fractions (Stockinger et al., 2019) in the years after deforestation. In contrast, the aim and novelty of our study here was to establish a link between these observed changes in the water balance components and deforestation-induced changes in (sub-)surface system properties (and thus model parameters) and to explore and quantify possible mechanistic processes that cause these changes, eventually providing a possible explanation of __why__ these changes occurred.*

*Our results provide evidence that changes in $S_{U,max}$ (and to a minor degree, in $I_{max}$), a hydrologically relevant and directly quantifiable subsurface property/model parameter, can explain much of __why__ the hydrological response as well as travel times and young water fractions changed after deforestation.*

*We will clarify this in the revised manuscript.*

Comment:

Visual assessment of model results: Figures are all high-quality but they are not always informative. Figure 2 is particularly important because it shows time series of modelled and observed variables, but it is rather ineffective and it should be redesigned.

*Reply:*

*We agree that Figure 2 can be improved. We will adapt as described in detail below.*

Comment:

Model purpose: the model produces a large number of outputs and I appreciate that the authors clearly discuss the limitation of the modelling framework. I think it would be good to declare upfront the capabilities of such a model. For example, given its design and the data used to assess its performance, what can the model be reasonably used for in this context?

*Reply:*

*We agree. We will provide some more information on that in the model description section. Briefly, the model used is a current-generation catchment scale, process-based, semi-distributed model based on conceptual parametrizations and which is coupled with a tracer tracking module formulated based on the SAS-function concept. As such it is valuable to quantify catchment-scale water balance and tracer dynamics. In particular and most importantly, instead of merely aggregating largely unknown surface and subsurface heterogeneities, it allows to account for their integrated effects on the hydrological response. The main limitation of such a model approach is missing spatial detail and the use of catchment-scale effective parameter that sometimes cannot be well constrained by the model calibration process.*

Comment:

Also it seems that the model would in principle be able to estimate the age of evaporative fluxes, but the authors do not show such results. Why? And why is the model not used to estimate the (change in) partitioning between Q and ET?

*Reply:*

*The reviewer is right in assuming that the model also provides age estimates of evaporative fluxes (and any other system internal flux). In a deliberate decision we chose not to use these model outputs in the analysis as these outputs do not have any direct data support and are thus subject to considerable uncertainty. While stream water age estimates are of course also subject to uncertainties, they are more reliably and directly constrained by stream water tracer samples. We will clarify this in the revised manuscript.*

*The model is in fact already used to estimate the partitioning, although not directly between Q and ET, but instead (in fact containing the equivalent information to $Q/E_A$) expressed as the runoff ratio $C_R$ (i.e. Q/P or $(1-E_A)/P$). The runoff ratio is also one of the calibration objectives in our model implementation (i.e. $E_{R,CR}$) as shown in Table 3 and Figure 5 and discussed at several points in the original manuscript (e.g. l.454-456 or l.488-490).*

*In addition, we believe that the analysis is already quite comprehensive and the manuscript rather long, as also remarked on above by the reviewer. Additional information will make it difficult to keep the manuscript focussed.*

Comment:

Estimates of SU,max: the reduction in SU,max after deforestation is very pronounced. I'd appreciate some discussion of how a 21% reduction of forest cover may lead to a 50% reduction in SU,max. Does the location (riparian area VS hillslope) play a role? Would you expect important differences if the deforestation had occurred away from the river network? Could this model be used to make such a prediction?

*Reply:*

*This is a very interesting observation. We are similarly surprised by the strong effect and also remarked on that in the original manuscript (l.421). One statement we can make here, and which is supported by data, is that the model results suggest that the reduction of $S_{U,max}$ in the completely clear-cut riparian zone was considerably higher than on the only partially deforested hillslopes (see Table 2, Figure 6 and l.509-511). While this indicative ranking and also the strong reduction of $S_{U,max}$ in the clear-cut riparian zone are plausible (~ 130 mm; l.511), the ~ 75 mm reduction in the only partially deforested hillslope zone (l.510) indeed generates questions. Of course, it cannot be excluded that deforestation affects other properties in the subsurface and that these changes may feed back onto $S_{U,max}$ (as discussed in l.503-505 in the original manuscript). An alternative explanation could be that after deforestation in 2013, the remaining forest on the hillslopes, that was homogeneously planted in 1946 (Graf et al., 2014) was thinned in 2015, thereby likely reducing catchment-scale vegetation water demand and thus also $S_{U,max}$, as compared to the pre-deforestation period. We, however and unfortunately, do at this point not have the necessary data to further test and substantiate this hypothesis.*

*We will extend the discussion on this in the revised version of the manuscript.*

Comment:

Age analysis: as mentioned before, I think this is a very complete analysis but it seems to go too deep compared to the scope of the paper. There are several concepts and results that appear complex (e.g. not just the TTD/RTD ratio or the young water fraction, but also their sensitivities to wetness, before and after deforestation) and would fit a paper that specifically focuses on the effects of deforestation on catchment residence times. But given the broader objectives, a selection of the results might help clarify the message.

*Reply:*

*We agree, that a comprehensive selection of results is presented and that the manuscript may benefit from a bit less detail. We will therefore remove the results/discussion on the differences between TTD and RTD and try to shorten some of the other aspects.*

Comment:

Title: isn't it obvious that deforestation "affects" travel times? Can suggest how it affects them

*Reply:*

*Probably it is indeed intuitively obvious. However, to our knowledge it has not yet been explicitly shown in a detailed analysis. To highlight the direction of the effect, we suggest to adapt the title to: "Reduction of vegetation-accessible water storage capacity after deforestation affects travel time distributions and __increases__ young water fractions in a headwater catchment."*

Comment:

50-56: very long sentence. Consider breaking it

*Reply:*

*Agreed. Sentence will be rephrased.*

Comment:

70-71: unclear sentence

*Reply:*

*Agreed. Sentence will be rephrased.*

Comment:

93-96: this sentence includes some slang. Consider rephrasing. Also, it partly seems a repetition of lines 49-52

*Reply:*

*We are not sure what the reviewer is exactly referring to as "slang" in this sentence: "Transpiration extracting soil water below that therefore effectively generates a root-zone water storage reservoir between field capacity and permanent wilting point that is characterized by a storage capacity $S_{U,max}$, i.e. a maximum vegetation-accessible storage volume, and that is at any given moment filled with a specific water volume $S_U(t)$, depending on the past sequence of water inflow and release."*

*It is true that the sentence is partly a repetition, but we believe it is necessary to explicitly clarify the difference between $S_{U,max}$ and $S_U(t)$. We will try to rephrase the sentence.*

Comment:

98: "With increasing storage, the hydrological memory of a system increases" → can be easily misinterpreted because one may think that, for a system with a given water storage, streamflow age would increase when the storage increases, but it is usually found (e.g. Harman, 2015, Water Resources Research) that the opposite applies.

*Reply:*

*This is indeed a delicate problem, mainly due to the unfortunate and confusing terminology available to us. In fact the term "storage" can be used in ways that have subtle differences. Consider the following simplified, illustrative thought experiment, distinguishing 4 situations:*

*(1) a small storage reservoir that is half-filled with water and from which water can only slowly drain at the bottom. Water will stay and mix in this volume for some time (i.e. little storage leads to old ages that eventually leave the reservoir) before being drained.*

*(2) In a second step, this storage reservoir is filled and overtops. Much of the overtopping water has little opportunity to mix with older water in the reservoir and bypasses the storage volume (i.e. at such high storage states, much of the outflowing water will thus be very young)*

*(3) a second storage reservoir that is __larger__ than the one above is, just as above in (1), half-filled with water and from which water can only slowly drain at the bottom. Water will stay and mix in this volume for some time (i.e. little storage leads to old ages that eventually leave the reservoir). However, the difference to (1) is that although the degree of filling of the storage reservoir is the same, the actual storage volume is larger, which leads – under complete mixing to older ages than in (1), as reflected in basic equation for mean turnover times T=volume/flux.*

*(4) In a last step, this larger storage reservoir is filled and overtops. Much of the overtopping water has little opportunity to mix with older water in the reservoir and bypasses the storage volume (i.e. at such high storage states, much of the outflowing water will thus be very young). However, under most conditions there is still __some__ exchange with the old water in the large storage reservoir. Although younger than in a half-filled large storage volume (3), the outflowing water may still be older than the outflowing water from the full small storage volume (2).*

*In summary, while the use of the term storage in the Harman (2015) paper refers to the degree-of-filling of a storage reservoir with given size, i.e. (1) and (2), we use the term storage here to refer to the absolute size of a storage reservoir, i.e. (3) and (4).*

*We will clarify this in the revised manuscript.*

Comment:

Figure 2: this figure is very important for the manuscript but I find it rather inefficient. Subplot (a) does not seem to show anything that a reader could grasp. Subplot (c) is very compressed –it is never easy to compress 7 years of data into a single pane l– but then it is almost impossible to inspect the results. Subplot (d) is uninformative because tracer in stream water is not shown at an adequate scale. Is tracer precipitation data really useful here? If you really want to show it, maybe you could report some monthly means (so we get at least the seasonality of the input) and on a rescaled y-axis?

*Reply:*

*We agree that efficiently showing the most relevant results over longer time periods for multiple variables in figures is challenging. We also agree that Figure 2 can benefit from some adaptions. We therefore propose the following changes.*

*Subplot (a): we will replace daily values by monthly values. This should make the plot easier to read.*

*Subplot (c): we think it is important to give the reader an overall impression of the hydrological response to be able to place it into sufficient context. We thus would like to keep subplot (c) as is, but to add either (1) an additional subplot, showing zoomed-in version of one individual year pre- and one year post-deforestation, or (2) detailed individual plots for each year in figures in the Supplementary Material. We will test and evaluate both alternatives.*

*Subplot (d): We had a long and intensive discussion on exactly this question and, in the end, made the deliberate decision in the original manuscript to show the tracer signals in the way we did. Of course the reviewer is right in saying that detailed, small scale fluctuations in stream water tracer cannot be*

*seen in this figure. The underlying question for us was, which information is in fact the most relevant to convey here. We finally decided that it is more relevant to illustrate the degree of damping that precipitation tracer signals experience before they reach the stream. This is a direct indicator of the age of stream water and can only be seen when both variables are plotted at the same scale. In the case of this study catchment, when the tracer signal is attenuated to a degree that the stream water composition plots almost as a straight line, little fluctuations around this line are difficult to interpret: how much of it is due to real effects? How much is due to mere noise, resulting from observational uncertainties? This is also discussed in some detail in the original manuscript (l.474-479). However, we also acknowledge that the reader may want to see exactly this detail. Thus we propose the same as for subplot (c) and we will add an additional subplot with a zoomed-in version for one individual year pre- and one year post-deforestation. In addition, we will follow the reviewer's excellent suggestion to integrate the precipitation signals to monthly time-scales for better readability of the figure.*

Comment:

162: "To quantify effects of deforestation on SU,max and, as a consequence of that, on the age structure of water" I find this slightly misleading as it seems that the age structure is only affected through SU,max (and not directly).

*Reply:*

*Deforestation reduces the catchment-scale pore volume that can be accessed by roots, i.e. $S_{U,max}$. This also reduces $S_{U,max}$ as mixing volume. The size of a mixing volume directly affects water ages (see replies above). Changes in water age structure are thus a consequence of a reduction of this mixing volume. We will clarify this in the revised manuscript.*

Comment:

166: similar to 162, I don't see why you estimate the effects of these changes only. Aren't the two problems partially separate? You can get to (2) even without (1) or am uncorrect?

*Reply:*

*It is correct that quantifying changes in $S_{U,max}$ does not **require** an a priori check if the general water balance partitioning pattern changed. However, under relatively stable climatic conditions (as in the study catchment during the study period), changes in the partitioning are a reliable indicator of changes in some (a priori unknown and unquantified) system properties (and thus model parameters), frequently related to changes in vegetation cover (e.g. van der Velde et al., 2014; Jaramillo et al., 2018). If, in contrast, no such changes in the partitioning are observed, changes to system properties are much less likely.*

*We therefore wanted to first establish that there actually was a significant change between the pre- and post-deforestation partitioning pattern detectable with our method (and not only with the method used in Wiekenkamp et al., 2016) and to use this as basis for the subsequent hypothesis that deforestation affects $S_{U,max}$ and that this effect can be quantified.*

Comment:

Eq 39 and 41: what is the horizontal bar above the lowercase omega?

*Reply:*

*We agree, this looks confusing. This is not an overbar – the italic style of this MS Word font, when used in the equation editor only appears to have an overbar. It is in fact only a normal omega character.*

Comment:

336: this explains how you translate a physical process into a model component. But I recommend explaining why a "passive" storage is needed. It is not a model artefact, but rather the real amount of water that is involved in the transport process. Other good classic references for this is "Birkel, C., C. Soulsby, and D. Tetzlaff (2011), Modelling catchment-scale water storage dynamics: Reconciling dynamic storage with tracer- inferred passive storage, Hydrol. Processes, 25(25), 3924–3936, doi:10.1002/hyp.8201"

*Reply:*

*This is indeed an excellent suggestion. We will make this more explicit and add the above reference.*

Comment:

353: Ok fine, but how do you initialize the model? Which d18O initial composition is assigned to the different compartments?

*Reply:*

*We initialized the model using different initial isotope compositions for the individual storage components: assuming that only groundwater flows from $S_S$ sustain low flows during the dry season, we used the mean isotope stream water composition of the time steps with the 5% lowest flows as initial composition in $S_S$ and $S_{S,p}$. For $S_U$ we used the long-term volume weighted mean precipitation value of the months April-September, preceding the start of the model in October. $S_I$ and $S_F$ were, due to their small sizes, assumed to be empty. The model was then primed with a 5-year warm-up period before 10/2009, which was a copy of the data from actually available observations.*

*We will add this information in the revised manuscript.*

Comment:

385: I suggest replacing "the data" with "results"

*Reply:*

*Agreed.*

Comment:

408: start with "In our study, "

*Reply:*

*Agreed.*

Comment:

417: SU,max = 90+-149? If the standard deviation is much larger than the mean, wouldn't it be better to avoid completely this exercise? Or maybe mention it here but not stress it later as a real result, given it comes with considerable uncertainty?

*Reply:*

*We agree that these uncertainties are quite high. We still believe it is worth to report it, to give the reader a sense of a, albeit uncertain, plausible range. Due to the uncertainties, we do, on purpose, not use these values for any quantitative analysis. We have explicitly stressed and acknowledged that in the original manuscript (l.418-420). We will further clarify this in the revised manuscript.*

Comment:

428 and 435: as noted in comments on Figure 2, it is difficult to judge from this figure because too much data is compressed in it and the scaling does not look appropriate

*Reply:*

*We will adapt Figure 2 as described in our reply to one of the comments further above.*

Comment:

Figure 5: please add some legend to make the colouring more intuitive

*Reply:*

*Agreed.*

Comment:

Figure 6: why showing cumulative frequencies? I find it difficult to evaluate how much parameters are constrained from these curves.

*Reply:*

*Ok. We will add histograms to also show the associated frequency distributions in these plots.*

Comment:

440: "...show that most model parameters are reasonably well identified" but it is difficult to actually evaluate

*Reply:*

*We will adapt Figure 6 as described above to make it easier to see.*

Comment:

494: if I haven't stressed this enough, this is invisible in the figure

*Reply:*

*Agreed. Figure 2 will be adapted as described above.*

Comment:

513: I do not see how a value of 120 falls "reasonably well" into a "plausible" range [-59, 239]

*Reply:*

*We are not sure what the reviewer exactly wants to express here. In our understanding, a value of 120 is very close to the central value (i.e. 149) of the interval spanned by [-59, 239] and thus also falls into this range.*

Comment:

519: I get the point, but cannot really evaluate overlapping from figure 6. Rather from table 2. Then I see that Lp is the only other parameter with a significant change pre-post deforestation. Might this be worth of a comment?

*Reply:*

*We will adapt Figure 6 as described above. It is true the parameter $L_P$ does change, but, and this seems to be a misunderstanding, it is by far __not__ the only parameter with a significant change as shown in Table 2 in the original manuscript.*

*The parameters that are directly related to vegetation ($S_{U,max}$, $I_{max}$) change the most and they do so in a more pronounced way in the fully deforested riparian zone, i.e. the range of $S_{Umax,R}$ is reduced from 194-287 mm to a range of 53-122 mm, while $I_{max,R}$ is reduced from 0.8-4.5 mm to 0.0-0.9 mm.*

*The reductions in the partially deforested hillslope parameters are less pronounced but still statistically significant ($p<0.05$) for $S_{U,max,H}$ from 233-309 mm to 118-249 mm and for $I_{maxh,H}$ from 0.8-4.5 to 0.1-1.7 mm.*

*Most other posterior parameter distributions (i.e. parameters not directly related to vegetation) do not experience a significant change (Table 2).*

Comment:

534: [again on figures] "is evident", but nothing is evident from figure 2d

*Reply:*

*Agreed. Figure 2 will be adapted as described above.*

Comment:

541-566: this is a nice analysis but I wonder whether it is really necessary as it is more about general TTD and RTD dynamics rather than on effects of deforestation. In other words, this seems to go beyond the scope of the manuscript.  A large simplification would make the manuscript easier to read.

*Reply:*

*We will try to shorten this section. In particular, we will remove the analysis and interpretation of RTDs. However, we would really like to keep the mechanistic interpretation of the results, which is ultimately the part of the objectives of this paper.*

Comment:

544: please specify how you computed Fyw because a reader will expect it is estimated using kirchner's 2016 method.

*Reply:*

*The young water fraction was here defined as the fraction of water younger than three months. This value was here directly extracted from the associated travel time distributions (TTDs). We will clarify this in the revised manuscript.*

Comment:

33: the change in Fyw from 0.11 to 0.13 does not seem significant considering the possible uncertainties in the estimate.

*Reply:*

*This is correct. We will clarify this in the revised manuscript.*

Comment:

Figure 2d: typo in the permil symbol in the y-axis label.

*Reply:*

*Ok.*

Comment:

70: the vegetation...presentS

*Reply:*

*This is a misunderstanding. The sentence should read as: "The vegetation, i.e. a collective of individual different plants within an area of interest that is present at any given moment at any given location, has survived."*

Comment:

264: it is reaches

*Reply:*

*Will be corrected.*

Comment:

548: "Stream water can contain Fyw" is not a correct formulation. Rather "stream water can contain up to 30% of young water" or similar

*Reply:*

*Will be corrected.*

Comment:

532-534: please reformulate this initial sentence, which is unclear The terminology"partly much lower" (452) and "partly considerable" (572) is a bit odd

*Reply:*

*Will be corrected.*

Comment:

425: title: Deforestation effects "on the catchment model" sounds a bit odd

*Reply:*

*Will be rephrased.*

References:

Graf, A., Bogena, H.R., Drüe, C., Hardelauf, H., Pütz, T., Heinemann, G., & Vereecken, H. (2014). Spatiotemporal relations between water budget components and soil water content in a forested tributary catchment. Water resources research, 50(6), 4837-4857.

Jaramillo, F., Cory, N., Arheimer, B., Laudon, H., Van Der Velde, Y., Hasper, T. B., ... & Uddling, J. (2018). Dominant effect of increasing forest biomass on evapotranspiration: interpretations of movement in Budyko space. Hydrology and Earth System Sciences, 22(1), 567-580.

Stockinger, M. P., Bogena, H. R., Lücke, A., Stumpp, C., & Vereecken, H. (2019). Time variability and uncertainty in the fraction of young water in a small headwater catchment. Hydrology & Earth System Sciences, 23(10).

Van der Velde, Y., Vercauteren, N., Jaramillo, F., Dekker, S. C., Destouni, G., & Lyon, S. W. (2014). Exploring hydroclimatic change disparity via the Budyko framework. Hydrological Processes, 28(13), 4110-4118.

Wiekenkamp, I., Huisman, J. A., Bogena, H. R., Lin, H. S., & Vereecken, H. (2016). Spatial and temporal occurrence of preferential flow in a forested headwater catchment. Journal of hydrology, 534.

Wiekenkamp, I., J.A. Huisman, H.R. Bogena, and H. Vereecken (2020): Spatiotemporal Changes in Sequential and Preferential Flow Occurrence after Partial Deforestation. Water 12(1), 35.

---

## Author Comment (AC3) · 8 Oct 2020

Comment:

This manuscript presents a study on the effect of deforestation on catchment hydrology. In this manuscript mainly a modeling approach is used. Using this modeling approach, the authors find that runoff increases after deforestation and also catchment travel time distributions. While this is an interesting point, I have a number of concerns with the study as it is presented now.

*Reply:*

*We appreciate the reviewer's interest in our work and her/his thoughtful comments. We provide detailed clarifications in the list below.*

Comment:

While the study uses a valuable data set is not fully clear to me how exactly this paper goes beyond the studies that have already been published using this catchment and its data set. Conclusions 1 and 2 basically confirm earlier studies, and the other conclusions are based on modeling with a number of assumptions (as discussed be-low). In general, it would be important that the authors relate their findings more to the previous findings using the same catchment to show the added value of this study clearly.

*Reply:*

*We agree with the reviewer that the objective and novelty of our study was not communicated in a clear enough way in the original manuscript.*

*Briefly, previous analyses in the study catchment documented changes in the individual water balance components (e.g. evaporation and discharge, Wiekenkamp et al., 2016) and also some minor fluctuations in young water fractions (e.g. Stockinger et al., 2019) in the years after deforestation. Yet, these contributions did not provide quantitative mechanistic explanations of **why** these changes occurred.*

*The aim and novelty of our study is therefore to explore and quantify a possible mechanistic process that causes these changes. Our results provide some evidence that changes in $S_{U,max}$, a hydrologically relevant and directly quantifiable subsurface property/model parameter, can explain much of **why** the hydrological response as well as travel times and young water fractions changed after deforestation.*

*We will make this clearer in the revised version and we will also more explicitly discuss our results with respect to the results of the above studies.*

*It is also true that Conclusion #1 in the original manuscript essentially only supports the results of a previous study (i.e. Wiekenkamp et al., 2016), albeit estimated with a different method and over a different time period. We will clarify this in the revised manuscript.*

*In contrast, we are not aware of any study in the Wüstebach that quantifies the effect of deforestation on $S_{U,max}$, i.e. Conclusion #2. Our results, however, provide a puzzle-piece of supporting evidence for the potential generality of $S_{U,max}$ reductions due to deforestation as they are consistent with results from catchments in different climatic and geomorphic settings (Nijzink et al., 2016; cited in the original manuscript).*

Comment:

An obvious limitation of this study is in the use of only one catchment. As valuable as the data set is results might not be generally valid unless they can be confirmed using a larger data set including several catchments.

*Reply:*

*We completely agree. However and unfortunately, we are not aware of any other catchment world-wide, where the necessary data are available and the necessary conditions for such a study are met.*

*Briefly, which minimum information is required to do such an analysis?*

*At least a few years, pre- and deforestation respectively, with (1) daily water balance observations (P, $E_P$, Q; which are without doubt available from a many catchments world-wide), (2) (bi-)weekly tracer composition in precipitation and stream water and (3) well **documented deforestation** of a **significant fraction** of a catchment and that occurred during a sharply defined, **short period**, which falls **within the time period of available water balance and tracer data**.*

*For example, while for the Plynlimon experimental catchment, long time series of water balance and tracer observations are available, deforestation occurred only gradually over many years (or decades), each time only affecting a small fraction of the catchment, and was partly offset by regrowth. In addition, much of the deforestation did temporally not overlap with the availability of tracer data. In another example, for the HJ Andrews experimental watershed, where well documented, significant deforestation took place over short, well-defined time periods, no tracer data are available for that period. Similar limitations apply to other locations, making such an analysis highly problematic.*

*As much as we wanted to provide a more generally valid analysis, we remain limited by the available data. We nevertheless strongly believe that in such a case, detailed analyses and process understanding developed from anecdotal case studies, such as this one, can be valuable to shape our understanding of what could happen and which processes occur – without claiming generality of the phenomenon.*

*We also agree that the title and the framing of the original manuscript may have misled the reader into hoping for a more general analysis. We will therefore adapt the title to "Reduction of vegetation-accessible water storage capacity after deforestation affects travel time distributions and increases young water fractions in a headwater catchment." to emphasize the local character of this study. In addition, we will make this clearer in the introduction.*

Comment:

The most obvious effect of deforestation on catchment hydrology obviously is the re-moved interception. Here the authors largely ignore interception by the use of so-called effective precipitation. It is important to note that effective precipitation was determined by modeling using a very simple approach with a constant interception storage. This means that the results might have been implicitly affected by the calculation of the effective precipitation.

*Reply:*

*While we agree with the reviewer that interception plays a role in forest systems, we respectfully disagree with the statement that the "most obvious effects" of deforestation relate to interception. Leaving aside the fact that, of course, visually the effect is large, many studies world-wide demonstrate*

*that transpiration not only exceeds interception evaporation (e.g. Jasechko et al., 2013; Coenders-Gerrits et al., 2014; Schlesinger et al., 2014; Wei et al., 2017; Mianabadi et al., 2019), but that transpiration is actually the largest water flux from many terrestrial systems (Jasechko, 2018).*

*The reviewer's notion that we "largely ignore interception" is probably a misunderstanding as we use a standard technique, successfully tested and used in current-generation catchment-scale model formulations to account for interception (e.g. Fenicia et al., 2008; Samaniego et al., 2010; Gao et al., 2014; Nijzink et al., 2016). It is true that we use an interception capacity $I_{max}$ which defines the __maximum__ volume of water that can be stored in the interception storage reservoir $S_I(t)$ at each time step t before overflowing, as defined by equations 9, 17, 18 and 19 in the original manuscript. At each time step t, this reservoir fills (via precipitation P(t)), drains (via evaporation $E_I(t)$ and overflow $P_E(t)$) and stores water (whatever water volume does not overflow as $P_E$ and cannot be evaporated as $E_I$ within a time step is carried over as storage $S_I(t)$ to the next time step).*

*It is thus important to note that __not the interception storage $S_I(t)$__, i.e. the intercepted volume of water at each time step, is constant, __but the maximum volume of water $I_{max}$__ that can be stored in the reservoir at each time step is constant. In other words, the degree of filling of the interception reservoir varies each day up to a maximum volume of $I_{max}$.*

*The modelled fluxes related to interception evaporation are broadly consistent with plots-scale observations thereof: the model suggests a mean ratio $P/P_E \sim 1.35$ (5/95th interval: $1.19 - 1.51$) for the growing season, while Stockinger et al. (2015) reported an observed $P/P_E \sim 1.41$ (± 0.19).*

*The underlying problem that prevents a more detailed formulation of this process is that, on larger scales, such as catchments, we have by far insufficient data (even in experimental catchment such as the Wüstebach) to warrant a meaningful, more detailed formulation of the interception process (and many other processes), as also described with an illustrative example in Hrachowitz and Clark (2017; see section S2 of the Supplementary Material therein).*

*The limited influence of interception evaporation on $S_{U,max}$ and the associated interpretations are explicitly described in lines 411-415 and shown in Figures 4b and 4c in the original manuscript.*

Comment:

The isotope data is used to parameterize the passive storage volume. As I understand, this passive volume is only used for groundwater storage. This makes me wonder whether any passive storage is also being considered for the unsaturated storage. Sorry in case I missed this, but I am feeling a bit confused here. It is also not clear to me how mixing between active and passive storage has been calculated.

*Reply:*

*We do not consider an explicit passive storage volume for the unsaturated zone. In reality, there may be such hydrologically passive volumes (i.e. dS/dt = 0) in the unsaturated zone, though.*

*The first one is the water stored in pores at water content below the permanent wilting point and thus essentially bound indefinitely, as it cannot be drained nor extracted by vegetation. The absolute magnitude of this volume is, depending on the wilting point, very small and thus considered to be negligible here and in similar studies (e.g. Birkel et al., 2010; Harman, 2015; Benettin et al., 2017 and many others).*

*The second storage compartment that could be seen as a hydrologically passive storage volume is the unsaturated zone **below** and/or **outside** the root zone. In that zone, vegetation cannot extract significant volumes of water below field capacity. The pores therefore remain filled to field capacity much of the year, except for moments when a wetting front passes through. However, exchange (i.e. "mixing") with water in the "active" stores can and does occur. Indeed, we have previously added such a hydrologically passive mixing volume, representing the transition zone between the root zone and the groundwater, in a similar study (Hrachowitz et al., 2015). However, we found that the additional parameters cannot be reliably constrained with the available data, thereby increasing model uncertainty.*

*In the present study, we also tested initial model formulations including such a store. However, it was found that it does not improve the results nor that it is meaningfully constrained by the available data. Here in the Wüstebach, the passive mixing volume in the unsaturated zone has an upper plausible limit of <1000 mm due to the position of the groundwater table (see also replies to Reviewer #1). In the way the model is implemented now, this additional mixing volume is implicit in $S_{S,p}$. In other words, were we to introduce an individual passive unsaturated mixing volume of ~1000 mm, $S_{S,p}$ would be reduced by approximately the same value. Such a trade-off, however, does not influence any of the results, as water going through such an unsaturated passive volume, subsequently only passes through $S_S$. Thus, the two can be aggregated without introducing relevant uncertainties. The same implicit reasoning was applied in many successful previous tracer studies (e.g. Birkel et al., 2010; Harman, 2015; Benettin et al., 2017 and many others).*

*The mixing between active and passive stores was done using a standard SAS-function: a technique that has been well described and successfully tested in a large body of literature cited in the original manuscript (e.g. Botter et al., 2011; van der Velde, 2010; Benettin et al., 2013, 2017; Harman, 2015; Rinaldo et al., 2015 and many others), but also reproduced in this manuscript in some detail (section 4.2.2 and in particular l.336-339). Briefly, the estimated outflow $Q_S(t)$ from the active groundwater store $S_{S,a}(t)$ at each time step t is a composition of ages that is sampled (i.e. "mixed") according to the SAS-function from the **total** groundwater storage, i.e. $S_{S,T}(t)=S_{S,a}(t)+S_{S,p}$. We will further clarify this in the revised version of the manuscript.*

Comment:

Another crucial issue is the assumption that roots only take from the unsaturated zone. While this might be the case for the direct uptake, indirectly the uptake of water from the unsaturated zone will have the effect that an upward gradient is established which will cause groundwater to rise into the unsaturated zone. So, indirectly roots can access water from the saturated zone. This process seems to be ignored here.

*Reply:*

*We agree that roots can create suction pressures that, together with pure capillary rise and upwelling groundwater, can result in upward fluxes. While the latter is explicitly accounted for in the model as flux $R_{S,R}$ (Figure 3; Equations 11, 13 and 24; l.261-262), the first is implicit in how $S_{U,max}$ is estimated: it is the volume of water that was in the past accessible and accessed by vegetation. It therefore includes any upward capillary water fluxes.*

Comment:

Based on the points above, I would argue that the calibration parameter S_Umax is difficult to interpret. It sounds like it is the size of the unsaturated storage, but I would argue it is more a parameter describing catchment functioning.

*Reply:*

*As clarified above (as well as explained in the original manuscript and the references therein), $S_{U,max}$ has indeed a clear and hydrologically meaningful definition. It is the maximum water volume that can be stored in the pores of the unsaturated root zone between field capacity and permanent wilting point at the catchment scale and, most importantly, which is __accessible and accessed by roots__ (sometimes also referred to as "available water" or "plant available water"). This vegetation accessible water volume at the catchment scale is therefore a meaningful measure for the catchment-scale effective influence zone of roots, implicitly accounting for spatial heterogeneity of soils as well as for the spatial heterogeneity in the depths and densities of root systems. It is also a subsurface property that regulates much of the hydrological response and which could be, provided we have suitable observation technology available at some time in the future, practically obtained by measuring __all__ roots and __all__ soil porosities in a catchment multiple times over a certain period. As it could be __directly measured__, albeit only theoretically at this point, it __represents also a real catchment-scale quantity__ and not only some abstract number. Note, that $S_{U,max}$ does not describe the unsaturated zone outside/below the root-zone. Also note that $S_{U,max}$ is a conceptual volume (and as such it is a theoretically directly measurable quantity, as explained above) that does not necessarily match to a specific position in the soil. It is rather a catchment-integrated pore volume that aggregates all locations and depths where roots are present.*

Comment:

Overall, I am afraid that the results obtained in the study are both catchment- and model-dependent. The authors need to make a more convincing case why their results are generally valid.

*Reply:*

*We agree, the results depend on the decision taken and choices made throughout the modelling process. This is the case for any modelling study. For this reason and to limit uncertainties we only report results from processes that could at least to some degree be confronted with data. For example, we estimated $S_{U,max}$ (1) from water balance data alone and (2) as model calibration parameter and then compared these values. In spite of uncertainties, they are broadly consistent with each other. Similarly, we only analysed stream water travel time distributions and not, for example, evaporation travel time distributions, as in the model only the stream water isotopic composition could be directly confronted with and compared to observed data. Furthermore, in an attempt to make the model as robust and reliable as possible, we constrained, evaluated and tested the model with an extended multi-objective and multi-variable calibration strategy, that goes far beyond of what many other modelling studies do. Lastly, we have also devoted the entire Section 5.4 to a detailed and honest discussion of the limitations of the modelling experiment, stressing that the results and interpretations are of course model-dependent (l.599). Given the above points, we nevertheless think that the results provide a rather reliable test of our research hypotheses. However, as with any other hypothesis, the test could be even stricter if the necessary data were available (Hrachowitz and Clark, 2017).*

*As discussed in one of the replies above, as much as we would like, the available data does not allow us to generalize the result. As also mentioned above, we will therefore emphasize the local character of the results more in the revised manuscript.*

References:

Benettin, P., Soulsby, C., Birkel, C., Tetzlaff, D., Botter, G., & Rinaldo, A. (2017). Using SAS functions and high-resolution isotope data to unravel travel time distributions in headwater catchments. Water Resources Research, 53(3), 1864-1878.

Birkel, C., Tetzlaff, D., Dunn, S. M., & Soulsby, C. (2010). Towards a simple dynamic process conceptualization in rainfall–runoff models using multi-criteria calibration and tracers in temperate, upland catchments. Hydrological Processes: An International Journal, 24(3), 260-275.

Botter, G., Bertuzzo, E., & Rinaldo, A. (2011). Catchment residence and travel time distributions: The master equation. Geophysical Research Letters, 38(11).

Coenders-Gerrits, A. M. J., Van der Ent, R. J., Bogaard, T. A., Wang-Erlandsson, L., Hrachowitz, M., & Savenije, H. H. G. (2014). Uncertainties in transpiration estimates. Nature, 506(7487), E1-E2.

Fenicia, F., Savenije, H. H., Matgen, P., & Pfister, L. (2008). Understanding catchment behavior through stepwise model concept improvement. Water Resources Research, 44(1).

Harman, C. J. (2015). Time-variable transit time distributions and transport: Theory and application to storage-dependent transport of chloride in a watershed. Water Resources Research, 51(1), 1-30.

Hrachowitz, M., & Clark, M. P. (2017). HESS Opinions: The complementary merits of competing modelling philosophies in hydrology. Hydrol. Earth Syst. Sci, 21(8), 3953-3973.

Hrachowitz, M., Fovet, O., Ruiz, L., & Savenije, H. H. (2015). Transit time distributions, legacy contamination and variability in biogeochemical 1/fα scaling: how are hydrological response dynamics linked to water quality at the catchment scale?. Hydrological Processes, 29(25), 5241-5256.

Hrachowitz, M., Benettin, P., et al. (2016). Transit times—The link between hydrology and water quality at the catchment scale. Wiley Interdisciplinary Reviews: Water, 3(5), 629-657.

Jasechko, S. (2018). Plants turn on the tap. Nature Climate Change, 8(7), 562-563.

Jasechko, S., Sharp, Z. D., Gibson, J. J., Birks, S. J., Yi, Y., & Fawcett, P. J. (2013). Terrestrial water fluxes dominated by transpiration. Nature, 496(7445), 347-350.

Mianabadi, A., Coenders-Gerrits, M., Shirazi, P., Ghahraman, B., & Alizadeh, A. (2019). A global Budyko model to partition evaporation into interception and transpiration. Hydrology and Earth System Sciences, 23(12), 4983-5000.

Nijzink, R., Hutton, C., Pechlivanidis, I., Capell, R., Arheimer, B., Freer, J., ... & Hrachowitz, M. (2016). The evolution of root-zone moisture capacities after deforestation: a step towards hydrological predictions under change?. Hydrology and Earth System Sciences, 20(12), 4775-4799.

Rinaldo, A., Benettin, P., Harman, C. J., Hrachowitz, M., McGuire, K. J., Van Der Velde, Y., ... & Botter, G. (2015). Storage selection functions: A coherent framework for quantifying how catchments store and release water and solutes. Water Resources Research, 51(6), 4840-4847.

Samaniego, L., Kumar, R., & Attinger, S. (2010). Multiscale parameter regionalization of a grid-based hydrologic model at the mesoscale. Water Resources Research, 46(5).

Schlesinger, W. H., & Jasechko, S. (2014). Transpiration in the global water cycle. Agricultural and Forest Meteorology, 189, 115-117.

Stockinger, M. P., Lücke, A., McDonnell, J. J., Diekkrüger, B., Vereecken, H., & Bogena, H. R. (2015). Interception effects on stable isotope driven streamwater transit time estimates. Geophysical research letters, 42(13), 5299-5308.

Van Der Velde, Y., Torfs, P., Van Der Zee, S., & Uijlenhoet, R. (2012). Quantifying catchment-scale mixing and its effect on time-varying travel time distributions. Water Resources Research, 48(6).

Wei, Z., Yoshimura, K., Wang, L., Miralles, D., Jasechko, S., & Lee, X. (2017). Revisiting the contribution of transpiration to global evapotranspiration. Geophysical Research Letters, 44(6), 2792-2801.

---

## Author Comment (AC4) · 8 Oct 2020

Comment:

This manuscript presents the effects of partial deforestation on water storage and water ages in the German Wüstebach catchment. For this study, the authors performed water balance analyses and modelling exercises based on 7 years of hydrometric and water stable isotope data. One major finding of the study is that the vegetation-accessible storage volume in the unsaturated zone, SUmax*, was significantly reduced after the partial deforestation; the authors hypothesize that this reduction in *SUmax* can largely be explained with young water being routed quickly to the stream during wet conditions, so that less water reached the unsaturated zone *SU*.

The paper is well written and the figures are informative. I only have some minor comments and questions that the authors should address.

*Reply:*

*We highly appreciate the reviewer's positive assessment of our manuscript and thank her/him for the thoughtful detailed comments.*

Comment:

The physical meaning of *SUmax* not fully clear to me: its definition in the introduction is "water-filled pore volume between field capacity and permanent wilting point that is within the reach of active roots". This suggests that *SUmax* depends on water content in the soil and the active rooting depth. Does this mean that *SUmax* will decrease when water influx is reduced and/or roots become shorter? Then, the major result of the study (i.e., *SUmax* is reduced after deforestation; L421-424) is not surprising but rather expected because fewer roots will lead to a smaller catchment-average active rooting depth.

*Reply:*

*We believe that our addition "water-filled" made the definition unclear. $S_{U,max}$ is the "pore volume between field capacity and permanent wilting point that is within the reach of active roots". As such it describes the __maximum__ (i.e. the upper bound, "capacity") possible water volume that can be __held__ against gravity (i.e. above field capacity) __and__ that can be __accessed__ by vegetation.*

*Indeed, $S_{U,max}$ is completely independent of the actual water content in the soil at any time. For practical purposes, it is can be considered constant over short time-scales < 2-3 years (although of course in reality it is continuously changing and adapting, albeit at mostly very low rates). In more humid climates, $S_{U,max}$ is typically smaller than elsewhere (e.g. Gao et al., 2014) – when there is constant water supply, e.g. when it is frequently raining, these frequent rains sustain rather near-surface soil water contents for much of the year. Vegetation therefore does not need to develop an extensive root-system to be able to access sufficient water. The opposite is true e.g. for more arid environments. In other words, when the water influx is reduced, $S_{U,max}$ will need to increase if vegetation wants to survive. Vegetation does so by developing more extensive root systems (either by individual plants growing deeper/denser roots or by specific, not sufficiently adapted plants dying and being replaced by more adapted ones). Indeed, the major result of this study is not surprising – fewer roots lead to a smaller catchment-scale root-accessible pore space $S_{U,max}$.*

*As actual root observations are very scarce in space and time, the critical questions for hydrology are: __(1) how large is this root-accessible pore space at the catchment scale__ and __(2) how much does it change after deforestation?__*

*We showed that $S_{U,max}$ can not only be quantified at the catchments-scale (as a few previous studies also suggested, e.g. Gentine et al., 2012; Gao et al., 2014) but that also its post-deforestation change can be quantified and that this change is a plausible explanation for younger water ages during wet conditions.*

*We will clarify the definition of $S_{U,max}$ in the revised manuscript.*

Comment:

L135: How many measurements of rooting depth are available to justify the general assumption that the maximum rooting depth across the catchment is 50cm? What is the depth of the groundwater table and is it possible that capillary rise from the groundwater supplies these shallow-rooted plants?

*Reply:*

*There is only anecdotal and indicative information about root-systems in the study catchment. However, there is no indication of systematic and wide-spread presence of deeper roots.*

*In the riparian zone, the groundwater table can reach the surface for a few days during the wet winter months, when transpiration is very low and which is therefore largely irrelevant for the estimation of $S_{U,max}$. During the growing season, which is critical for $S_{U,max}$, the groundwater table remains well below 1 m most of the time in the riparian zone, as shown by Bogena et al. (2015) and it can be expected to be considerably deeper on the hillslopes.*

*It is indeed possible and likely that groundwater sustains soil moisture levels and thus indirectly supplies vegetation. This is explicitly accounted for in our model as flux $R_{S,R}$ (Figure 3; Equations 11, 13 and 24; l.261-262).*

*We will clarify this in the revised manuscript.*

Comment:

Are there any additional data that support your claim of a large groundwater storage in the Wüstebach catchment? It is surprising to me that no groundwater table and soil moisture observations have been considered for explaining many of the processes you propose.

*Reply:*

*We agree that such a large mixing volume is indeed surprising and apart from the tracer observations we do not have direct evidence for the underlying reasons. However, we discuss potential explanations for that in the original manuscript. Although a "large groundwater storage" **can** be the cause, it **does not necessarily have to be**. One alternative hypothesis is that old groundwater may enter the system from outside the defined catchment and replace (i.e. push out) younger water as unobserved groundwater export (l.606-621).*

*In case there is no such groundwater exchange, something needs to buffer the high precipitation variability to the much dampened pattern observed in the stream tracer compositions, which do exhibit almost no fluctuations that go beyond measurement uncertainty. Following our current, rather well-developed understanding of tracer hydrology, such an effect can almost exclusively be caused by a large water storage volume that allows for sufficient "mixing" (e.g. Maloszewksi and Zuber, 1982; McGuire and McDonnell, 2006).*

*Based on our reply to the previous comment and thus assuming a conservative upper bound of catchment-average depth of the groundwater table at ~ 5 m (assuming that the lowest groundwater table at each point in the catchment is at the elevation of the nearest stream), porosity of the silty clay loam (Bogena et al., 2018) soil of 0.4 and field capacity at a relative pore water content of 0.5 suggests an upper storage limit ~ 1000 mm in the unsaturated zone. As no further significant storage volumes besides the unsaturated zone are known and in case there is no significant exchange of groundwater (see above) in the study catchment, only the groundwater remains as the required storage volume.*

*To answer the second part of the above comment, it is true that in the study catchment a lot of data is available. However, these data are largely inferred from point-scale measurements. Incorporating this information in catchment-scale models is challenging if not impossible to do in a meaningful way. For example, while our root-zone reservoir $S_U$ represents the catchment-average water content in the root-zone, which soil moisture observations should this be compared to? The ones in 5 cm depth? In 10 cm? 50 cm? Or an average of that? What if there are in reality no roots at that depth at a given location? To avoid introducing a further level of assumptions, we therefore did not make use of point-scale measurements in our study.*

*We will clarify that in the revised manuscript.*

Comment:

How dry, drying, wet and wetting-up periods were defined (L545, L561, Fig. 8)?

*Reply:*

*We imposed four thresholds to classify time steps along a spectrum to these periods. Briefly, periods with flows above $Q_{25}$ were classified as wet, periods with flows below $Q_{75}$ were defined as dry, increasing flows between $Q_{25}$ and $Q_{75}$ as wet-up period and receding between $Q_{25}$ and $Q_{75}$ as drying.*

Comment:

Fig. 8 and Sect. 5.3: How was the daily young water fraction calculated and what is the associated uncertainty? Are your interpretations robust with respect to the uncertainties in *Fyw*?

*Reply:*

*The daily young water fractions were extracted from the daily travel time distributions as the fraction of water volumes that is younger than 3 months (i.e. 32 days). Due to the lack of computational capacity we were unfortunately only able to provide uncertainty estimates for the long-term average $F_{yw,avg}$ as shown in Figure 8a and 8b in the original manuscript. We will clarify this.*

Comment:

L434: From Fig 2d it is hard to see how well the model simulated the δ18O time series because the data points cover each other too much.

*Reply:*

*We agree that Figure 2 will benefit from some adaptations. Specifically for Figure 2d we will therefore (1) aggregate the precipitation tracer composition to monthly values and (2) add an additional subplot with a zoomed-in version for one individual year pre- and one year post-deforestation.*

Comment:

Fig 8d, c: It is not clear to me, which data points were used to obtain these regression lines? Especially the dark-blue regression lines (wet conditions) do not seem to fit the blue data points at all, and thus, the associated regression slopes should be considered with caution (e.g. in L588).

*Reply:*

*We agree that the individual groups of data points are difficult to distinguish. To nevertheless allow the reader to better assess the strength of these relationships we will add the associated $R^2$ and p-values.*

References:

Bogena, H. R., Bol, R., Borchard, N., Brüggemann, N., Diekkrüger, B., Drüe, C., ... & Missong, A. (2015). A terrestrial observatory approach to the integrated investigation of the effects of deforestation on water, energy, and matter fluxes. Science China Earth Sciences, 58(1), 61-75.

Gao, H., Hrachowitz, M., Schymanski, S.J., Fenicia, F., Sriwongsitanon, N., & Savenije, H.H.G. (2014). Climate controls how ecosystems size the root zone storage capacity at catchment scale. Geophysical Research Letters, 41(22), 7916-7923.

Małoszewski, P., & Zuber, A. (1982). Determining the turnover time of groundwater systems with the aid of environmental tracers: 1. Models and their applicability. Journal of hydrology, 57(3-4), 207-231.

McGuire, K. J., & McDonnell, J. J. (2006). A review and evaluation of catchment transit time modeling. Journal of Hydrology, 330(3-4), 543-563.

---

## Author Response (AR2)

**Editor Comments**

Comment:

the reviewers clearly indicated a considerable improvement of the paper, however, one reviewer still has several points that should be addressed and looking at these points, I can only support the assessment of the reviewer. Hence, I would propose that the authors should careful examine the points and they can hopefully improve the paper regarding these observations.

*Reply:*

We appreciate the Editor's interest in our work and his additional efforts to help strengthening our work. We have addressed all remaining comments in detail below and revised the manuscript accordingly.

Comment:

Personally, I am also not happy, that the authors did not consider to use additional data collected in the Wüstebach catchment - certainly, there are only point data, but there is a wealth of point data and there has been many studies showing how to use such a wealth of data to improve the models. Uncertainty should not be an issue as discharge and isotope data as well as the forcing data are also uncertain which is not considered explicitly in the model either.

*Reply:*

*We thank the Editor for insisting on this point. Although very challenging, mostly for spatial commensurability issues, we believe that we have found an effective way of incorporating soil moisture and throughfall observations as additional constraints on the model. More specifically, we have included them as constraints following a simplified limits-of-acceptability approach. As such we believe that we managed to strike a reasonable balance to limit both Type I and Type II errors. Overall, the additional constraints proved to be very valuable to identify and reject unsuitable parameter sets. The model solutions retained as feasible do now not only simultaneously reproduce multiple system signatures (as quantified by 14 individual performance metrics) but also reasonably well mimic the observed time series of soil moisture and the average throughfall ratio $P_E/P$. Please note that this resulted in some minor changes in the results but did not affect the overall interpretation of the results.*

*(in the track-changed revised manuscript: p.5, l.160ff; p.13, l.396-406; p.16, l.482-486; p.18, l.542-543; p.22, l.693-694; Figure 7)*

Comment:

Interception - many studies show that interception in forest can make up easily 30% of actual ET, so why not considering this process explicitly and limiting I_Max after removing of the forest. I do not understand how Si(t) would be different after the forest removal when I_max is constant.

*Reply:*

*We completely agree with the Editor that interception evaporation can easily account for 30% of actual ET. We therefore decided already in the original analysis to explicitly represent this process in our model. That is why we are rather confused by the notion of the Editor that we kept the model parameter for interception capacity $I_{max}$ constant after deforestation. We of course explicitly accounted for a reduction of that storage capacity $I_{max}$ after deforestation, as would be expected. This was described, illustrated and discussed at length in all previous versions and also here, in the latest revised version of this manuscript. More specifically, the reduction of $I_{max}$ is shown in Figure 8 and Table 2, where it can be seen that, in particular in the riparian zone that was fully deforested, $I_{max,R}$ is reduced from 2.5 mm (1.9 – 4.8 mm) to 1.1 mm (0.1 – 1.3 mm). These effects are also described and discussed in some detail in the main text (in the track-changed revised manuscript: p.18, l.572-574; p.23, l.715-717; p.23, l.737-741, p.24, l.742-744; p.24, l.749-752).*

Comment:

Model performance with respect to isotope data: Does this mean that the value of the isotope data is rather limited after all? (assuming that if the fit is poor for the best parameterization, this criterion might not be that helpful in distinguishing between good and not so good parameter sets.

*Reply:*

*Although the Nash-Sutcliffe efficiency of the stream tracer signal is rather modest ($E_{NS}<0.4$) this does not entail that the model does a poor job in reproducing the tracer response: the modelled tracer composition remains much better than the mean of the observations. As there is very little fluctuation around this mean, with a variance of ~ 0.1‰, it is realistically seen close to impossible for a model to achieve much higher values of $E_{NS}$, given the limitations in the available data, as described in more detail in the direct reply to the reviewer below. In addition, the model does a rather good job in reproducing the second and here probably even more important aspect of the tracer response: the high level of damping between the precipitation and stream water tracer signals. To explicitly acknowledge and emphasize this, we have now added one more calibration objective function: the relative error between the observed and modelled damping ratios (see below for more detail; in the track-changed revised manuscript: p.13, l.393; p.14, l.413; p.16, l.487-489; p.17, l.527-529; p.18, l.551-554; Figure 6; Table 3).*

Comment:

Monte Carlo runs

*Reply:*

*The flux tracking module based on SAS-functions used to model the tracer response requires considerable computational power and is associated with very long run times. We have now added a further $2*10^6$ model realizations to a total of $3*10^6$ (in the track-changed revised manuscript: p.13, l.391). In spite of some minor improvements in terms of performance metrics, the additional parameter sets did not change the overall results nor the major conclusions of our analysis. We believe that this, together with*

*the relatively well constrained posterior distributions of most parameters as shown in Figure 8 and Table 2 of the revised manuscript provides a robust description of the feasible parameter space. For a more detailed response please see below the direct reply to the reviewer.*

Comment:

Due to the lengthy combined result and discussion section, I would also support that the results and discussion should be separated.

*Reply:*

*We have now separated results (in the track-changed revised manuscript: section 5, p.14-21) and discussion (section 6, p.21-27)*

Comment:

I would also like to ask the authors to provide a manuscript with track changes, so it is easier for me and the reviewer to follow the changes between the manuscript.

*Reply:*

*Such as in the previous round of revisions we have in detailed followed the HESS guidelines and again attached the track changed version of the revised manuscript at the end of this authors' response document and we have, in addition, now also uploaded an individual track-changed version.*

**Comments Reviewer #1**

Comment:

The authors have done a good job in implementing my suggestions and revising the paper. My only remaining concern is data accessibility. I have not found links or indications on how to access the catchment data and model data. I think the authors should provide such data.

*Reply:*

*We highly appreciate the reviewers' positive assessment. We have added this information in the revised manuscript.*

***Comments Reviewer #2***

Comment:

The authors took great care in revising the manuscript and replying to reviewers' comments. The manuscript goals are clearer and results are easier to follow. I believe the paper is now ready for publication on HESS.

*Reply:*

*We highly appreciate the reviewer's positive assessment of our manuscript.*

**Comments Reviewer #3**

Comment:

Overall the authors did a good job in revising the manuscript. The rebuttal reads a bit strange as it often says that something will be changed, but I assume this is just the text from the discussion phase and things actually have been changed by now. The manuscript has clearly improved by the revision, however, I have still a few points that should be considered.

*Reply:*

*We highly appreciate the overall positive assessment of the reviewer and thank him/her for the additional time invested for providing further valuable comments to strengthen the manuscript.*

Comment:

I appreciate the clarifications regarding the representation of interception. The authors argue that forest removal affects transpiration more than interception. While this obviously is case-specific I would, in general, maintain that the changed interception is the more obvious and often also more important change. While decreased transpiration will be partly compensated for by increased soil evaporation, interception is more of a 'completely lost process'.

*Reply:*

*We fully agree with the reviewer that after deforestation interception becomes almost a "lost process". This is illustrated in our results by the significant reduction of the model parameter representing the interception capacity. In particular, in the fully deforested riparian zone, the interception capacity is significantly reduced from of $I_{max,R}$ = 1.9 – 4.8 mm before deforestation to a value close to zero ($I_{max,R}$ = 0.1 – 1.3 mm) after deforestation, which indicates that this process is indeed almost "lost", as remarked by the reviewer (Figure 8, Table 2). In contrast, no discernible change in the interception capacity $I_{max,H}$ was found for the hillslopes of which only ~ 10% were deforested (Figure 8, Table 2).*

*We have further emphasized this in the revised version of the manuscript (in the track-changed revised manuscript: p.18, l.572-574; p.23, l.715-717; p.23, l.737-741, p.24, l.742-744; p.24, l.749-752).*

Comment:

I am not fully satisfied with the responses regarding passive water storage. Yes, water below the wilting point cannot be extracted by the plants, but for most of that water, it can be assumed that some mixing/exchange occurs with more mobile water. I am still wondering what it means if this is not represented in the modelling.

*Reply:*

*We fully agree with the reviewer, that although the water volumes below wilting point are not accessible to plants, they will experience some exchange with the more mobile water. Water stored below the wilting point will thus act as an additional passive mixing volume.*

*The overall effect of different mixing volumes on young water fractions however, critically depends on the magnitude of those individual volumes. For example, the passive mixing volume $S_{s,p}$ in the model (Figures 4, 8; p.12, l.363-370), which defines the hydrologically and hydraulically passive mixing volume of the groundwater and which is constant over time (i.e. $dS_{s,p}/dt = 0$), was here and in many tracer studies found to be several orders of magnitude larger than the hydrologically and hydraulically active volume $S_{s,a}$, which is variable over time (i.e. $dS_{s,a}/dt \neq 0$) – here: $S_{s,p} > 7600$ mm, Table 2, Figure 8; $S_{s,a} \sim 10 - 200$ mm (see also e.g. Birkel et al., 2010,2012,2014; Fenicia et al., 2010; McMillan et al., 2012; Hrachowitz et al., 2013,2015; Harman, 2015; Benettin et al., 2016,2017). This passive mixing volume $S_{s,p}$ therefore has a significant and discernible effect on the overall water volume $S_{s,tot} = S_{s,a} + S_{s,p}$ from which water outfluxes are "sampled" and thus on the age composition and consequently on the young water fractions of the outflows.*

*In contrast, the volume of water stored in the passive mixing volume below the wilting point is typically much lower than the volume of the more mobile water and thus also of the total mixing volume. More specifically, for silty clay loam soils such as in the Wuestebach catchment the water content at the wilting point is typically found at $\sim 0.1$ m³/m³, while the water content at field capacity is up to 4 times higher at $\sim 0.35 - 0.45$ m³/m³ (e.g. Romano and Santini, 2002). The hydrologically and hydraulically active mixing volume combines the volumes between wilting point and field capacity as well as the transient water volume above field capacity that can eventually not be held against gravity. Depending on the wetness conditions, this active mixing volume can in the extreme case be up to 10 times (under fully saturated conditions) higher than the passive mixing volume and still $\sim 4$ times higher under average conditions when field capacity is reached. Water ages and in particular the young water fractions in the unsaturated zone are therefore largely controlled by the larger, hydrologically active mixing volume. This is further exacerbated by the fact that under drier conditions, i.e. when the soil water content approaches the water content at the wilting point, by definition the fraction of young water is very low in any case, as the water released from the system cannot have entered the system very recently under such dry conditions. In addition, the rather tightly bound water below the wilting point will, due to the strong adhesive forces close to the surface of the soil particles, only experience rather low diffusive exchange rates with mobile water. By extension, this implies that water volumes stored below the wilting point are, on average, characterized by rather old ages, which will only have very limited effect on the young water fractions (i.e. younger than 3 months) analysed here.*

*Altogether this implies that while omitting this exchange in the model will cause a slight under-/over-estimation of older ages, the absolute effects of this omission will be very minor: under very dry situations when a proportionally larger effect of exchange with water stored below the wilting point may be expected (as the water volume in the passive store is then larger than the volume in the active store), the fractions of young water, which are analysed here in this manuscript, are close to zero (Figure 9c,d,j; Figure 10) and the absolute effect of the passive storage volume below wilting point cannot be meaningfully discerned anymore.*

*In Figure R1 hereafter we provide a simplified illustration of the effects of including the water content below the wilting point as an additional passive mixing volume on the estimates of young water fractions $F_{yw}$, i.e. water younger than 3 months, under (a) very wet and under (b) dry conditions. As can be seen, the differences are rather limited.*

*We have clarified this and also briefly discussed the potential effects of this in the revised version of the manuscript (p.12, l.369-372).*

[Figure]

*Figure R1: Fractions of young water $F_{yw}$ released from (a) the active mixing volume of the soil storage $S_a$ under wet conditions after a few weeks of rainfall, i.e. soil storage filled to field capacity, with most of the water being younger than 3 months due to recent rainfall. Outflow released from that example is sampled with a SAS-function with strong preference for young water following a Beta-distribution ($\alpha=0.75$, $\beta=1$; Eq.40 in the manuscript). (b) under wet conditions accounting for the additional passive mixing volume $S_p$ below the wilting point, assuming a volume ratio of 4:1 between the total water content $S_a+S_p$ and the water content below wilting point $S_p$, thereby reflecting the soil hydraulic properties of silty clay loams. Due to the rather low diffusive exchange rates between $S_a$ and $S_p$, the fraction of water < 3 months is limited in $S_p$. The same SAS-function to sample water of different ages as in (a) is used. However, in this case the outflows are sampled from the total volume $S_a+S_p$. (c) shows the situation under dry conditions without considering the passive mixing volume $S_p$ below the wilting point. After a prolonged dry period, only a small fraction of water in the soil is younger < 3months, as it did not rain for several weeks. Under such dry conditions the preference for sampling young water is reduced following Eq.40 in the manuscript and the SAS-function therefore converges towards a uniform distribution, i.e. a Beta-distribution with parameters $\alpha=1$ and $\beta=1$. (d) shows dry conditions, sampling water according to the same SAS-function as in (c), but explicitly considering the passive mixing volume below the wilting point equivalent to (b). Note, that in this illustrative example $F_{yw}$ is higher under wet conditions than in the results reported in the manuscript. The reason for this is that here, in contrast to the results in the manuscript, $F_{yw}$ describe exclusively the time until release from the soil storage and before having travelled through the groundwater reservoir.*

Comment:

Looking closer at the model performance with respect to the isotope data (Fig 3), I am wondering whether the model really can be said to represent these data. The authors also discuss this issue and report some rather low model performance values. Modelling isotopes is hard, so I don't want to blame the authors here, but as the novelty of this study is the inclusion of isotope data in the modelling, I am a bit puzzled by these poor model fits. Does this mean that the value of the isotope data is rather limited after all? (assuming that if the fit is poor for the best parameterization, this criterion might not be that helpful in distinguishing between good and not so good parameter sets.

*Reply:*

*We highly appreciate this comment and there are several complementary points in response to this comment that we would like to clarify.*

*It is true that the model provides only a rather modest representation of the high-frequency dynamics of the stream tracer concentration, as illustrated by the values of Nash-Sutcliffe efficiencies $E_{NS}$ < 0.4. The limited ability of models to reproduce high-frequency dynamics of tracer compositions in stream water, in particular in strongly dampened systems, is widely reported in a wide range of environments, and reflected by Nash-Sutcliffe efficiencies of modelled stream tracer concentration that, similar to our study, very rarely exceed values of $E_{NS}$ ~ 0.4 (e.g. Page et al., 2007, Shaw et al., 2008; Birkel et al., 2010,2011a,b; Fenicia et al., 2010; McMillan et al., 2012; Hrachowitz et al., 2013; Benettin et al., 2015, 2016; Birkel and Soulsby, 2016; van Huijgevoort et al., 2016; Ala-aho et al., 2017; etc.).*

*However, the second important feature describing the tracer storage and release properties of this catchment is very well captured in this study: the degree to which the seasonal amplitudes in the precipitation tracer are dampened in the stream. This system property reflects the buffer function of the catchment. The model's ability to closely reproduce this buffer function is evidence that the model meaningfully represents the low pass filter characteristics, and thus the major characteristics of solute transport in the study catchment. This is also the reason why we decided to show the precipitation and stream water tracer signals on the same y-axis scale in Figure 3a, as this allows to visually appreciate the strong degree of dampening between the precipitation and stream signals in the study catchment and the model's ability to reproduce that. The damping ratio $R_D$, here expressed as the ratio of the standard deviation of the stream water signal to the standard deviation of the precipitation signal, describes this damping effect. The modelled dampening ratio matches the observed dampening ratio for most solutions very well, as indicated by the error metric $E_{RD,δO18}$= 1-$R_{D,mod}$/$R_{D,obs}$, which remains > 0.95 with a value of 1 indicating a perfect fit. This further underlines the model's ability to reproduce this critical characteristic of the study catchment. To better and explicitly emphasize the relevance of this catchment signature, we have added the relative error between the observed and the modelled damping ratio in the revised manuscript as additional, 14$^{th}$ objective function to the analysis (Table 3; Figure 6)*

*To further clarify the interpretation of the Nash-Sutcliffe efficiency $E_{NS}$, it is critical to see that this error metric essentially evaluates how much better a modelled variable is than the mean of the observed values. This implies that the lower the variance around that mean in the observed data is, the more difficult it is for a model to provide a better fit than the mean. This is because the mean provides already a rather strong representation of the signal. In contrast, for signals with much more pronounced amplitudes (and thus higher variance), it is easier for models to generate high values of $E_{NS}$. In the*

*following we illustrate this with a illustrative hypothetical example of two signals with considerably distinct amplitudes (Figure R2 here below). It can be seen that in the example with the high amplitudes, the $E_{NS}$ of the modelled signal reaches $E_{NS}$ = 0.83 while the mean squared error reaches MSE = 10.6. However, in the second example, where the variance in the observed signal is much lower, such that the signal itself plots very close around the mean, $E_{NS}$ = 0.13 and thus much lower than before, while according to MSE = 0.5 the model provides a much better fit. In other words, already very small errors can cause dramatic reductions of $E_{NS}$ in low-amplitude (and thus low variance) signals, while very high values of $E_{NS}$ can be sustained with much larger errors in high-amplitude signals, as these higher errors still allow improvements as compared to the mean. This further implies that the magnitude of $E_{NS}$ always needs to be interpreted together with the magnitude of the variance in the signal. In the specific case of the stream tracer signal in the analysis in our manuscript, we observe a very high degree of damping. The stream tracer compositions, both observed and modelled, therefore show only very limited variance around the mean, indicating that the absolute errors in the model are very low, but also that it is very difficult for a model to perform much better than the mean. Thus, while it is indeed questionable to use $E_{NS}$ as metric to compare signals from different systems, $E_{NS}$ is nevertheless useful to compare different models and to identify feasible parameter sets in one single system.*

[Figure]

*Figure R2: the dark line indicates an observed high-amplitude signal (amplitude = 10; left) and on observed low-amplitude signal (amplitude = 0.5; right). The light blue lines indicate models for these observations with high absolute errors (and thus also the squared errors MSE) for the high-amplitude signal (MSE=10.6; left) and much lower absolute errors for the low-amplitude signal (MSE=0.5). However, in spite of this, $E_{NS}$ remains much higher for the high-amplitude ($E_{NS}$=0.83) than for the low amplitude signal ($E_{NS}$=0.13)*

*The rather small variability around the mean in the observed stream tracer signal introduces an additional challenge, related to the information content of the available data: the fluctuations around the mean are very close to the observational uncertainty as the analytical precision of the spectrometer used for the isotope analysis is ~0.1‰ while the variance in the observed (and modelled) signals is equally ~ 0.1‰. Much of the observed fluctuations may therefore merely be errors/uncertainties in the observations, which the model cannot and should not reproduce. Another issue with respect to the information content in the available data is the fact that here, such as in the vast majority of tracer studies, precipitation isotope data were available as weekly bulk samples, while stream water samples were taken as instantaneous grab samples. This entails that the weekly bulk samples average out the*

*effects of potential extreme events, while the variability in the stream water composition from grab samples may reflect these events. As the model is forced with bulk samples taken at weekly resolutions, it can represent fluctuations at any higher frequencies only to a very limited degree.*

*Besides the effect of data on $E_{NS}$, please also note that we used an extended multi-objective model calibration and evaluation strategy based on 14 performance metrics. As such, the modelled output of any parameter combination had to simultaneously exceed thresholds of all 14 objective functions and, in addition, satisfy two newly added limits-of-acceptability constraints to ensure a meaningful representation of throughfall and soil moisture (for more detail please see replies to the Editor). Parameter combinations that failed to do so were discarded. If we had used a common calibration strategy based on only one or 2 calibration objectives, such as $E_{NS,Q}$ and $E_{NS,\delta O18}$, as is used in the vast majority of research papers, most of the discarded solutions would have been wrongly kept as feasible (although they cannot reproduce any of the other objectives), many of which will potentially have misleadingly resulted in $E_{NS,Q}$ and $E_{NS,\delta O18}$ much higher than the ones obtained here – we simply would have obtained the right results for the wrong reasons (cf. Kirchner, 2006). This phenomenon of pareto optimal solutions is well known and exhaustively described in literature (e.g Gupta et al., 1998; Yapo et al., 1998; Vrugt et al., 2003; Efstratiadis and Koutsoyiannis, 2010; Hrachowitz et al., 2014).*

*Overall, we have exposed our model to much more rigorous model constraints (i.e. 14 objective functions and 2 limits-of-acceptability constraints) than is done in most research papers. This resulted in the rejection of deceptively high-performance models, in favour of somewhat lower performance with respect to individual performance metrics, but which ensure a much higher internal consistency of the model and thus provide more reliable representations of real world dynamics.*

*Following the reviewer's observation and in a further step to improve the robustness of our model and to emphasize the relevance of the degree of damping in the isotope signal we have now added with the relative error of the damping ratio one more model objective function. This allowed us to move the emphasis away from considering only the isotope dynamics as evaluated by $E_{NS,\delta O18}$ to also explicitly consider the buffer effect of the catchment in a separate evaluation metric. As can be seen, this metric, the relative error in the damping ratio remains very well reproduced in all model cases with $E_{RD,\delta O18} = 0.95 - 0.99$ (in the track-changed revised manuscript: p.13, l.393; p.14, l.413; p.16, l.487-489; p.17, l.527-529; p.18, l.551-554; Figure 6; Table 3)*

Comment:

Term effective precipitation: the authors use this term not in its usual way. Usually, effective precipitation is the part of precipitation that makes it to groundwater (=recharge), e.g., http://www.fao.org […]. Here it is the precipitation minus interception (but not minus other evaporative fluxes!). I would recommend avoiding this term totally to prevent confusion.

*Reply:*

*Agreed, we have replaced the term "effective precipitation" by "throughfall" in the revised manuscript.*

Comment:

Monte Carlo runs: while 10e6 sounds like a lot, with 14 parameters this means just about 2.7 values along each (parameter)dimensions, i.e., the parameter spaces is sampled rather scarcely. This is an issue of many MC studies, but these days, much larger numbers of runs should be possible, why only one million? This was the standard years ago. Today more runs should be doable, any reason for using one million?

*Reply:*

*Indeed, under-sampling can become a limiting issue in inverse modelling approaches.*

*The challenge we were facing here was that process-based hydrological models which are coupled with tracer modules based on the SAS-function approach are computationally very demanding and require very long run-times, which is also one of their major limitations (cf. personal communication Paolo Benettin). These can be traced back to two major reasons: (1) for each time step SAS-functions need to be generated from programming language internal functions of the associated probability distributions, which is a comparably time-consuming process, and even more important (2) to avoid numerical instabilities related to problems in maintaining water and tracer mass balances, which occur when SAS-functions dictate to sample a higher proportion of a specific age than is actually present in a storage component. In such a case, the water ages and the relates water and tracer masses need to be accordingly redistributed and resampled in an iterative process, which can lead to long running loops in the code and which often are very time-consuming. In our specific case, $10^6$ model realizations require a model run time of approximately 1.5 months(!).*

*The initially $10^6$ model realizations led to most parameters being reasonably well identified between rather limited ranges as illustrated in Figure 8 and Table 2. It is thus not implausible to assume that these limited ranges largely delimit the region of the parameter space that contains the optimal parameter combination. Thus even if the actual optimal solution is not found, there will be only limited deviation from the solutions found and kept as feasible in this analysis.*

*To strengthen the model we have followed the suggestion of the reviewer and have for the revised version of the manuscript increased the number of model realizations by 200% to a total of $3*10^6$ (in the track-changed revised manuscript: p.13, l.391). Given the computational resources required by the model we unfortunately do not have the capacity to further extend this analysis. In any case and notwithstanding some minor improvements in terms of performance metrics, the additional parameter sets did not change the overall results nor the major conclusions of our analysis. However, we believe that the significantly increased number of realizations together with the rather well identified parameters allow a meaningful, albeit quite clearly not complete, exploration of the parameter hyperspace.*

Comment:

Personally, I would prefer if results and discussion could be separated, this could help to make the manuscript more readable

*Reply:*

*We have now separated results (in the track-changed revised manuscript: section 5, p.14-21) and discussion (section 6, p.21-27)*

Comment:

Eq 42 looks fancy, but I find this a bit confusing, there is only one measure related to O-18, or?

*Reply:*

*Yes, indeed, Eq.42 initially only considered one metric for O-18. We have now added a second one: the relative error of the damping ratio (see also reply above; in the track-changed revised manuscript: p.14, l.413; Table 3; Figure 6).*

[revised manuscript text omitted]